# A multivalent adaptor mechanism drives the nuclear import of proteasomes

Hanna L. Brunner [1,2], Robert W. Kalis [1,2,6,7,8], Lorenz Grundmann [1,2,8], Zuzana Hodáková [1], Zuzana Koskova [3], Irina Grishkovskaya [1], Melanie de Almeida[1], Matthias Hinterndorfer [1], Hannah Knaudt [1,2], Simon Höfflin[1], Florian Andersch [1], Harald Kotisch[1], Achim Dickmanns[4], Sara Cuylen-Haering [3], Johannes Zuber [1,5] ✉ & David Haselbach [1] ✉

Nuclear protein homeostasis, including transcription factor turnover, critically depends on the nuclear proteasomes that must be imported after cell division. This dynamic process requires AKIRIN2, a small unstructured protein whose mechanistic role has remained elusive despite its essential function. Using an integrated approach combining protein-wide saturation mutagenesis screens, cryo-EM, and biochemical reconstitution, we characterize AKIRIN2 as a scaffold protein that coordinates the assembly of an importin cluster around the proteasome. AKIRIN2 binds in multiple copies to the 20S proteasome and simultaneously interacts with importin IPO9 and the KPNA2/KPNB1 heterodimer. In the nucleus, RanGTP triggers importin dissociation, releasing the proteasome, while AKIRIN2 undergoes ubiquitin-independent degradation. Our findings reveal how AKIRIN2's multivalency facilitates the recruitment of multiple importins to the proteasome, a critical adaptation for transporting this large macromolecular complex into the nucleus and maintaining the nuclear proteome.

Thousands of proteins shuttle between the cytoplasm and the nucleus. This process is controlled by the nuclear pore complex (NPC), which acts as a selective gateway for nucleocytoplasmic transport. Current models of nuclear import describe how importin-β family proteins recognize motifs such as nuclear localisation signals (NLS) on cargo proteins and facilitate their passage through the NPC[1,2]. However, the mechanism by which large assemblies and complexes efficiently traverse the NPC to enter the nucleus is more intricate. Although studies on viral particle entry suggest that cargo size correlates with the number of importins required for effective import, this phenomenon has not been comprehensively explored for endogenous complexes[3–5].

Large complexes with critical nuclear functions include 20S and 26S proteasomes, which are essential for nuclear protein homeostasis and the regulation of fundamental processes such as the cell cycle and transcription. Despite the high abundance of proteasomes in the nucleus, and clear evidence that assembled proteasomes are actively transported through the NPC, the mechanisms governing their import have long remained elusive in higher eukaryotes[6–8]. In yeast, the adapter protein Sts1 is crucial for proteasome nuclear import, yet no homolog exists in vertebrates[9,10]. In *Xenopus*, studies demonstrated active import of 20S proteasomes associated with factors like HSP90 and the 19S subunit Rpn2[11], though their functional role remains unclear.

[1]Research Institute of Molecular Pathology (IMP), Vienna BioCenter (VBC), Vienna, Austria. [2]Vienna BioCenter PhD Program, Doctoral School of the University of Vienna and Medical University of Vienna, Vienna, Austria. [3]European Molecular Biology Laboratory (EMBL), Heidelberg, Germany. [4]Department of Molecular Structural Biology, Institute of Microbiology and Genetics, GZMB, Georg-August-University Göttingen, Göttingen, Germany. [5]Medical University of Vienna, VBC, Vienna, Austria. [6]Present address: Department of Medical Oncology, Dana-Farber Cancer Institute, Boston, MA, USA. [7]Present address: Broad Institute of MIT and Harvard, Cambridge, MA, USA. [8]These authors contributed equally: Robert W. Kalis, Lorenz Grundmann. ✉e-mail: johannes.zuber@imp.ac.at; david.haselbach@imp.ac.at

We recently uncovered AKIRIN2, a highly conserved and essential protein, as a key mediator of proteasome import[12]. Cells dividing in the absence of AKIRIN2 fail to import proteasomes into the nucleus. Consequently, nuclear factors accumulate, and cells rapidly undergo apoptotic cell death, demonstrating that AKIRIN2 is strictly required for nuclear proteasome import. Consistent findings in *C. elegans* involving the homolog AKIR-1 have confirmed a conserved nuclear import mechanism across species[13]. However, how this small protein mediates the import of these large protein complexes has remained elusive.

Here, we establish a comprehensive map of functionally relevant residues and binding interfaces of AKIRIN2's structured and disordered protein regions by surveying every possible single amino acid substitution in AKIRIN2 using FACS- and microscopy-based genetic screens. Integrating these results with cryo-EM analysis reveals several binding interfaces, including a wing helix in a disordered region of AKIRIN2 that stabilize interactions with the 20S proteasome. Upon primary binding, AKIRIN2 homodimers recruit the importin IPO9, which in turn facilitates the binding of a second AKIRIN2 homodimer that recruits additional importins. Together, this multivalent molecular assembly amplifies the number of nuclear localisation signals and, thereby, triggers efficient proteasome translocation into the nucleus.

## Results

### Saturation mutagenesis screens identify crucial features of AKIRIN2

Human AKIRIN2 was found to directly interact with mature 20S and 26S proteasome particles via a conserved YVS motif. Further, it contains a bipartite classical nuclear localisation sequence (cNLS) and a C-terminal coiled-coil dimerization domain[12]. While our prior structural work provided insights into interactions between AKIRIN2 and the proteasome, 76% remained structurally unresolved due to its flexible nature (Fig. 1a). To gain insights into molecular functions and underlying features of AKIRIN2, we devised a series of saturation mutagenesis screens (SMScreens) in which we systematically probed the functional relevance of each amino acid for nuclear proteasome import. To this end, we constructed a lentiviral cDNA library containing 4,500 AKIRIN2 variants harbouring a specific mutation that either deletes a specific codon or converts it into an optimized codon of each of the other 19 amino acid or a STOP codon.

To quantify the effect of these variants on nuclear proteasome import, we engineered iCas9.RKO cells[12] to express two complementary reporter alleles. First, we replaced an endogenous subunit of the 20S proteasome with a fluorescently tagged version (PSMB4-StayGold) to enable the monitoring of proteasome localisation through live-cell microscopy. Second, we stably expressed an mCherry-MYC fusion protein which serves as a highly dynamic reporter. As shown previously, the transcription factor MYC is rapidly degraded under normal conditions by nuclear proteasomes[12], making it a sensitive indicator of proteasomal activity in the nucleus. In addition, we stably expressed a pair of single guide RNAs (sgRNAs) targeting AKIRIN2, which together with doxycycline (dox)-inducible Cas9 (iCas9) enable efficient, temporally controlled knockout of endogenous AKIRIN2 (Fig. 1b, Supplementary Fig. 1a,b). Following cDNA library transduction, selection, and dox-induced depletion of endogenous AKIRIN2, we quantified rescue effects mediated by sgRNA-resistant AKIRIN2 variants contained in our library using three complementary high-throughput assays: (I) Using fluorescence-activated cell sorting (FACS), we isolated cells displaying elevated mCherry-MYC levels, indicating dysfunctional proteasome import. (II) We employed a prototype of a high-speed image-enabled cell sorting device[14] to isolate cells displaying reduced nuclear proteasome content based on our PSMB4-StayGold reporter. (III) Since AKIRIN2 is essential for cell survival, we also performed a proliferation-based screen, where cells

expressing dysfunctional AKIRIN2 variants are expected to drop out over time. In all three screens, we quantified the relative representation of each AKIRIN2 variant using next-generation sequencing of cDNA cassettes PCR-amplified from genomic DNA of sampled cells (Fig. 1b).

Despite differences in dynamic range and resolution, all three screening approaches yielded highly consistent results and reliably detected functionally relevant features of AKIRIN2. A merged analysis of the data from all screens (Fig. 1c, d, Supplementary Fig. 1c-e) shows that, in line with the critical role of the C-terminal YVS-motif for proteasome binding, generally STOP-variants and mutations within YVS were identified as dysfunctional, while amino acid self-replacements with alternative codons remained broadly neutral. Similarly, our screens confirmed the critical role of the N-terminal bipartite cNLS. Moreover, deletions or helix-breaking prolines within the coiled-coil domain (residues 142-190) were detrimental to AKIRIN2 function, likely due to the disruption of the domains' essential role in homo-dimerization. Specific residues mediating key interactions between two AKIRIN2 protomers (A and B) included C152$^A$:C152$^B$ (disulfide bond), R159$^A$:E160$^B$, E160$^A$:R159$^B$, Y167$^A$:Y167$^B$ (π-π), L171$^A$:L171$^B$, and F182$^A$:F185$^B$ (π-π) (Fig. 1c, d). In addition, our SMScreens revealed several previously unknown functionally relevant regions. To validate these findings, we individually tested dysfunctional variants by introducing single point mutations into our reporter cell line. We assessed their functional impact by imaging proteasome subcellular localization and mCherry-MYC reporter signals with confocal microscopy, and western blot analysis to quantify AKIRIN2 protein levels (Supplementary Fig. 2a-c). We found a critical domain spanning residues 78-101, which we termed the wing helix. Deletion of this helix or mutations in the residues E79, I85, K94 severely impaired AKIRIN2 function, suggesting its importance in protein-protein interactions or regulatory roles (Fig. 1c, d, Supplementary Fig. 2a-c). Moreover, we detected a striking intolerance to mutations at residue D188 (Supplementary Fig. 2a-c), hinting at a critical, yet undefined function. This residue does not appear to be involved in dimerization or proteasome binding, suggesting a role in interactions with other factors. Our SMScreens reveal a detailed functional map of AKIRIN2, uncovering numerous critical interaction sites beyond its established NLS and proteasome-binding motifs, suggesting that previously unrecognized residues may serve as regulatory or interaction points involved in coordinating the proteasome and nuclear import machinery.

### IPO9 is a key component of AKIRIN2-mediated proteasome nuclear import

We previously demonstrated that AKIRIN2 interacts with several importins and therefore sought to identify additional proteins involved in AKIRIN2-mediated nuclear import. Knockout of KPNA2 or KPNB1 did not affect nuclear proteasome localisation (Fig. 2a, Supplementary Fig. 3a). However, the redundancy among the six human importin-αs suggests functional compensation among these proteins. In the case of KPNB1, crucial for many canonical import processes, its knockout led to increased MYC levels and rapid cell death before mitosis could even occur. Most promising was the deletion of IPO9, which resulted in a phenotype similar to AKIRIN2 knockout (Fig. 2a, Supplementary Fig. 3a), though with a delayed onset. To investigate this further, we focused on potential interactions with IPO9. Co-immunoprecipitation coupled with mass spectrometry (Co-IP/MS) identified AKIRIN2 as one of the most enriched interactors of IPO9 (Fig. 2b), along with known partners like Ran and histones H2A/H2B (Proteomics data have been deposited in PRIDE under accession: PXD052012). These results suggest that IPO9 is a major, though not exclusive, component of the proteasomal import pathway.

We next examined direct interactions between 20S proteasome, AKIRIN2, and IPO9 using in vitro reconstitution in sucrose density gradients (Fig. 2c, Supplementary Fig. 3b). We focused here on 20S proteasomes to reduce the complexity. IPO9 alone failed to bind the

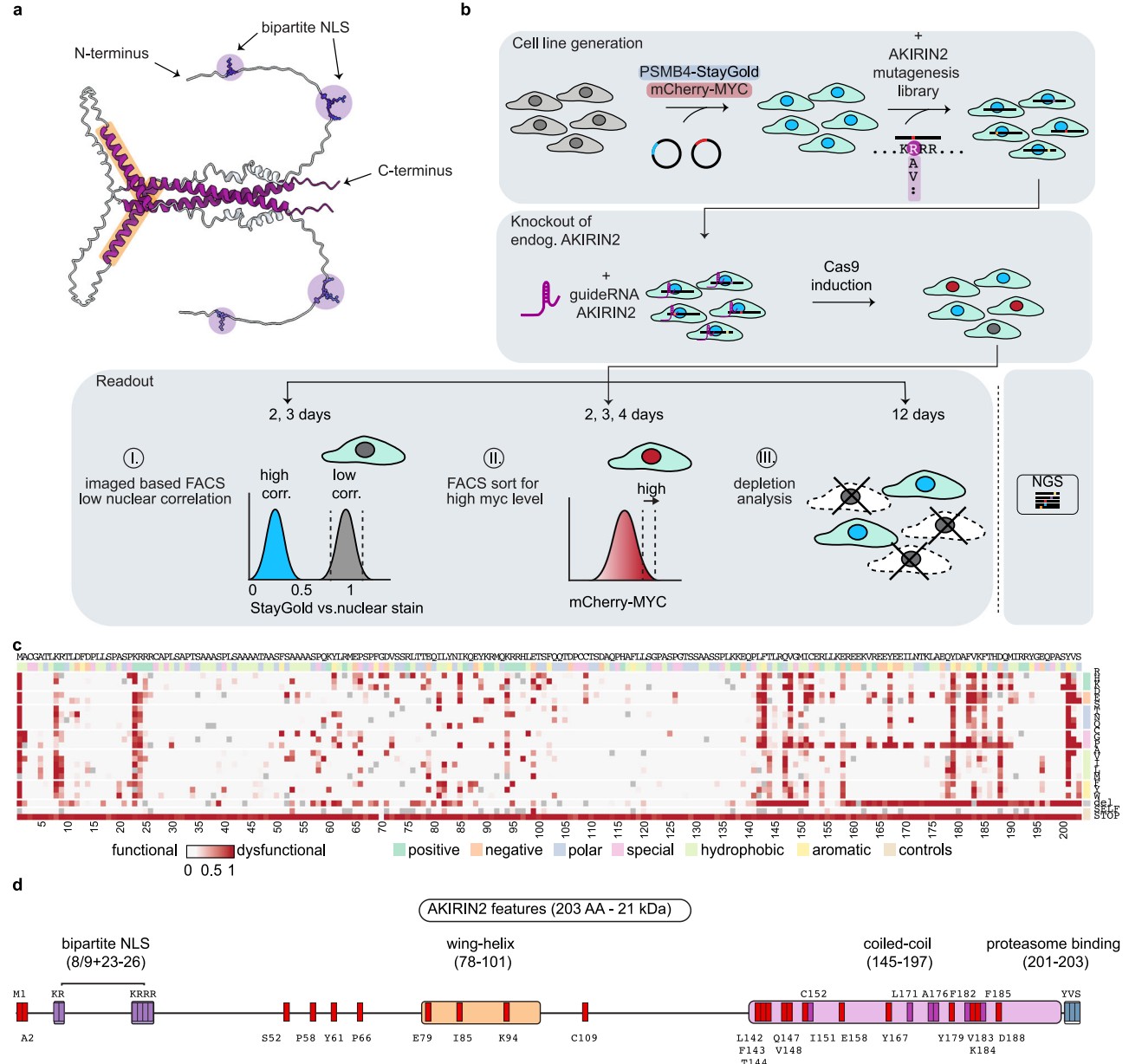

**Fig. 1 | Saturation mutagenesis screens (SMScreens) map functional importance of AKIRIN2 residues. a** AlphaFold3 predicted a dimeric structure of AKIRIN2 showing the coiled-coil dimerization domain (dark pink), wing α-helices (dark pink highlighted with orange) and the bipartite cNLSs (purple). **b** Schematic representation of the generation of the 20S reporter cell line expressing StayGold-labelled proteasomes and mCherry-MYC, transduced with an AKIRIN2 mutagenesis library and a dual-single guide (dsg) AKIRIN2 for the knockout of endogenous AKIRIN2. Followed by a schematic workflow of the screen starting with Cas9 induction for 2-4 days before sorting either for: I. low-correlation between nuclear staining and StayGold-proteasome or, II. high mCherry levels, additionally III. long-

term dropout cells after 12 days are analysed. Subsequently, all samples are sent for next generation sequencing (NGS) to quantify mutants. **c** Heat plot representing cumulative analysis of the three individual readouts of the SMScreens results, showing dysfunctional mutations in red shades. **d** Schematic representation of AKIRIN2 features, coloured as in (**a**). Residues that emerged to be crucial but with unassigned function are marked with red rectangles, residues that were assigned to be important for dimerization by AlphaFold3 predictions or cryo-EM structure are marked with purple rectangles, blue rectangles mark residues known to be essential for proteasome binding, dark purple rectangles mark the bipartite cNLS.

proteasome, but addition of AKIRIN2 led to formation of a ternary complex (Fig. 2c, Supplementary Fig. 3b-d). Additionally, we observed the formation of binary AKIRIN2-IPO9 complexes with a fraction that tends to precipitate, hinting towards oligomerisation of the proteins (Supplementary Fig. 3c). AKIRIN2 alone was distributed across multiple gradient fractions, indicating the formation of higher oligomers. Quantitative binding assays revealed a low μM binding affinity for AKIRIN2-proteasome and a higher affinity interaction between AKIRIN2 and IPO9 ($K_D \approx 40$ nM) (Supplementary Fig. 3e,f). These findings establish AKIRIN2 as a crucial adaptor that anchors IPO9

to the proteasome, forming a key component of the nuclear import complex.

## Structural basis of AKIRIN2-mediated proteasome-importin assembly

To elucidate the molecular architecture of the reconstituted 20S:AKIRIN2:IPO9 complex, we performed electron cryo microscopy (cryo-EM), focusing on particles with additional density compared to 20S:AKIRIN2 (pdb: 7NHT) alone. The resulting 3.4 Å resolution map (Supplementary Fig. 4, Supplementary Fig. 5a-e) revealed an

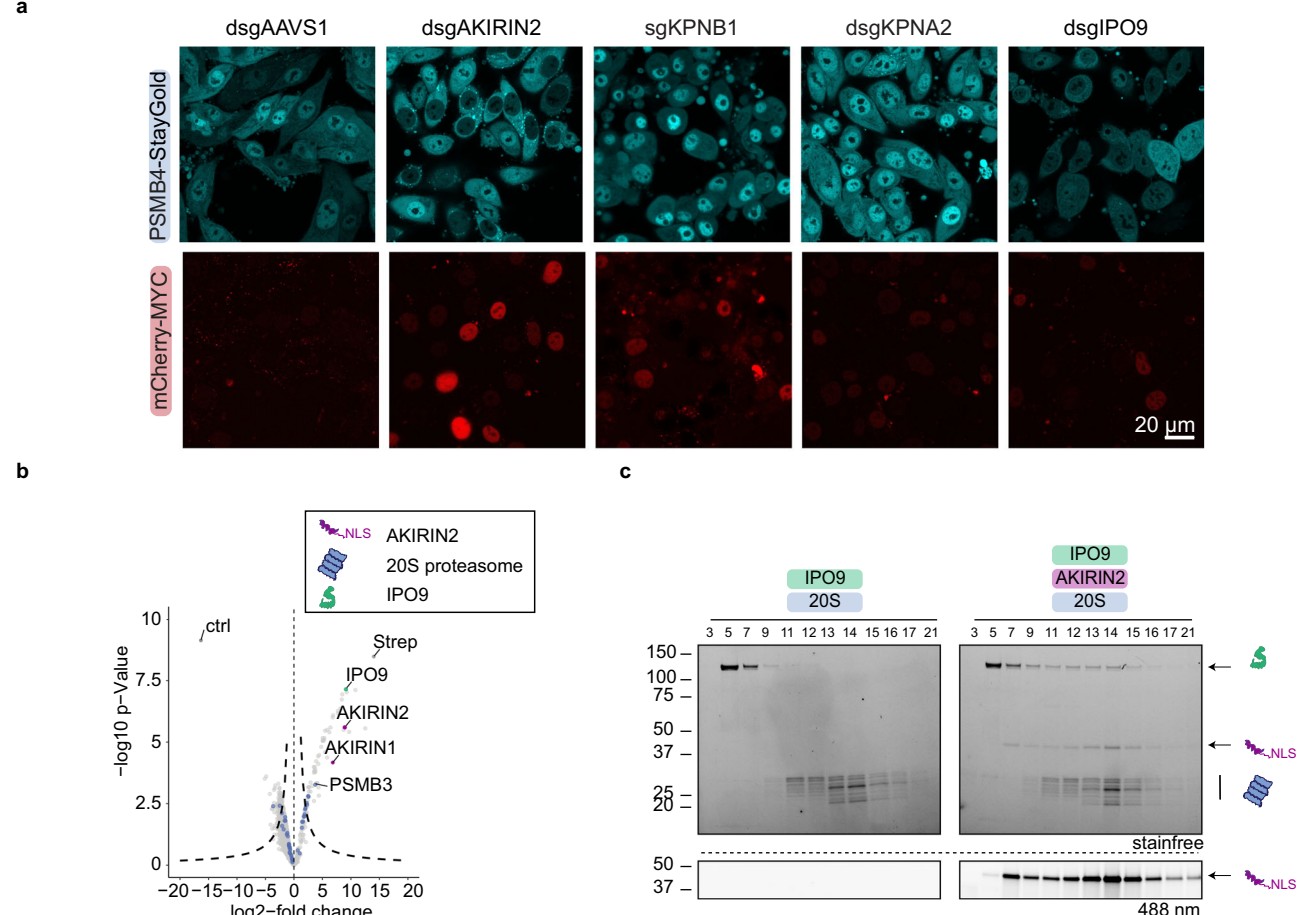

**Fig. 2 | Importin IPO9 engages proteasomes via AKIRIN2. a** Representative confocal image (scale bar: 20 μm) of StayGold-PSMB4 RKO reporter cell line showing localisation of proteasomes as well as mCherry-MYC after 2- or 5-days knockout induction (depending on protein half-life) of AAVS1 (2 days), AKIRIN2 (2 days), KPNA2 (5 days), KPNB1 (5 days) or IPO9 (5 days). **b** Volcano plot of Co-IP/MS of Strep-tagged-IPO9 in HeLa extract. **c** SDS-PAGE Biorad TGX stainfree gel of fractionated 10–30% sucrose density gradient showing proteasome and IPO9 distribution in the absence (left) or presence (right) of AKIRIN2. Fractions are arranged from left to right in order of increasing density. The lower panels display the corresponding GFP signal of the GFP-AKIRIN2 fusion protein in the same gradient fractions.

unexpected arrangement: two AKIRIN2 dimers (I and II) and one IPO9 molecule bound to the same side of the 20S proteasome (Fig. 3a, b). IPO9 adopts its characteristic HEAT repeat solenoid structure, but interestingly engages AKIRIN2 through a unique interface. The C-terminal HEAT repeats H17-H18 bind the coiled-coil of AKIRIN2 dimer I on the outer surface of IPO9's concave structure (Fig. 3a, c) mediated by several residues identified in our SMScreens as crucial (Q178, Y179, F182, V183, F185, D188, M191). This binding mode contrasts with interactions of IPO9 with other known partners like H2A/H2B or RanGTP, which bind the inner concave surface[15].

Although IPO9 binding does not significantly alter the interaction of AKIRIN2 dimer I with the proteasome, we can now resolve the wing helix of protomer AKIRIN2[A] raising the total resolved part of the sequence to 44%. This reveals additional contacts, particularly the functional important residue K94, which binds to a negatively charged pocket at the C-terminal end of the α3 (PSMA3) subunit (Fig. 3d). This identified binding interface likely stabilizes the interaction with the proteasome.

The coiled-coil of AKIRIN2 dimer II is sandwiched between IPO9 and the periphery of the alpha ring of the proteasome, with its N-terminal part contacting IPO9's HEAT repeat H1 and its C-terminus engaging HEAT repeats H13-14 (Fig. 3b, e). Notably, we resolved the wing helix of AKIRIN2[C] where a basic patch (residues: 86-95) interacts with the negatively charged surface of α6 (PSMA6) (Fig. 3e). Also here,

the crucial residue K94 appears to be important for this interaction. Lastly, despite the limited resolution in this region, three glutamate residues (E165, E166, E169) on AKIRIN2[D] are positioned near a putative NLS (K181, K184, K185) of α6 (PSMA6)[16], suggesting potential salt bridge formation (Fig. 3f). This interaction may explain the previously described[16] crucial role of this motif in nuclear localisation of the proteasome, although direct interactions with importins were not detected.

Strikingly, mapping the SMScreen results onto our ternary complex structure shows that functionally crucial residues cluster at the interaction interfaces with IPO9 and the 20S proteasome (Supplementary Movie 1). Our structure reveals at least four AKIRIN2 molecules bound to one side of the 20S proteasome, each of them exposing a bipartite NLS, facilitating transport through their accumulation. These findings demonstrate that AKIRIN2 functions as a multivalent adaptor, nucleating the assembly of a large nuclear import complex through specific protein-protein interactions.

## Conformational plasticity of AKIRIN2 mediates complex assembly

The observed clustering of AKIRIN2 on the proteasome in the presence of IPO9 prompted us to investigate whether this arrangement is specific to the ternary complex or a general feature of AKIRIN2-IPO9 interactions. We determined the structure of the AKIRIN2-IPO9 binary

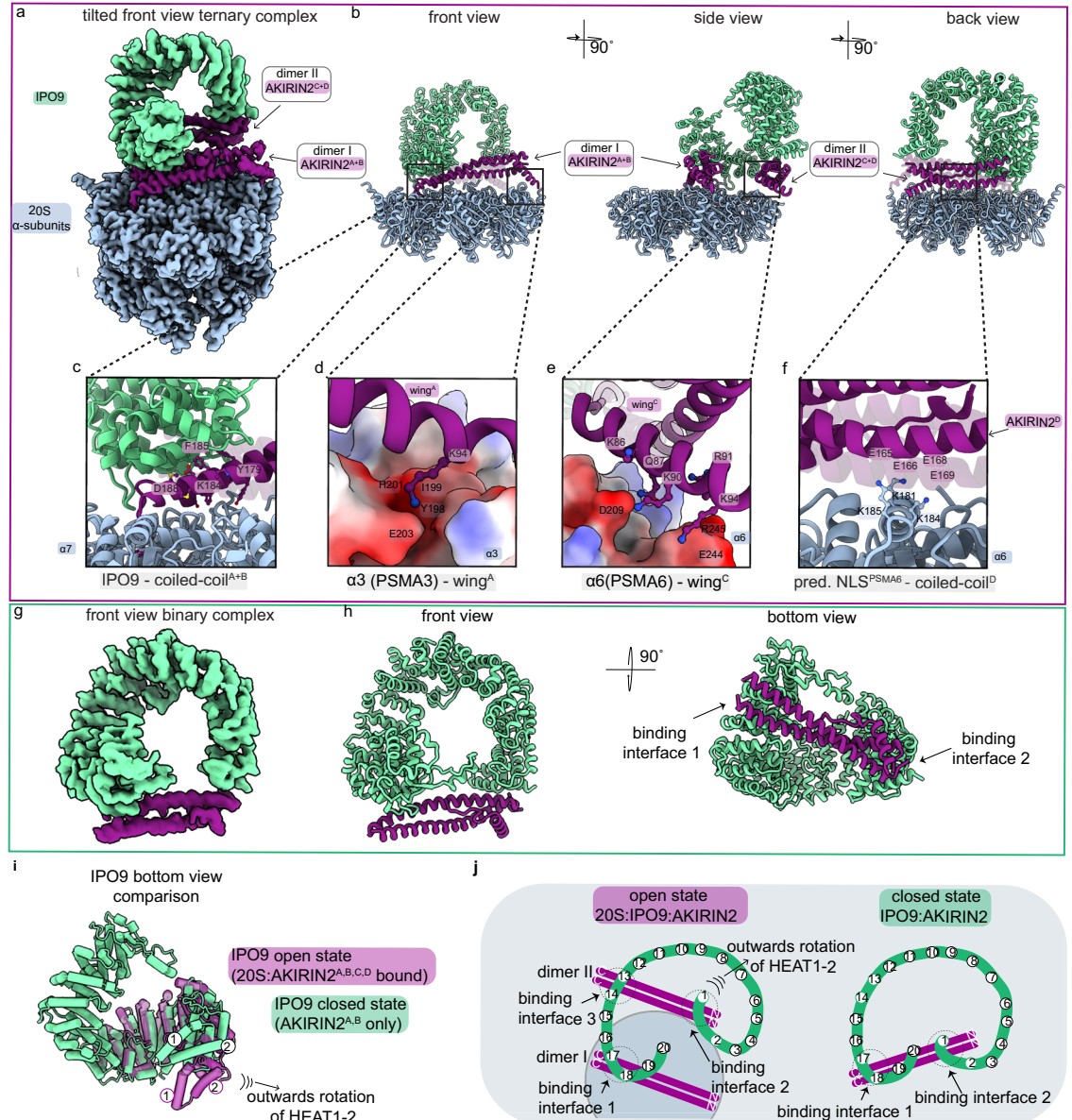

**Fig. 3 | Structure of the 20S:AKIRIN2:IPO9 ternary complex reveals clustering of AKIRIN2 on the proteasome.** Map (**a**) and model (**b**) of the ternary complex 20S (blue), AKIRIN2 (dark pink) and IPO9 (green) in front side and back view. **c** Close-up of binding interface of IPO9 with AKIRIN2 dimer I. **d** Close-up of the wing helix residue K94 of AKIRIN2 protomer A binding into a negatively charged binding pocket of proteasome subunit α3 (PSMA3) visualized with its electrostatic surface. **e** Close-up of binding interface with AKIRIN2 dimer II with IPO9 as well as α6 (PSMA6) visualized with its electrostatic surface. **f** Close-up of four Glutamic acid residues of AKIRIN2 protomer D in the coiled-coil domain which are in close proximity to three lysine residues of α6 (PSMA6). Map (**g**) and model (**h**) of the binary IPO9:AKIRIN2 complex in front and bottom view showing two individual binding interfaces. **i** Overlay of IPO9 in the open (20S:AKIRIN2 bound, dark pink) and closed (AKIRIN2 bound, green) state, visualizing the rotation of HEAT1-2. **j** Schematic representation of the IPO9:AKIRIN2 interaction in complex with the proteasome, showing dimer I interacts with its C-terminal end of the coiled-coil with HEAT H17-18, and dimer II interacts with its N-terminal end of the coiled-coil with HEAT H1 and with its C-terminal end of the coiled-coil with HEAT H13-14 (left), schematic representation of the IPO9:AKIRIN2 interaction, showing two binding interfaces between the C-terminal end of the coiled-coil of the AKIRIN2 dimer and HEAT17-18 of IPO9, and it's N-terminal end of the coiled-coil with HEAT H1 (right).

complex at 3.8 Å resolution, revealing a strikingly different binding mode compared to the ternary complex (Fig. 3g, h and Supplementary Fig. 6a-d). In the binary complex, a single AKIRIN2 dimer bridges the N- and C-terminal domains of IPO9 through two binding sites (Fig. 3h). This interaction compacts IPO9's solenoid structure, rotating HEAT repeats H1-2 inwards to a closed state (Fig. 3i, j, Supplementary Movie 2). The transition to the ternary complex involves major

conformational changes: HEAT repeats H17-H18 maintain an association with AKIRIN2 dimer I, while H1-2 open to accommodate dimer II, which also contacts HEAT repeats H13-14 (Fig. 3j). These rearrangements, stabilized by association with the proteasome, reveal at least three distinct AKIRIN2 binding sites on IPO9. The intrinsic flexibility of both proteins likely facilitates this binding plasticity, enabling the transition from one to two AKIRIN2 dimers in the ternary complex.

Functionally, this reconfiguration significantly increases the number of NLSs exposed on the proteasomal cargo, potentially enhancing import efficiency. This structural plasticity illuminates a dynamic assembly mechanism for large nuclear import complexes, balancing adaptable cargo recognition with efficient nuclear pore translocation.

## AKIRIN2 recruits multiple importins for efficient proteasome import

Our SMScreens confirmed the critical role of AKIRIN2's bipartite cNLS in proteasomal nuclear localisation. In the canonical import pathway, cNLSs are recognised by importin-α (KPNA) proteins which form a heterodimer with importin-β1 (KPNB1)[2]. This aligns with our previous co-immunoprecipitation studies[12] that identified interactions of AKIRIN2 with KPNA2/3/4 and KPNB1, suggesting their involvement in this pathway. To test this hypothesis, we examined whether KPNA2/KPNB1 heterodimers recognize AKIRIN2's cNLS and facilitate proteasome recruitment. In vitro reconstitution using sucrose density gradients demonstrated that AKIRIN2 forms a complex with KPNA2/KPNB1 and mediates their association with the proteasome (Fig. 4a). This interaction depends on AKIRIN2's cNLS, as an AKIRIN2ΔNLS mutant failed to recruit KPNA2/KPNB1 despite retaining proteasome binding (Fig. 4a, Supplementary Fig. 7a,b). These results indicate that KPNA2/KPNB1 and IPO9 engage distinct binding sites on AKIRIN2, allowing non-competitive recruitment.

We next assessed the sufficiency of these components for proteasome nuclear import using a minimal in vitro system (Fig. 4b). Proteasomes entered the nucleus only when both AKIRIN2 and importins (IPO9 and/or KPNA2/KPNB1) were present, with import efficiency increasing with AKIRIN2 concentration and time-dependent manner (Fig. 4c, d, Supplementary Fig. 8a-c). Notably, either IPO9 or KPNA2/KPNB1 was sufficient for import in the presence of AKIRIN2, underscoring AKIRIN2's pivotal role.

The transient nature of import complexes, combined with AKIRIN2's tendency to form aggregates and adhere to surfaces, makes the exact determination of stoichiometry very challenging. To approximate the increase in molecular mass associated with import complex formation, we performed size exclusion chromatography and mass photometry. These analyses revealed a stepwise increase in molecular weight following the pattern: 20S < 20S:AKIRIN2 < 20S:AKIRIN2:IPO9 < 20S:AKIRIN2:KPNA2:KPNB1 < 20S:AKIRIN2:IPO9:KPNA2: KPNB1 (Fig. 4e, Supplementary Fig. 9a-e). When KPNA2 and KPNB1 were present, mass photometry (Supplementary Fig. 9a-e) revealed a very broad spectrum of species with varying molecular weights. This observation indicates the formation of heterogeneous complexes, which likely reflects the dynamic and variable stoichiometries characteristic of these import assemblies.

While structural characterisation of the entire import complex proved challenging, likely due to the flexible nature of AKIRIN2 in its cNLS region, we generated a hybrid model combining our cryo-EM structure of the ternary complex with AlphaFold2 predictions of AKIRIN2:KPNA2:KPNB1 interactions (Fig. 4f, Supplementary Fig. 7b). This model suggests that each of the four AKIRIN2 protomers bound to IPO9 and the proteasome could independently associate with a KPNA2/KPNB1 heterodimer without steric hindrance (Fig. 4f). The flexibility of AKIRIN2's unresolved N-terminal region potentially allows importins to be positioned up to 18 nm from the proteasome surface, enabling simultaneous accommodation of at least five importin-β molecules (one IPO9 and four KPNA2/KPNB1 heterodimers) on one side of the 20S proteasome. Together, these findings reveal AKIRIN2 as a crucial adaptor that oligomerizes on the proteasome, with its extended, flexible structure providing the spatial arrangement necessary to engage multiple importin-β family members. This mechanism explains how proteasomes, incapable of directly binding importins, can efficiently traverse the nuclear pore complex.

## Disassembly dynamics of the proteasome import complex

Efficient nuclear enrichment of imported proteins requires a disassembly mechanism upon nuclear entry[17], which is commonly mediated by RanGTP binding to importin-β family proteins[1]. To investigate whether the proteasome import complex follows this principle, we incorporated a dominant GTP-bound Ran mutant, Ran(1-180)Q69L[18], into our reconstitution experiments. The addition of RanGTP destabilized importin-proteasome complexes, resulting in a complete release of KPNA2/KPNB1. However, IPO9 and AKIRIN2 were only partially released from the 20S proteasome (Fig. 5a, Supplementary Fig. 3b), reminiscent of previously observed effects of RanGTP on the H2A/H2B:IPO9 complex[15].

The incomplete dissociation of AKIRIN2 and IPO9 in vitro prompted us to investigate the release mechanism within the nucleus. Our previous work indicated that AKIRIN2 itself might be targeted for degradation by nuclear proteasomes[12], potentially providing an alternative pathway for import factor removal. To test this hypothesis, we treated RKO cells with either a proteasome inhibitor (MG132) or an E1 ubiquitin-activating enzyme inhibitor (TAK-243). MG132 treatment increased both AKIRIN2 and MYC levels compared to the DMSO control (Fig. 5b). E1 inhibition decreased overall ubiquitination and increased MYC levels but, did not prevent AKIRIN2 degradation, indicating that AKIRIN2 undergoes ubiquitin-independent proteasomal degradation, potentially facilitating the final release of proteasomes from their import machinery. To test whether proteasomes can directly degrade AKIRIN2, we performed in vitro degradation assays with purified 20S and 26S proteasomes. 20S proteasomes generated characteristic GFP-AKIRIN2 degradation fragments, reflecting degradation of AKIRIN2 while the GFP tag remained stable, whereas 26S proteasomes did not produce these products, consistent with ubiquitin-independent degradation of AKIRIN2 (Supplementary Fig. 10a,b). We further analyzed AKIRIN2 levels across the cell cycle to determine whether its abundance correlates with the timing of proteasome nuclear import. Our data revealed that AKIRIN2 expression peaks during mitosis and then clearly declines in G1 (Fig. 5c). Together, our findings suggest a two-step disassembly process for the proteasome import complex: an initial RanGTP-mediated release of canonical importins, followed by ubiquitin-independent degradation of AKIRIN2 (Fig. 5d). This mechanism ensures complete liberation of imported proteasomes in the nucleus, allowing them to perform their critical functions in protein homeostasis and transcriptional regulation.

## Discussion

Our study characterised the largely unstructured and multivalent import adaptor AKIRIN2. We reveal how AKIRIN2 recruits multiple importins to facilitate proteasome nuclear entry. Our structural and functional data elucidate a mechanism where AKIRIN2's conformational flexibility allows it to engage both the proteasome and importins, forming a transport-competent complex. AKIRIN2 binds to multiple sites on the proteasome and IPO9, leading to its oligomerization on the complex. In this arrangement, each N-terminal NLS of every AKIRIN2 protomer is accessible for recognition by the canonical import machinery. The resulting multimeric complex mobilizes numerous importins, enabling the large proteasome to traverse the nuclear pore complex. Upon nuclear entry, RanGTP dissociates most importins, while AKIRIN2 undergoes ubiquitin-independent proteasomal degradation, releasing the proteasome for its nuclear functions. It remains unclear what triggers AKIRIN2 degradation in the nucleus, and how AKIRIN2 degradation is prevented pre-import. It is possible that the binding to importins stabilises AKIRIN2 and after release degradation signals are exposed. Whether additional components similar to the recently discovered protein midnolin[19] are required to target nuclear AKIRIN2 for nuclear degradation remains to be investigated.

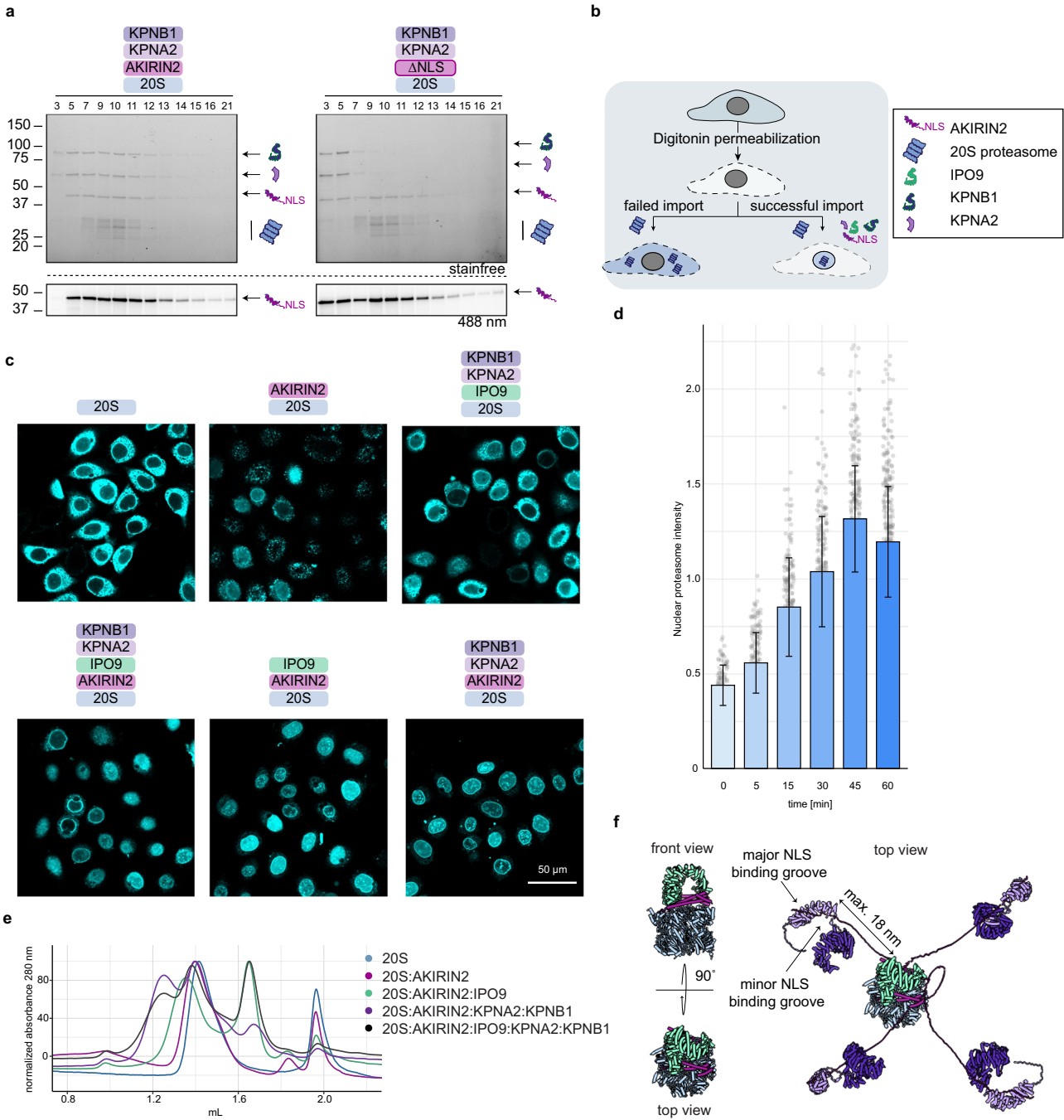

**Fig. 4 | AKIRIN2 mediates formation of the canonical nuclear import machinery. a** SDS-PAGE Biorad TGX stainfree gel of fractionated 10–30% sucrose density gradients comparing proteasome co-sedimentation with KPNA2/KPNB1. Left panel shows complex formation in the presence of wild-type AKIRIN2, while right panel demonstrates abolished complex formation with the AKIRIN2ΔNLS mutant. The lower panels display the corresponding GFP signal of the GFP-AKIRIN2 fusion protein in the same gradient fractions. **b** Schematic of the nuclear import assay procedure. **c** Representative confocal image of nuclear import of A647-20S labelled proteasome in digitonin permeabilized cells in different combinations of import factors after 60 min incubation time, *n* = 111-206 cells. **d** Quantified nuclear signal of

A647-labelled proteasomes in a time-dependent manner using 7.5 μM AKIRIN2, 1 μM IPO9, 1 μM KPNA2, 1 μM KPNB1, image brightness had to be adjusted differently for images 1 and 3. **e** Size exclusion profiles of import complexes. **f** Front and top view (left) from the proteasome of the ternary complex showing IPO9 (green) bound on top, and AKIRIN2 protomers (dark pink). Right panel presents a hybrid structural model integrating the ternary complex with AlphaFold2 predictions of full-length AKIRIN2 bound to KPNA2/KPNB1 heterodimers (purple). This model illustrates how multiple AKIRIN2 protomers can simultaneously engage with KPNA2/KPNB1 heterodimers in complex with the proteasome and IPO9 without steric hindrance.

This mechanism likely extends beyond proteasomes to other large protein complexes. Our findings, along with observations from viral capsid import studies, suggest that large cargos generally require multiple importins for nuclear entry[3,4,20]. While viral capsids like HBV achieve multi-importin recruitment through repetitive NLS display[20],

our study reveals how endogenous complexes like proteasomes use adaptors like AKIRIN2 to similar effect.

Importantly, recent studies have identified proteins with analogous functions to AKIRIN2 for other large complexes. These include Sts1/Cut8 for yeast proteasomes[9,10], HAPSTR1 for the giant E3 ligase

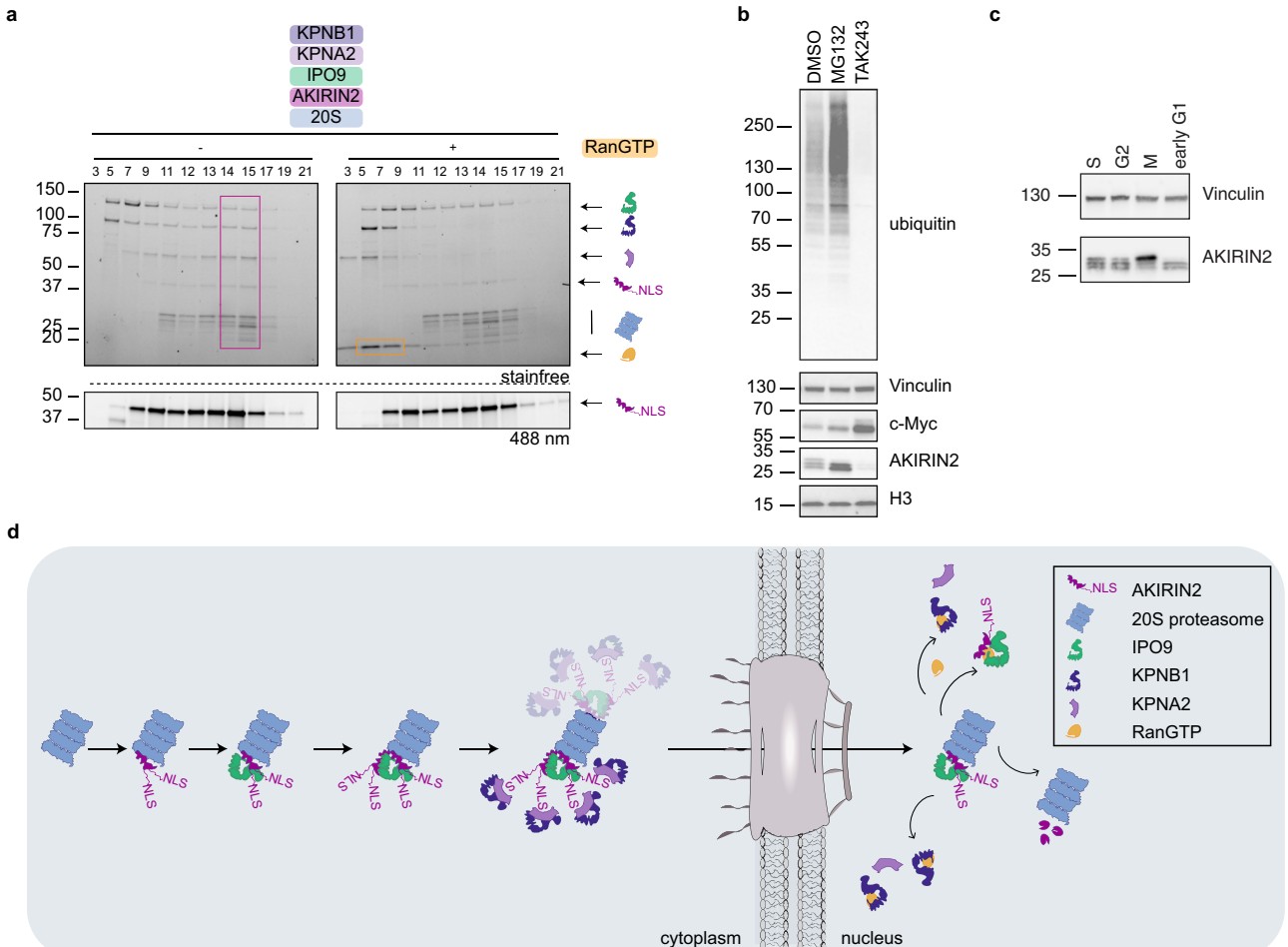

**Fig. 5 | Dissociation of nuclear proteasome from the import machinery and ubiquitin-independent AKIRIN2 degradation. a** SDS-PAGE Biorad TGX stainfree gel of fractionated 10–30% sucrose density gradient showing RanGTP addition disassembles complexes with KPNA2/KPNB1. The lower panels display the corresponding GFP signal of the GFP-AKIRIN2 fusion protein in the same gradient fractions. **b** Western blot of whole cell lysate of RKO cells treated with either proteasome inhibitor MG132, E1-inhibitor TAK243 or DMSO as control for 4 h against ubiquitin, AKIRIN2, MYC, vinculin, and H3. **c** Western blot of whole cell lysate of RKO cells, which were synchronized with double-thymidine or STLC and harvested after release in S, G2, mitosis (M), or early G1, probed against AKIRIN2 and vinculin. **d** Schematic representation of suggested nuclear import pathway of the proteasome. Showing assembly and disassembly.

HUWE1[21], VCF1/2 for the proteasomal activator p97[22], and Fam172A for AGO2[23]. While these adaptor proteins lack sequence homology, they share a common architecture: a cargo-binding region, a long flexible linker, and an NLS for importin recruitment. This conserved structure suggests a universal mechanism for nuclear import of large complexes. These nuclear import adaptors serve three critical functions. First, they eliminate the need for cargos to display multiple NLSs, which could interfere with protein function. Second, they allow cells to modulate the localisation of large complexes without altering their expression levels, enabling rapid responses to cellular stress and other stimuli[24–26]. Lastly, the oligomerization of nuclear import adaptors on cargo proteins may be a common strategy to present sufficient NLSs for efficient import, ensuring rapid nuclear translocation of essential complexes. This strategy allows cells to modulate the nuclear localisation of large complexes without altering their structure or compromising their function. It remains to be determined whether this proposed common architecture is essential for a nuclear import adaptor function and whether the constituent building blocks are interchangeable among different adaptors.

While this study was focusing on the 20S proteasome, the nuclear transport and assembly of different proteasome forms remain incompletely understood, representing an important area for future investigation. Our previous study[12] demonstrated that 19S subunit nuclear import also depends on AKIRIN2, as AKIRIN2 knockout resulted in nuclear exclusion of tagged 19S subunits that were confirmed to be fully incorporated into functional 19S/26S complexes rather than orphaned. However, it remains unclear whether 19S subunits are imported independently or piggyback with 20S proteasomes. Studies in *Xenopus* suggest that 20S import occurs independently from most 19S components[11], while yeast nuclear import adaptors like Sts1/Cut8 interact directly with 19S components[9,10], suggesting species-specific differences. Our quantitative analysis of nuclear versus cytoplasmic distribution shows that 20S proteasomes are more than twice as abundant in the nucleus, while 19S subunits show only 30% nuclear enrichment[12]. This differential distribution suggests that nuclear assembly and regulation of different proteasome forms may involve distinct mechanisms. Other proteasome activators like PA200 and PA28γ contain their own NLS and are exclusively nuclear suggesting that they enter independently.

Our discovery of AKIRIN2's role in proteasome import unveils a previously unknown mechanism for nuclear import of large protein complexes. This finding has broad implications for understanding how cells rapidly adjust their nuclear proteome in response to various stimuli. Moreover, it opens further avenues for exploring the regulation

of nuclear protein homeostasis and potentially developing therapeutic strategies targeting these processes in diseases involving aberrant protein degradation or nuclear function.

## Methods

### Cell culture
Human RKO (sex unspecified; American Type Culture Collection (ATCC) cat. no. CRL-2577, RRID:CVCL_0504) were cultured in RPMI 1640 (Gibco) supplemented with 10% FBS (Sigma-Aldrich), L-glutamine (4 mM, Gibco), sodium pyruvate (1 mM, Sigma-Aldrich) and penicillin/ streptomycin (100 U/mL, Sigma-Aldrich). Lenti-X lentiviral packaging cells (female, Clontech, cat. no. 632180) and HeLa cells (female; ATCC cat. no. CCL-2, RRID:CVCL_0030) were cultured in high-glucose Dulbecco's modified Eagle's medium (DMEM; in-house media kitchen) supplemented with 10% FBS, L-glutamine (2 mM, Gibco), sodium pyruvate (1 mM, Sigma-Aldrich) and penicillin/streptomycin (100 U/mL/, Sigma-Aldrich). All cell lines were maintained at 37 °C with 5% $CO_2$, routinely tested for mycoplasma contamination and authenticated by short tandem repeat analysis.

### Lentivirus production and infections
Semiconfluent Lenti-X cells (Clontech, 632180) were co-transfected with lentiviral plasmids, pCMVR8.74 helper (Addgene plasmid no. 22036) and pCMV-Eco (Cell Biolabs) envelope plasmids using polyethylenimine (PEI) transfection (MW 25,000, Polysciences) as previously described[27]. The following day, media was replaced in the morning with target cell media and viral supernatant was harvested 48 and 72 h post transfection. Virus containing supernatant was clarified by centrifugation. Target cells were transduced by seeding them into a new 6-well tissue culture dish and a few hours later addition of 0.5–1.5 mL virus supernatant.

### Generation of a clonal iCas9 20S reporter cell line
RKO:iCas9-BFP cells[12] were lentiviral transduced with a lenti-pRRL.SFFV plasmid expressing human PSMB4 fused to StayGold (pcDNA3/er-(n2)oxStayGold(c4)v2.0 from Atsushi Miyawaki, RRID:Addgene_186296, construct was modified removing the ER-localisation signal) and a TwinStrep tag. Cells were then transiently transfected with lentiCRISPRv2-PSMB4_16-iRFP670 using transduction reagent Lipofectamine 3000 and treated with 0.5 μg/mL of doxycycline (dox) (Sigma-Aldrich) to induce the knockout of endogenous PSMB4. Double-positive iRFP670/StayGold cells were sorted by flow cytometry and further transduced with a lenti-SFFV-mCherry-MYC-IRES-BFP construct. Final reporter cells were isolated as single cells by flow cytometry, selecting for StayGold/BFP positives and iRFP negatives. The knockout of endogenous PSMB4 and the insertion of SFFV-mCherry-MYC-IRES-BFP was verified by PCR from genomic DNA and Sanger sequencing. The resulting clone was functionally tested for Cas9 induction/tightness by perturbation of essential genes +/- dox, PSMB4-StayGold localisation by microscopy and mCherry-MYC induction by treatment with MG-132.

### Generation of individual knockout or ectopic expression cell lines
The newly generated 20S reporter cell line was lentiviral transduced with either a dual-sgRNA (hU6-sgRNA-mU6-sgRNA-EF1αs-hCD2-Puro) targeting AKIRIN2-, KPNA2-, IPO9- or AAVS1, a single sgRNA (U6-sgRNA-mPGK-hCD2) targeting KPNB1 or PSMA3, or with a lentiviral plasmid containing an AKIRIN2 variant (pRRL-SFFV-AKIRIN2mutant-IRES-Thy1.1-Neo). Following transduction, cells were selected with antibiotics or FACS sorted by antibody staining of the co-expressed surface-markers (hCD2 and Thy1.1). For the knockout cells, Cas9 induction was induced by 0.2–0.5 μg/mL dox for 2–3 days. Cells were analysed by confocal microscopy. Therefore, they were seeded on ibidi 8-well μ-slides the day before and imaged on a LSM 880

Axio Observer (inverted) with Airyscan /Airyscan fast with incubation chamber.

### Construction of the human AKIRIN2 cDNA mutagenesis library
The construction of the AKIRIN2 cDNA mutagenesis library was organized into four sub-libraries, each covering approximately one-fourth of the AKIRIN2 coding sequence (50 amino acid residues). These sub-libraries together encompass every possible single amino acid substitution, including changes to any of the other 19 amino acids, deletions of individual residues, self-replacements (negative control), and mutations to STOP codons (positive control). A codon-optimized AKIRIN2 cDNA, modified to harbour point mutations at sgRNA cut sites, was engineered and tested for its ability to rescue endogenous AKIRIN2 knockout.

For each sub-library, a different recipient vector was constructed, incorporating a specific landing site for oligonucleotide DNA inserts at the designated positions. Oligonucleotide pools encoding approximately 50 amino acids, flanked by Esp3I sites and specific primer binding sites, were synthesized by Twist Bioscience (-1200 oligos per pool). To ensure robust detection of mutants by next-generation sequencing (NGS), codons were altered to differ by at least two nucleotides from the template sequence wherever possible.

The oligo pools were PCR amplified separately and cloned into the corresponding recipient vectors (pRRL-SFFV-PBSa-AKIRIN2mutant[Esp3I-filler-Esp3I]-PBSb-IRES-Thy1.1-Neo) using Esp3I Golden Gate assemblies with 48 cut-ligation cycles. The assembled constructs were purified and introduced into MegaX DH10B electrocompetent cells (ThermoFisher) by electroporation, followed by overnight culture and plasmid isolation after overnight culture. All four sub-libraries were cloned to achieve a representation exceeding 10,000 bacterial colonies per construct, verified by dilution series.

### Saturation mutagenesis screen (SMScreen)
AKIRIN2 cDNA sub-libraries were transduced at a single-copy level into the 20S reporter cell line (RKO:iBFPCas9:PSMB4-StayGold:ΔPSMB4:mCherry-MYC-IRES-BFP) and selected using 500 μg/mL Neomycin. After confirming the selection of Thy1.1 positive cells, they were further transduced with an optimized dual-sgRNA construct targeting endogenous AKIRIN2 (dual-hU6-sgRNA1-mU6-sgRNA2s-hCD2-P2A-PuroR) and subsequently selected with 2 μg/mL Puromycin. The efficiency of each transduction step was assessed via FACS, analysing surface marker expression. Cells were cultured for up to four days and large batches, exceeding 10,000-fold library representation, were cryopreserved. To account for cell cycle variations, editing of endogenous AKIRIN2 was induced in a staggered manner (96 h, 72 h, and 48 h before the experiment) by adding 0.5 μg/mL dox. Each induction time point and sub-library were maintained at representations above 10,000-fold.

Cell sorting involved two separate flow cytometry approaches: First, -2–3% of cells displaying the highest mCherry levels were sorted by FACS. Second, cells were stained at a 1:25 ratio with DRAQ5 nuclear stain, and roughly 5% of cells at both the lowest and highest DRAQ5:StayGold(-PSMB4) correlations were sorted using an image-enabled cell sorter (ICS) from BD Biosciences. Sorted cells were centrifuged at 600 x g for 5 min, and the cell pellets were immediately flash-frozen and stored at −80 °C until DNA isolation. Additional cells were collected 12 days post-dox induction (dropout). Genomic DNA (gDNA) from dropout and t0 (uninduced) samples was isolated via phenol extraction, while gDNA from sorted samples was purified using NucleoSpin Blood Mini kits. All samples were amplified in nested PCR reactions using barcoded primers. The initial PCR amplified the region of interest within the AKIRIN2 cDNA over 27 cycles using one primer flanking the cDNA and one intra-cDNA primer. DNA from all samples was then purified, quantified using Qubit Fluoro-DNA HS, and pooled at appropriate concentrations for a secondary PCR of 6 cycles to

append sequencing adapters. Samples were sequenced on a NextSeq2000 P1 flow cell in paired-end 300 read mode.

## Saturation mutagenesis data analysis

The oligonucleotide insert libraries were indexed for Bowtie (v1.3.1) analysis (https://doi.org/10.1186/gb-2009-10-3-r25). Initially, random 4-mer nucleotides were trimmed using Cutadapt (v4.8) (https://doi.org/10.14806/ej.17.1.200) prior to the demultiplexing of reads by 8-mer sample barcodes, without allowing for mismatches. Following demultiplexing, additional trimming of barcodes and spacers was performed using Cutadapt, resulting in reads that exclusively covered the region of interest. These trimmed reads were then perfectly aligned using Bowtie with the parameters (--best --tryhard -v 0) and quantified using featureCounts (v1.6.1) (PMID: 24227677).

The raw counts were median-normalized, and fold changes between the sorted samples and reference points (either t0 or drop-out) were calculated. The effects were scaled from 1 (representing strong enrichment, median from STOP controls) to 0 (indicating neutral effects, median from self-replacement controls), facilitating a more accurate cross-comparison of sub-libraries. To summarize the data effectively, a single score value was derived for each mutation, integrating individual comparison contrasts. The base for this integration was the scaled effects from the mCherry-MYC screen, which were then adjusted by a fraction derived from the PSMB4($\alpha$4) localisation screen results. Detailed analysis scripts are available upon request.

## Validation of hits from the saturation mutagenesis screen

To validate individual hits from the saturation mutagenesis screen, cell lines expressing an AKIRIN2 variant were created using the methods described above– with DNA synthesized by Twist Biosciences. These cells were further transduced with the optimized dual-sgRNA targeting endogenous AKIRIN2 and selected using 500 µg/mL Neomycin and 2 µg/mL Puromycin. Transfection efficiency was evaluated by FACS, staining for the surface markers hCD2 and Thy1.1. Cells were analysed by confocal microscopy as described above. Additionally, cell lines with ectopic expression were assessed by FACS (Novocyte Penteon, Agilent), gating for mCherry-positive cells. Lysates from these cells were harvested after 3 days of dox induction and analysed by Western blot, probing AKIRIN2 protein levels and using vinculin as a loading control.

## Competitive profiling assay

To assess cell viability, cells were lentiviral transfected with sgRNA expression vector with 20–70% infection efficiency. sgRNA positive cells were sorted by flow cytometry staining the co-expressing surface or fluorescent marker. Cells were mixed with not transfected cells to a 70/30 (transfected/not-transfected) ratio. Knockout of the cells was induced for 1-9 days, cells were stained with an APC-conjugated anti-CD2 or anti-Thy1.1 antibody and the ratio of transfected cells was measured for each time point by flow cytometry.

## Cell synchronization to probe for AKIRIN2 levels over cell cycle

To synchronize cells at different cell cycle stages, two strategies were employed. For S- and G2-phase, RKO cells were synchronized using a double thymidine block. Cells were treated with 2 mM thymidine (GE healthcare) for 16 h, washed twice with pre-warmed PBS, and released into pre-warmed medium for 8 h. This was followed by a second 16 h treatment with thymidine and subsequent release. Cells in S-phase were harvested 5 h after release, and cells in G2-phase 9 h after release. For M-, early G1-, and late G1-phase, RKO cells were synchronized with 10 µM S-Trityl-L-cysteine (STLC; Merck) for 16 h. Mitotic cells were enriched by mitotic shake-off and washed four times with pre-warmed PBS. Mitotic cells were harvested 1 h after release, early G1 cells 3 h after release, and late G1 cells 5 h after release. For each condition, 1-2

million cells were collected, washed with PBS, and lysed in RIPA buffer supplemented with 1× cOmplete and 1 mM PMSF. Lysates were incubated on ice for 10 min, sonicated twice for 10 s, and centrifuged at 15,000 x g for 10 min to pellet cell debris. The supernatant was transferred to a new tube, and protein concentration was determined by Bradford assay. Twenty micrograms of total protein were resolved on a 4-20% TGX Stain-Free™ Protein Gel (Bio-Rad) at 200 V for 45 min, transferred to a nitrocellulose membrane, and probed for AKIRIN2 protein levels, with vinculin used as a loading control.

## 20S purification

For expression and purification of 20S a slightly adapted protocol from[28] was used. The initial plasmids pACEBac1 containing α-subunits, pIDS vector containing β-subunits and pACEBac1 containing assembly chaperones PAC1-PAC2, PAC3-PAC4 and POMP were kindly provided by Prof. Fonseca. α-subunits from pACEBac1 vector and β-subunits from pIDS vector were combined on one pGBdest vector by Protech facility using golden gate. pGBdest vector containing all subunits and pACEBac1 vector with all chaperones were transformed into DH10EmBacY chemically competent cells. Positive clones were selected using blue/white screening. The bacmids were purified using isopropanol precipitation and the recombination of the gene of interests were verified by PCR. The bacmids were transfected into Sf9 cells using polyethyleneimine (PEI) for the generation of the V0 stock, using YFP fluorescence as a measure of transfection efficiency. Following virus amplification in Sf9 suspension insect cell culture, constructs were expressed in Trichoplusia ni High-Five insect cells (Thermo Fisher) infected at a density of $1 \times 10^6$ cells/mL using 1:1 ratio of the 20S and the assembly chaperones viruses. Cells were grown at 100 rpm at 27 °C shaking using ESF921 serum-free growth medium (Expression Systems). Cells were harvested 3 days after transfection by centrifugation at 2000 x g for 20 min, and pellets were flash-frozen in liquid nitrogen and stored at −80 °C.

The cell pellet was resuspended in 3x (w/v) of Buffer L1 (50 mM HEPES/NaOH pH 7.5, 500 mM NaCl, 1 mM EDTA, 0.5 mM TCEP) supplemented with Benzonase (IMP Molecular Biology Service). Cells were lysed using a glass Dounce homogenizer and subsequently centrifuged at 40,000 x g for 1 h. The supernatant was applied on a pre-equilibrated StrepTrap HP column (Cytiva) equilibrated in Buffer L1. The column was washed with 10 column volumes (CV) of buffer P1 (50 mM HEPES/NaOH pH7.5, 150 mM NaCl, 1 mM TCEP) and eluted with Buffer P1 supplemented with 5 mM d-Desthiobiotin. The eluate was collected in fractions and desired fractions were combined, concentrated with a 100 kDa cut-off concentrator (Cytiva), and loaded on a Superose 6 16/70 column (Cytiva) equilibrated in buffer P1. For long-term storage, the desired fractions were combined, concentrated and flash-frozen in liquid nitrogen. Typical yield 5 mg from 1 L of Hi5 culture.

## His-GFP-AKIRIN2/ His-GFP-AKIRIN2ΔNLS purification

His-GFP-AKIRIN2 was purified as previously described[29]. The mutant His-GFP-AKIRIN2ΔNLS (deletion of residues 8-28) was purified in the same way. Briefly, His-GFP-AKIRIN2 was expressed in BL21(DE3). Culture was inoculated into 6 L of autoinduction media supplemented with 50 µg/mL kanamycin and incubated at 37 °C for 5 h followed by 18 °C for 18 h. Cells were harvested at 2000 x g for 20 min at 4 °C. The cell pellet was resuspended in buffer L2 (50 mM Tris-HCl pH 8, 500 mM NaCl, 0.5 mM TCEP) supplemented with 1x cOmplete protease inhibitor and Benzonase. The cell suspension was homogenized using French Press at 1.5 kbar and subsequently centrifuged at 40,000 x g for 1 h at 4 °C. The supernatant was applied on a 5 mL HisTrap HP column (Cytiva) pre-equilibrated with buffer L2. Column was washed with 10 CV of 5% of buffer P2 (50 mM Tris-HCl pH8, 500 mM NaCl, 0.5 mM TCEP, 500 mM Imidazole pH8) and eluted with 50% of buffer P2. Fractions were pooled, concentrated with a 30 kDa cut-off

concentrator (Cytiva) and loaded on a Superdex S200 16/60 column (Cytiva) equilibrated in buffer P3 (25 mM HEPES pH 7.5, 100 mM NaCl, 0.5 mM TCEP). Desired fractions were pooled, concentrated and flash-frozen in liquid nitrogen for long-term storage.

## IPO9 purification

Strep-IPO9 (TS_pGC-01;02 from VBSF) was expressed in Arctic Express cells in autoinduction media supplemented with 20 μg/mL Gentamicin and culture was incubated for 37 °C for 5.5 h, followed by 16 °C for 24 h. Cells were pelleted 2000 x g, 20 min. Pellet was flash frozen until further processing. Cells were resuspended in 250 mL buffer L3 (50 mM Tris 7.5, 300 mM NaCl, 0.5 mM TCEP) supplemented with 1x cOmplete protease inhibitor and Benzonase. The cell suspension was homogenized using French Press at 1.5 kbar and subsequently centrifuged at 40,000 x g for 1 h. The supernatant was applied to a Strep Trap HP 5 mL which was before equilibrated with 5 CV buffer L3. Column was washed with 5 CV buffer L3 and eluted with 5 CV 3 mL/min buffer P4 (buffer L3 + 5 mM d-Desthiobiotin). Desired fractions were pooled and applied on a ResQ (6 mL) column which was equilibrated with buffer P5 (50 mM Tris 7.5, 50 mM NaCl, 0.5 mM TCEP). After sample application column was washed with 5 CV buffer P5 and then eluted with 20 CV up to 37% gradient increasing concentration of buffer P6 (50 mM Tris pH7.5, 2 M NaCl, 0.5 mM TCEP). Desired fractions were pooled and loaded on a Gel Filtration (Hiload 16/600 Superdex 200) column which was equilibrated with buffer P7 (40 mM Tris 7.5, 50 mM NaCl, 0.5 mM TCEP). Sample was eluted with 1 CV of buffer P7. Desired fractions were pooled and concentrated in 30 kDa cutoff 15 mL tubes.

## KPNB1 purification

Human KPNB1 was cloned into pQE-60 and expressed in E. coli SG13009 cells grown in 2YT to an OD600 = 0.6–0.8 at 37 °C and afterwards induced with 0.5 mM IPTG (additional 30 mM K2HPO4 added for buffering). Cultures were incubated at 18 °C overnight and harvested by centrifugation 5749 x g for 30 min at 4 °C. The pellet was dissolved in 30 mL 1x PBS (2.7 mM KCl, 140 mM NaCl, 8.6 mM $Na_2HPO_4$, 1.5 mM $KH_2PO_4$ pH 7.4). Cells were centrifuged at 4147 x g for 15 min at 4 °C and the pellet was flash frozen in liquid nitrogen and stored at −20 °C until further use. For purification the pellet was dissolved in buffer L4 (50 mM Tris/HCl pH 7.5, 150 mM NaCl, 10 mM KCl, 1 mM $MgCl_2$, 10 mM Imidazole) and lysed with a high-pressure homogenizer (Microfluidics) with 15,000 psi, at 4 °C. Lysate was centrifuged at 30,000 x g for 30 min at 4 °C. The supernatant was applied to a His-Trap column (10 mL final volume, either from Macherey Nagel or Cytiva). Column was washed with buffer L4 + 70 mM Imidazole to remove unspecific bound proteins and afterwards KPNB1 was eluted by increasing the Imidazole concentration to 250 mM. Desired fractions were analysed by SDS-PAGE Biorad TGX stainfree gel and the respective fractions pooled and concentrated to 3.5-4 mL before loading on a size exclusion chromatography (Superdex 200 26/60 or 16/60), which was equilibrated in buffer P8 (20 mM Tris/HCl pH 7.5, 150 mM NaCl, 2 mM $MgCl_2$). Eluted peak fractions were analysed by SDS-PAGE Biorad TGX stainfree gel. Desired fractions were combined and concentrated in 50 kDa cutoff concentrators to a concentration of >10 mg, flash frozen and stored at −80 °C.

## KPNA2 purification

Human KPNA2 cDNA was cloned into His vector pRSET (Invitrogen) using BamHI and XhoI restriction sites[30]. For expression of KPNA2 was transformed into the E. coli strain BL21 (DE3) LysS and the cells were grown to OD600 of 0.6 and induced with 0.5 mM IPTG for 4 h at 30 °C. Cells were harvested by centrifugation 5749 x g for 30 min at 4 °C. The pellet was dissolved in 30 mL 1x PBS (2.7 mM KCl, 140 mM NaCl, 8.6 mM $Na_2HPO_4$, 1.5 mM $KH_2PO_4$ pH 7.4). Cells were centrifuged at 4147 x g for 15 min at 4 °C and the pellet was flash frozen in liquid

nitrogen and stored at −20 °C until further use. Cells from a 3 L culture were lysed in buffer L5 (50 mM Tris pH 7.5, 500 mM NaCl) with a French Press and recombinant KPNA2 was purified twice over Ni-nitrilotriacetic acid agarose (Qiagen) under native conditions step wise eluting with buffer P9 (50 mM Tris pH 7.5, 500 mM NaCl, 20, 60 and 300 mM Imidazole). Respective elution fractions were combined and concentrated to 3.5–4 mL and loaded on a size exclusion chromatography (Superdex 200 26/60 or 16/60) which was equilibrated in buffer P8 (20 mM Tris/HCl pH 7.5, 150 mM NaCl, 2 mM $MgCl_2$). Proteins were eluted with 1 CV buffer. Desired fractions were combined and concentrated in 30 kDa cutoff concentrators to a concentration of >1 mg, flash frozen in liquid nitrogen and stored at −80 °C.

## His-GST-mCherry purification

mCherry construct was cloned into a pETM33 plasmid and was expressed in BL21(DE). Culture was inoculated into 4 L of autoinduction media supplemented with 50 μg/mL Kanamycin. Culture was growing at 37 °C for 5.5 h, followed by 16 °C for 24 h. Culture was spun down at 2000 x g for 30 min. Pellet was resuspended in 150 mL lysis buffer L3 supplemented with 1x cOmplete protease inhibitor and Benzonase. The cell suspension was homogenized using a French Press at 1.5 kbar and subsequently centrifuged at 40,000 x g for 1 h at 4 °C. Supernatant was loaded on two HisTrap columns (5 mL each) in a row which were before equilibrated with 2 CV ddH2O, and 5 CV buffer L3. Columns were washed with 20 CV wash of 5% buffer P10 (50 mM Tris 7.5, 0.5 M Imidazole, 300 mM NaCl, 0.5 mM TCEP) and afterwards eluted with 10 CV 50% buffer P10, followed by 5 CV 100% buffer P10. Desired fractions were combined and loaded on a GST Trap crude column 5 mL which was equilibrated with buffer P11 (50 mM Tris pH8, 0.3 M NaCl, 0.5 mM TCEP). Column was washed with 20 CV buffer P11 and eluted with 5 CV buffer P11 supplemented with 15 mM GSH and re-adjusted pH to 8. Desired fractions were combined and dialyzed overnight in 1 L buffer P12 (50 mM Tris 7.5, 150 mM NaCl, 0.5 mM TCEP) and concentrated in a 6 mL 10 kDa cutoff concentrator at 4000 x g for 30 min at 4 °C to a concentration of 12 mg/mL.

## Ran purification

Human $His_{10}$-tagged Ran (residues 1:180) was cloned into a (pETM14 plasmid) was expressed in Rosetta2 cells. Culture was inoculated into 4 L of autoinduction media supplemented with 50 μg/mL Kanamycin. Culture was spun down at 2000 x g for 30 min at 4 °C. Pellet was resuspended in 200 mL buffer L6 (20 mM Tris pH8.0, 0.5 M NaCl, 0.5 mM TCEP) supplemented with 1x cOmplete protease inhibitor and Benzonase. The cell suspension was homogenized using French Press at 1.5 kbar and subsequently centrifuged at 40,000 x g for 1 h at 4 °C. Supernatant was loaded on a His Trap column 5 mL which was equilibrated with buffer L6. After sample application column was washed with 5% buffer P13 (20 mM Tris pH8.0, 0.5 M Imidazole, 0.5 M NaCl, 0.5 mM TCEP) and eluted with 10 CV 50% & 5 CV 100% buffer P13. Desired fractions were combined and diluted with buffer P14 (50 mM Tris 8.0, 50 mM NaCl, 0.5 mM TCEP) to a salt concentration of about 50 mM NaCl and loaded on a ResQ column which was equilibrated with buffer P14. Elution with 20 CV up to 37% gradient increasing concentration of buffer P15 (50 mM Tris, 2 M NaCl, 0.5 mM TCEP). Sample was afterwards applied to a S200 size exclusion column which was equilibrated with buffer P16 (20 mM HEPES 7.5, 100 mM NaCl, 0.5 mM TCEP). Desired fractions were combined and concentrated with a 10 kDa cutoff concentrator to either 1 or 1.6 mg/mL.

## RanGTP purification

Human $His_{10}$-tagged RanGTP Q69L (residues 1-180) was expressed in Rosetta (DE3) pLys, 4 L TB with Kanamycin and Chloramphenicol, induced at OD600 ~ 1, grown overnight at 18 °C. Resuspended in buffer P17 (50 mM K-Phosphate pH 7.0, 500 mM NaCl, 5 mM Mg(OAc)$_2$, 1 mM EDTA, 10 mM imidazole and 2 mM DTT) supplemented with 100 μM

PMSF, 1x cOmplete protease inhibitor, 10 µg/mL DNAse I and 1 mg/mL lysozyme, and lysed by sonication (3:30, 70%, 0.5/0.5). 20 µM GTP was included in all purification buffers. Purification was done as previously described[31].

## Antibodies

The following antibodies were used for western blot analysis: AKIRIN2 (HPA064239, Atlas Antibodies, RRID:AB_2685222, 1:1,000), c-MYC (D84C12, no. 5605, Cell Signaling Technology, RRID:AB_1903938, 1:1,000), HRP-anti-Histone H3 (3H1, Cell Signaling (1:1,000)), IPO9 (PA5-25477, Invitrogen, rabbit, polyclonal antibody, RRID:AB_2542977) (1:1,000), Anti-vinculin ms (1:400) (V9131, Sigma, RRID:AB_477629).

Secondary antibodies: HRP anti-rabbit IgG (7074, Cell Signaling Technology, RRID:AB_2099233, 1:10,000), HRP anti-mouse IgG (7076, Cell Signaling Technology, RRID:AB_330924, 1:10,000).

Antibodies used for FACS analysis: APC anti-CD90.1/Thy1.1 (202526, BioLegend, RRID:AB_1595470, 1:500), APC anti-Hu/NHP hCD2 (17-0029-42, Invitrogen, RRID:AB_10805740, 1:500).

## Immunoprecipitation

For Co-immunoprecipitations assays 1 nmol bait protein which is either Strep-IPO9 or as a control uncharacterised zinc finger protein 1 (organism: leptonema illini) as a control was added each in triplicates to 6.5 mL HeLa lysate which was supplemented with 4 mM ATP, 1 mM DTT (both needed to keep proteasomes stable), 0.1 mM PMSF, 2 mg/mL BSA (to block unspecific binding). Per trial 50 µL slurry of Strep-Tactin high performance Sepharose (Cytiva 28-9356-00) was used and washed twice with work buffer (25 mM Bis Tris pH 6.5, 50 mM KCl, 5 mM MgCl$_2$, 4 mM ATP, 1 mM DTT) and to block unspecific binding beads were incubated for 30 min, rotating on RT with work buffer supplemented with 2 mg/mL BSA. Afterwards beads were added to the lysate and incubated for 2 h, 4 °C, rotating. Afterwards beads were spun down 500 x g, 5 min, 4 °C. Aliquots from the supernatant were taken for analysis and rest was discarded. Beads were washed 6x with 1 mL work buffer. Proteins were eluted two times by adding 200 µL elution buffer (work buffer + 5 mM d-Desthiobiotin), incubating for 30 min, 4 °C, rotating. Beads were centrifuged 500 x g, 1 min and supernatant was transferred. Samples were analysed on SDS-PAGE Biorad TGX stainfree gel. Remaining samples was acetone precipitated (add 4x excess of ice cold (−20 °C) Acetone to a final concentration of 80% to the samples, vortex and incubate −20 °C, overnight, pelleted for 10 min at 10,000 x g at 4 °C, dissolve pellet in 50 µL 20 mM Tris pH 7.5, 50 mM NaCl), dissolved in 20 mM Tris pH 7.5, 50 mM NaCl and submitted for mass spectrometry analysis.

## Mass spectrometry analysis

Samples were supplemented with 10 mM Dithiothreitol (DTT) and were heated to 95 °C for 10 min. Iodoacetamide (IAA) was added to a final concentration of 20 mM and the samples were incubated for 30 min at RT in the dark. The reaction was quenched by addition of DTT to 5 mM and incubated again 30 min at RT. 400 ng trypsin (Promega, Trypsin Gold) were added followed by an overnight incubation at 37 °C. The samples were acidified with trifluoroacetic acid (TFA, Pierce) to a final concentration of 1%. The nano HPLC system (UltiMate 3000 RSLC nano system) was coupled to an Orbitrap Exploris 480 mass spectrometer equipped with a Nanospray Flex ion source (all parts Thermo Fisher Scientific).

Peptides were loaded onto a trap column (PepMap Acclaim C18, 5 mm × 300 µm ID, 5 µm particles, 100 Å pore size, Thermo Fisher Scientific) at a flow rate of 25 µL/min using 0.1% TFA as mobile phase. After loading, the trap column was switched in line with the analytical column (PepMap Acclaim C18, 500 mm × 75 µm ID, 2 µm, 100 Å, Thermo Fisher Scientific). Peptides were eluted using a flow rate of 230 nL/min, starting with the mobile phases 98% A (0.1% formic acid in

water) and 2% B (80% acetonitrile, 0.1% formic acid) and linearly increasing to 35% B over the next 120 min. This was followed by a steep gradient to 95%B in 5 min, stayed there for 5 min and ramped down in 2 min to the starting conditions of 98% A and 2% B for equilibration at 30 °C. The Orbitrap Exploris 480 mass spectrometer was operated in data-dependent mode, performing a full scan (m/z range 350–1200, resolution 60,000, normalized AGC target 300%, minimum intensity set to 10,000 and compensation voltages CV of −45 V and −60 V), followed by MS/MS scans of the most abundant ions for a cycle time of 1.0 s per CV. MS/MS spectra were acquired using an isolation width of 1.0 m/z, normalized AGC target 200%, HCD collision energy of 28 and Orbitrap resolution of 45,000. Precursor ions selected for fragmentation (include charge state 2-6) were excluded for 45 s. The monoisotopic precursor selection (MIPS) mode was set to Peptide and the exclude isotopes feature was enabled.

Data processing protocol: For peptide identification, the RAW-files were loaded into Proteome Discoverer (version 2.5.0.400, Thermo Scientific). All MS/MS spectra were searched using MSAmanda v2.0.0.16129[32]. The peptide mass tolerance was set to ±10 ppm and fragment mass tolerance to ±8 ppm; the maximum number of missed cleavages was set to 2, using tryptic enzymatic specificity without proline restriction. Peptide and protein identification was performed in two steps. For an initial search the RAW-files were searched against the human reference database (20,541 sequences; 11,3957,48 residues), supplemented with common contaminants and sequences of tagged proteins of interest using Iodoacetamide derivative on cysteine as a fixed modification. The result was filtered to 1 % FDR on protein level using the Percolator algorithm[33] integrated in Proteome Discoverer. A sub-database of proteins identified in this search was generated for further processing. For the second search, the RAW-files were searched against the created sub-database using the same settings as above and considering the following additional variable modifications: oxidation on methionine, glutamine to pyro-glutamate conversion at peptide N-terminal glutamine and acetylation on protein N-terminus. The localisation of the post-translational modification sites within the peptides was performed with the tool ptmRS, based on the tool phosphoRS[34]. Identifications were filtered again to 1% FDR on protein and PSM level, additionally, an Amanda score cut-off of at least 150 was applied. Proteins were filtered to be identified by a minimum of 2 PSMs in at least 1 sample. Protein areas have been computed in IMP-apQuant[35] by summing up unique and razor peptides. Resulting protein areas were normalised using iBAQ[36] and sum normalisation was applied for normalisation between samples.

## Sucrose density gradients

Density gradient centrifugations were performed by preparing the low-density solution of 10% (w/v) sucrose and high-density solution of 30% (w/v) sucrose in reconstitution buffer (20 mM phosphate buffer pH7.5, 50 mM NaCl, 4 mM ATP, 1 mM DTT). Gradients were made by mixing the two solutions using the Gradient Master (Biocomp Systems) to create a continuous sucrose gradient. All compared gradients were performed in parallel using the same input protein concentrations (if not mentioned otherwise, 40 pmol 20S proteasomes, 400 pmol of AKIRIN2 and 200 pmol of IPO9, KPNA2, KPNB1). The sample was prepared in 200 µL reconstitution buffer and applied on top of the density gradient prepared in an open-top 4.2 mL ultracentrifuge tube and run for 16 h at 165,000 x g in an SW60Ti rotor. Gradients were manually fractionated into 200 µL fractions and analysed with stain-free and fluorescence imaging using Chemi-DocMP System (BioRad). For western blot analysis, up to 20 µL samples were usually applied to 4–20% TGX Stain-Free™ Protein Gel (BioRad), and run at 200 V for 45 min. Gel was transferred to a nitrocellulose membrane and probed for the desired proteins (e.g. AKIRIN2, IPO9).

## Proteasome labelling with A647

To label 20S proteasomes with Alexa-647, recombinantly expressed 20S proteasomes with a concentration of 3 g/L were adjusted to a pH8 by adding 1.5 M HEPES pH8.3. Alexa-647-NHS dye was added to a final concentration of 100 μM. Reaction was incubated at 4 °C overnight (18 h). Reaction was quenched with 100 mM Tris pH 7.5. labelled 20S proteasomes were separated from free remaining dye with size exclusion chromatography S6 increase 10/300 GL. Column was equilibrated with 20 mM HEPES pH7.3, 150 mM NaCl, 0.5 mM TCEP. Sample is applied over a 1 mL loop, which is emptied with 2 mL of the buffer and then eluted with 1 CV, fractionated into 500 μL fractions. Desired fractions are analysed by SDS-PAGE Biorad TGX stainfree gel and combined.

## In-vitro degradation assay 20S proteasomes vs. 26S proteasomes

Degradation reactions were carried out as 100 μL reaction in activity buffer (25 mM Tris pH 7.5, 5 mM MgCl$_2$, 5% glycerol, 4 mM ATP, 1 mM DTT, 1 mM GTP). The following concentrations were used: 25 nM 20S or 26S proteasomes, 50 nM IPO9, 250 nM RanQ69L, 0.5 mM MG132, 100 nM GFP-AKIRIN2. To reduce unspecific binding, reaction tubes were blocked for 30 min with 100 μL blocking buffer (2 mg/mL BSA in activity buffer) prior to sample preparation. All components were assembled on ice, with GFP-AKIRIN2 added last. Samples were mixed immediately, and an aliquot was taken as the 0 min time point. Reactions were incubated at 37 °C, and additional aliquots were collected at multiple time points (5–60 min), then quenched by the addition of SDS loading buffer. Samples were resolved on 4–20% gradient SDS–PAGE gels (Biorad) run at 200 V for 40 min. Gels were imaged using a Biorad ChemiDoc system with stain-free detection and GFP fluorescence channels.

## Cryo-EM sample preparation

For cryo-EM analysis, complexes were reconstituted by sucrose density gradients, desired fractions were buffer exchanged in the reconstitution buffer excluding the sucrose using Zeba spin desalting columns or 30/100 kDa cutoff concentrators (ThermoFisher). Grids were glow-discharged with BalTec SCD 005 sputter coater for 60 s at 25 mA using residual air. For the 20S:AKIRIN2:IPO9 sample R3.5/1 200 mesh Cu preloaded with 2 nm Carbon (Quantifoil) were used for plunge freezing. For the IPO9:AKIRIN2 sample open-hole R1.2/1.3 grids (Quantifoil) with 200 mesh. 4 μL of the sample was applied onto the treated side, and subsequently incubated for 2 s and front-side blotted using the Leica GP2 plunge freezer. Initial grid screening was performed on the 200 kV Glacios microscope (ThermoFisher) using the Falcon III detector (Thermo-Fisher). For the high-resolution 20S:AKIRIN2:IPO9 structure determination, data collection was performed on a 300 kV Titan Krios G2 equipped with a cold field emission gun and a K3 BioQuantum (Gatan) with a slit width of 20 eV. Images were collected at a pixel size of 1.06 Å/pix with a cumulative dose of 44.4 e/Å$^2$ in EER format using a defocus range in −1.0 to −1.9 μm in 0.2 μm increments.

For the high-resolution AKIRIN2:IPO9 structure determination data collection was performed on a 300 kV Titan Krios G4 equipped with a cold field emission gun and a post-column Selectris energy filter (ThermoFisher) with a 10 eV slit width and a Falcon 4 or Falcon 4i direct electron detector (ThermoFisher). Images were collected at a pixel size of 0.951 Å/pix with a cumulative dose of 40 e/Å$^2$ in eer format using a defocus range in -1.0 to -2.0 μm in 0.25 μm increments.

## Cryo-EM image processing

The 9,366 movies of the 20S:AKIRIN2:IPO9 datasets were patch motion corrected and patch CTF estimated in cryoSPARC (v 3.2)[37]. 2D templates were generated by manually picking ~770 particles followed by 2D classification. Selected averages were used for template picking and particles were extracted (boxsize 300 px, 1.07 Å/px). Particles were sorted by performing three rounds of 2D classification with particles separated into 6 equally sized batches. Next, ab-initio reconstruction and heterogeneous refinement were used to exclude badly resolving particles and 20S particles bound by proteasomal chaperones as previously reported[38]. Particles were reextracted (400 px, 1.07 Å/px) and refined via non-uniform refinement imposing a C2 symmetry after one further round of 2D classification (cryoSPARC v 4.2.1). Resulting poses were used for C2 symmetry expansion and signal of one half of the particles was subtracted along the symmetry axis. Following ab-initio reconstruction and heterogeneous refinement, particles were downsampled (200 px boxsize, 2.14 Å/px), converted into star files using cs2star/pyem (https://github.com/brisvag/cs2star)[39] and refined in relion (v 5.0)[40]. Next, two rounds of 3D classification without image alignment (Regularization parameter T = 40) and blush regularisation[41] were performed. A map with a well-defined IPO9 density was used as a reference for a heterogenous refinement in cryoSPARC sorting particles which were initially imported into RELION. IPO9 bound particles were then again imported into Relion and 3D classified with image alignment (T = 4) further excluding badly resolving particles. Next, particles were reextracted (300 px boxsize, 1.43 Å/px) once refined without a mask followed by a focussed refinement with a mask encompassing the AKIRIN2 and IPO9 density. To reduce conformational heterogeneity, particles were 3D classified without image alignment (T = 100). The better resolving class was refined in cryoSPARC or in RELION before performing local refinements on the 20S alpha-ring and the IPO9: AKIRIN2 densities or the densities of IPO9 and the second AKIRIN2 dimer, respectively. Local resolution was estimated for both refinements in cryoSPARC. The final maps were post-processed using DeepEMhancer[42] using the developer provided 'highRes' weights. The pixel size was corrected to 1.055 Å/px (1.407 Å/px in the final map).

The composite map was generated using the 'volume zone' command in ChimeraX to exclude regions of either the consensus or local refinement. Due to these manipulations this artificial 'franken-map' was only used for visualisation purposes.

The movies of the apo-IPO9:AKIRIN2 dataset were patch motion corrected and patch CTF estimated in cryoSPARC live (v 4.1.1) and micrographs were curated based on defocus and CTF-fit values resulting in 7,734 exposures. Particles were picked using cryolo (v1.9)[43] by refining the pretrained weights, provided by the developers, using 10 manually picked micrographs. Particles were randomly split into 6 separate batches and sorted in three rounds of 2D classification using optimized parameters. Particles were sorted by performing ab-initio reconstruction and using the resulting maps as reference for a heterogeneous refinement. To increase the number of particles, the final coordinates of 20 micrographs were used to further refine the cryolo weights and pick the whole dataset. These particles were sorted by iteratively performing ab-initio reconstructions and using noisy and dirt maps as decoys to exclude respective particles in heterogenous refinements. The cleaned particle stacks of both cryolo trainings were combined, duplicates removed and reextracted (120 px boxsize, 1.59 Å/px), imported into relion and particles were refined using blush regularization. GSFC and local resolution were estimated in cryoSPARC using the relion half-maps. The 3D refined map was post-processed using DeepEMhancer.

## Cryo-EM model refinement

As an initial model for the 20S:AKIRIN2:IPO9 PDB-7NHT was used, together with AlphaFold2 predictions[44] for IPO9 and the AKIRIN2:dimers. Proteasomal subunits, the AKIRIN2:dimer I, AKIRIN2:dimer II wing helix and IPO9 residues 696-1041 were modelled in the consensus 3D refined map. The AKIRIN2:dimer II and IPO9 residues 1-695 were modelled in the map, locally refined on the 20S alpha-ring and the IPO9: AKIRIN2 densities. Structures were fitted into the cryo-EM maps

via rigid body fitting in chimeraX (v.18)[45,46] and unresolved regions of AKIRIN2 and IPO9 were removed and structures fitted into cryo-EM maps by flexible fitting using the namdinator web interface (https://namdinator.au.dk/)[47]. Structures were manually built by iteratively using Isolde (v 1.8[46,48,49]) and/or Coot (v 0.9.8[50–52]) followed by real space refinement in Phenix (v 1.21[52–55]). The composite model was generated by combining models from the consensus and local refinements, generating a covalent bond between IPO9 residues 695 & 696 and refining the bond's geometry in Isolde.

The initial models for the IPO9 and the AKIRIN2 dimer were generated using AlphaFold2; unresolved regions were removed and structure fitted into the cryo-EM maps using flexible fitting using the namdinator web interface (https://namdinator.au.dk/)[47]. The model was refined using Rosetta via iterative local rebuilding. The protocol was run 1000 times. The 25 best models according to the resulting energies were manually inspected[56].

## Nuclear import assays

For the nuclear import assays, HeLa cells were seeded 1-2 days before the experiment on ibidi 8-well μ-slides in 300 μL DMEM to reach a cell density of 70-80% the day of the experiment.

On the day of the experiment, the dish was placed on ice and cells were washed once with PBS and twice with freshly prepared transport buffer (20 mM HEPES pH7.3, 110 mM KOAc, 2 mM Mg(OAc)$_2$, 1 mM EGTA, 2 mM DTT, 1 mM PMSF, 1x cOmplete protease inhibitor, 250 mM sucrose). For permeabilization of the cellular membrane cells were incubated with 0.005% digitonin in transport buffer for 5 min on ice. Afterwards, cells are thoroughly but carefully washed 3x with transport buffer (3 × 5 min). In the meantime, import mixes are prepared in transport buffer supplemented with an ATP regeneration mix (1 mM ATP, 10 mM PEP, 20 U/mL phosphokinase), 2 mg/mL BSA, 4 μM Ran, 2 μM mCherry (exclusion marker). In the different trials combinations of 50 nM A647-labelled 20S proteasomes, 7.5 μM GFP-AKIRIN2, 1 μM IPO9, 1 μM KPNA2 and 1 μM KPNB1 are added (any deviations in protein concentrations are indicated in the figure text). Import mixes are added to the permeabilized cells and the dish is incubated for 60 min (if not indicated otherwise) at 30 °C in a humidity chamber. Cells are washed 2x with transport buffer and once with PBS. Cells are incubated with 0.2 μg/mL HOECHST 33342 in PBS for 10 min in the dark on RT. Cells are washed once more with PBS and imaged in PBS immediately after on an Olympus IX83 Spinning Disk Confocal SoRa equipped with a Yokogawa W1 spinning disk microscope.

## NIA quantification

To determine fluorescent intensities a custom Fiji[57] macro was created. Cellpose 2.0[58] was used to segment nuclei and cytoplasm and define ROIs for the measurements. Therefore we modified two pretrained models, retraining them with our own data and annotations, using Dapi for nuclear segmentation and Ch2 for defining the cytoplasm. Within the macro we ran Cellpose using our models on each image, then combining the resulting ROIs and measuring mean intensities in all channels.

## Microscale thermophoresis assays (MST)

For the sample preparation GFP-labelled AKIRIN2 and IPO9 were prepared in BufferMST (50 mM Tris pH7.5, 150 mM NaCl, 0.005% Tween-20). Samples were spin filtered (Ultrafree-MC-HV Durapore PVDF 0.45 μm Centrifugal Filters) after which concentrations were determined at 488 nm for GFP-AKIRIN2 and at 280 nm (corrected for scattering at 333 nm) for IPO9 in a 0.15 cm cuvette using the according extinction coefficients. For the MST measurement, different dilutions of GFP-AKIRIN2 were tested for its signal in the NT.115 Monolith in different capillary types at different MST powers. A final concentration of 13 nM for GFP-AKIRIN2, premium capillaries and medium MST power were concluded to be used for the binding experiments. Three

replicates of a 15 point 1:1 dilution series of IPO9 with 14.1 μM as highest concentration and a 16th point without IPO9 were measured. The interaction was measured once without additives and once with 7.7 μM Ran and 1 mM GTP present in all samples. Samples have been incubated for about 10 min at RT prior to the measurement. For the data analysis, hot and cold regions were selected from -[−1 s, 0 s] and [1 s, 5 s] IR-Laser on-time, respectively (Supplementary Fig. 3e). Data has been analysed and exported from MO. Affinity Analysis for replotting. PALMIST[59–61] was used to quantify the equilibrium dissociation constants ($K_D$s) of the binding curves. The best fit $K_D$s as well as the non-symmetric 95% confidence intervals, were returned.

## Grating-coupled interferometry (GCI)

The Creoptix WAVE system (Malvern Panalytical) was used to perform 20S Proteasome-AKIRIN2 binding experiments. All experiments were performed on 4PCP WAVEchip (Malvern Panalytical). The chip was first conditioned with 100 mM sodium borate pH9 and 1 M NaCl, followed by immobilization of 2000 pg/mm$^2$ 20S proteasome on the measurement channel and BSA on measurement and reference channel using standard amine-coupling (activation with 1:1 mix of EDC/NHS, binding of 20S proteasome and BSA with sodium acetate, pH 4, quenching with 1 M ethanolamine pH8). 0.1X PBS was used as running buffer during immobilization. BufferGCI (50 mM TRIS pH7.5, 100 mM NaCl, 0.005% Tween-20, 1 mg/mL BSA, 1 mM TCEP) was used during the experiment. AKIRIN2 was injected in a 1:1 dilution series in the range of 93.75 nM – 12 μM in BufferGCI at 25 °C. Measurements were performed in three replicates, with a regeneration step between each set of replicates using BufferGCI supplemented with NaCl to reach 300 mM. Data was double referenced using a reference channel subtracted from the measurement channel and blank injections. Individual injections were bulk corrected, correction and analysis were performed using the Creoptix WAVEcontrol software (Malvern Panalytical). A one-to-one binding model was applied to fit the experiment.

## E1 and proteasome inhibition

The day before the actual experiment, RKO cells were seeded on 3 × 10 cm tissue culture dishes. The next day cell density was about 60-70% and cells were either treated with 10 μM MG132 (proteasome inhibitor), 0.75 μM TAK-243 (E1-inhibitor) or 5 μL DMSO (control) each for 4 h. Afterwards, cells were harvested with a cell scraper, washed once with PBS and lysed in RIPA buffer (50 mM Tris pH8, 150 mM NaCl, 1% Triton, 0.5% sodium deoxycholate, 0.1% SDS) supplemented with 1x cOmplete protease inhibitor, incubated for 20 min on ice, spun down full speed 21,300 x g, for 10 min. Supernatant was transferred into a fresh tube and concentration was measured with Bradford. From each sample, aliquots containing 35 μg total protein were taken and SDS loading dye was added. Samples were cooked for 5 min, cooled down, and loaded on a 4-20% TGX Stain-Free™ Protein Gel (BioRad) and run at 200 V for 45 min. Gel was transferred to a nitrocellulose membrane and probed for the desired proteins (AKIRIN2, c-MYC, ubiquitin, vinculin and histone H3).

## Stoichiometry estimation by chromatography and mass photometry

Samples were prepared in buffer MP (50 mM HEPES pH 7.5, 50 mM NaCl, 0.5 mM TCEP) using the following input protein concentrations unless otherwise specified: 100 pmol 20S, 1000 pmol AKIRIN2, 500 pmol IPO9, 1000 pmol KPNA2, 1000 pmol KPNB1.

Following 1 h of dialysis in buffer MP, samples were applied to a size exclusion column (Superose 6 increase 3.2/300, Cytiva), which was equilibrated with buffer MP. 50 μL fractions were collected and peak fractions were further analysed using SDS-PAGE Biorad TGX stainfree gel and mass photometry.

Mass photometry measurements were performed as described by Sonn-Segev et al.[62] if not mentioned otherwise. Measurements were

performed on a TwoMP instrument (ReFeyn). Refeyn MassFerence P1 diluted in buffer MP was used for calibration using a medium field of view. AcquireMP software (ReFeyn) was used for data acquisition. DiscoverMP software was used for raw data processing and manual fitting of Gaussian peaks. Mass photometry measurements were plotted as histograms of distribution of molar mass (kDa).

## Reporting summary
Further information on research design is available in the Nature Portfolio Reporting Summary linked to this article.

## Data availability
Following atomic models and cryo-EM densities, for the 20S proteasome, IPO9 and AKIRIN2 complex, were deposited in the Protein Data Bank (PDB) and the Electron Microscopy Data Bank (EMDB); the composite map and model under the accession code PDB-9QOO/EMD-53265 (https://www.ebi.ac.uk/pdbe/entry/emdb/EMD-53265), local refinement focused on the 20S-alpha subunits, IPO9 and AKIRIN2 under the accession code PDB-9QON/EMD-53264 (https://www.ebi.ac.uk/pdbe/entry/emdb/EMD-53264), local refinement focused on IPO9 and one homodimer AKIRIN2 under the accession code PDB-9QNO/EMD-53248 (https://www.ebi.ac.uk/pdbe/entry/emdb/EMD-53248). Additionally, a lower resolution map for the consensus refinement of the complex was deposited under the accession code EMD-53246. For the complex of IPO9 and AKIRIN2 an atomic model and cryo-EM density were deposited under the accession code PDB-9QOP/EMD-53266 (https://www.ebi.ac.uk/pdbe/entry/emdb/EMD-53266). The motion corrected micrographs and final particle coordinates were submitted to the EMPIAR database under accession code EMPIAR-12783 for 20S-AKIRIN2-IPO9 and EMPIAR-12784 for IPO9-AKIRIN2. DNA sequencing data from the saturation mutagenesis screen have been uploaded to SRA under the accession code PRJNA1414642. Mass spectrometry datasets have been uploaded to PRoteomics IDEntifications Database (PRIDE) under project accession code PXD052012. All unique/stable reagents generated in this study are available from the lead contact with a completed material transfer agreement. Other source data are provided with this paper.

## Code availability
Code for the design and analyse of the saturation mutagenesis screen is available upon request.

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

## Acknowledgements

We thank all members of the Haselbach and Zuber group for their great support and scientific discussions, the Electron Microscopy facility of the Vienna BioCenter, in particular Thomas Heuser, Victor-Valentin Hodirnau at Institute of Science and Technology Austria EM facility for cryo-EM data collection. We are thankful to the Peters, Plaschka and Clausen group for sharing chemicals and lab equipment. This work was further supported by Vienna BioCenter Core Facilities: the Light Microscopy Facility at Vienna BioCenter Core Facilities (VBCF) and the IMP/IMB/GMI BioOptics Facility, in particular Thomas Lendl and Pawel Pasierbek; by the Protein Technologies Facility at the Vienna BioCenter Core Facilities (VBCF), in particular Arthur Sedivy; and Proteomics analyses were performed by the Proteomics Facility at IMP/IMBA/GMI using the VBCF instrument pool, in particular Otto Hudecz and Ines Steinmacher. We gratefully thank Alexander Schleiffer and Thomas Burkard for evolutionary conservation analysis; Dilantha Perera and Marius Pörschke for their help with nuclear import assays; Achim Dickmanns, Julian Ehrmann, and Ulrich Hohmann for providing purified proteins. We also thank Maximilian Spicer and Milka Kostic (Life Science Editors). This work was supported by the European Union's Framework Programme for Research and Innovation Horizon 2020 (2014-2020) (ZH), Marie Curie Skłodowska Grant Agreement Nr. 847548 (ZH), FWF SFB F79 (https://doi.org/10.55776/F79) targeted protein degradation (HLB, HKn) and Boehringer Ingelheim (HLB, Haselbach lab). For the purpose of open access, the author has applied a CC BY public copyright license to any author accepted manuscript (AAM) version arising from this submission.

## Author contributions

Conceptualization: H.L.B., R.W.K., J.Z., D.H. Methodology: H.L.B., R.W.K., L.G., Z.H., Z.K., I.G., M.A., M.H., H.Kn., S.H., F.A., H.Ko., A.D., S.C.H., J.Z., D.H. Investigation: H.L.B., R.W.K., L.G. Visualization: H.L.B., R.W.K., L.G. Funding acquisition: D.H. Project administration: D.H. Supervision: D.H., J.Z. Writing—original draft: H.L.B., R.W.K. Writing—review and editing: H.L.B., R.W.K., D.H., J.Z., Z.H.

## Competing interests

J.Z. is a founder, shareholder and scientific advisor of Quantro Therapeutics. J.Z., D.H. and the Zuber and Haselbach laboratories receive research support and funding from Boehringer Ingelheim. The remaining authors declare no competing interests.
