## [Transparent Peer Review file · Nature Communications]

A multivalent adaptor mechanism drives the nuclear import of proteasomes

Corresponding Author: Dr David Haselbach

Version 0:

Reviewer comments:

Reviewer #1

(Remarks to the Author)

Brunner et al. describe a nuclear import mechanism of the proteasome via Akirin2. Authors provide new insights on Akirin2 function that it serves as an adaptor for anchoring IPO9 to the proteasome, and that Akirin2 itself oligomerizes/clusters on the 20S proteasome, presenting multiple NLSs, which additionally recruit KPNA1/KPNA2 to form a nuclear import complex. These findings are intriguing and may potentially advance the fundamental understanding of nuclear import mechanisms of the proteasome. Cryo-EM structures of the nuclear import complex are exciting. However, there are several concerns in the current form of the manuscript. Structural data are not described in detail and do not include features that are directly relevant to the proposed Akirin2 clustering (key to multivalency) on the 20S. Also, validation of the cryo-EM data is lacking in several aspects. In total, major revision is required with specific points to address as follows.

1. Structural plasticity of Akirin2 (& IPO9) is interesting and may be crucial to its function in proteasome nuclear import. However, this aspect is not clearly explained. In their previous study (Almeida et al. Nature 2021), 76% of Akirin2 was unresolved. In the present study, what % of Akirin2 is resolved in binary & ternary structures (IPO9-Akirin2 & IPO9-Akirin2-proteasome complexes)? Also, are different regions of Akirin2 disordered vs. ordered between these 2 structures? Since certain arrangement/binding interface is stabilized in the ternary complex, detailed description of this aspect will be informative in understanding Akirin2 mechanisms.
2. As for Akirin2 clustering on the proteasome, authors should validate the proposed stoichiometry: 4 Akirin2 dimers relative to one IPO9 or 20S proteasome. In Fig. 2C co-sedimentation data, this information is unclear.
3. In addition to the schematics (Fig. 3J) depicting Akirin2 clustering on the proteasome, authors should include the actual cryo-EM data (several different views) to show the relative positioning of Akirin2 dimer I & II on the 20S. The following specific aspects need to be addressed.
 - 3a. As key regions that mediate Akirin2 binding/clustering to the proteasome, authors should clearly describe where the *4 identical* SYVS motifs (of Akirin2 dimer I & II) individually dock onto *4 different* binding sites on the heteroheptameric alpha ring of the proteasome. Also, where do 2 *identical* coiled-coil region & the wing helix binds to 2 *different* sites on the proteasome?
 - 3b. Authors should reconcile how Akirin2 dimer I & II can be arranged in parallel in the ternary complex (Fig. 3J). Akirin2 C-termini would be at the periphery of 20S, based on the previous paper (Almeida et al. Nature 2021). However, the N-termini should point toward the central pore (gate) of the 20S alpha ring based on the previous study where coiled-coil interacts with alpha 1, 5, 6 gating termini. The N-termini of Akirin2 dimers would then be expected to converge toward the center, rather than assuming the depicted parallel arrangement.
 - 3c. If alpha3 pocket is used to house the SYVS motif as described in the previous paper, authors should clarify how this same pocket is used for binding both SYVS motif as well as the critical lys94.
4. Line 189-190. The wing helix of the Akirin2 interacts with the negatively charged surface of alpha6 (fig. 3e). Which residues of alpha6 are involved in this interaction? Because the previous study showed that akirin2 coiled coil reaches out to alpha6 N-termini. Is it generally similar position of the alpha6 N-termini?

5. Authors claim that IPO9 assumes the “open” state in the ternary complex (Fig. 3J). However, functional significance of the open vs. closed state is unclear. Authors should test this aspect by forcing the closed state (mutating the contact sites specific to the open state either on Akirin2 or IPO9). How does such disruption (keeping IPO9 in the closed state) influence Akirin2 clustering, proteasome nuclear import, or further recruitment of other importins?

6. Line 199-203. Authors should clarify whether “multiple importins” in this sentence refers to multiple IPO9s or other importins, which are described later in the paper. Based on the arrangement of IPO9-Akirin2-20S ternary complex, it is difficult to imagine how multiple IPO9s can be recruited by 2 Akirin2 dimers to the same side of the proteasome.

7. Is akirin2 clustering on the proteasome specific to the ternary complex or also observed on Akirin2-20S complex? In Extended Data Fig. 3e, f, authors describe that Akirin2-20S affinity is low as compared to Akirin2-IPO9 affinity. What happens to Akirin2-20S affinity when IPO9 is present?

Reviewer #2

(Remarks to the Author)

This manuscript by Brunner et al. investigates the assembly mechanism of the nuclear import complex for the human proteasome. The authors focus on the essential role of the protein AKIRIN2 therein, a protein that was previously found to control the nuclear import of proteasomes in vertebrates (PMID: 34711951).

There is increasing interest in understanding the mechanisms of organelle-specific quality control; the nucleus represents a particularly interesting compartment because of its central role in cell biology, its important roles in aging and disease, and especially because protein quality control in the nucleus is highly proteasome-dependent.

Specifically, the authors present i. data on AKIRIN2 function by saturation mutagenesis; ii. a cryoEM structural analysis of AKIRIN2 that shows how it allows for interactions between the IPO9 importin and the proteasome; and iii. a mechanism for binding of AKIRIN2 to importins via its NLS, showing that binding of AKIRIN2 to either IPO9 or KPNA2/KPNB1 importins is sufficient for nuclear import of proteasomes.

The findings, which are the result of applying a variety of different methods, support a critical role for AKIRIN2 as a scaffold for the assembly of the importin cluster at the proteasome and are a logical follow-up to their previous work (PMID: 34711951) to further understand the role of AKIRIN2 in nuclear proteasome import.

The paper covers an important biological mechanism and the presented data appears to be of high quality. However, several issues need to be addressed before the paper can be recommended for publication:

1. Importantly, it is not clear whether the described phenotypes in the study depend exclusively on a direct effect of AKIRIN2 on proteasome shuttling, or whether independent functions of AKIRIN2 also contribute. Are there alternative explanations for the observed effects due to AKIRIN2 loss-of-function? Preferably, a rescue-type experiment should be performed, in which restored proteasome localization rescues the effect of AKIRIN2 loss-of-function. The authors should at least adequately discuss this issue concerning inference of causality. Additionally, would it be feasible to fuse a nuclear export sequence to AKIRIN2 and look for its effect on proteasome localization and function?

2. The introduction and discussion provide some examples on nuclear import of large protein complexes. However, there appears to be minimal mention of a number of pertinent previous studies on proteasome localization, besides references to some previous work done in yeast (Sts1). It would be appropriate to include a more detailed overview of key studies on proteasome localization in different model systems, which should indicate how AKIRIN2 compares to previously identified factors (e.g., Rpn2 bipartite NLS [PMID: 15210724]) and how the new findings fit into the larger picture. Are there different ways by which the proteasome can enter the nucleus?

3. The results and interpretation of the saturation mutagenesis screen should be discussed in more detail, i.e., how is a critical residue defined exactly and what are the key findings (specific mutation → phenotypes)? Critically, the authors should perform validation experiments to show that the identified critical residues from their screen are indeed responsible for the observed phenotypes. Validation of these key AKIRIN2 mutations in independent experiments can be performed without reliance on reporters, e.g., by looking at endogenous MYC. To what extent do specific mutations phenocopy AKIRIN2 KO?

4. The authors focus on the 20S proteasome “to reduce the complexity”, however, the implications of this are inadequately discussed. Other forms of the proteasome may also be transported. What is known about the distributions, transport, and functions of 19S, 20S, and 26S proteasomes in nucleus vs. cytoplasm? AKIRIN2 binding likely prevents double capped proteasome from being translocated into the nucleus? The authors should comment on this in greater detail and discuss the

limitations of their experiments and of their presented model, as well as how these issues can potentially be addressed in future work.

5. The cryoEM structures appear to be of good quality. Is there a potential role for other IPO9 adaptors? The authors highlight some specific features of the structures, e.g., relevance of residue K94 (e.g., its role in stabilizing the interaction with the proteasome), but it should be made clearer what is speculation based on the structure and what is supported by experiment.

6. The section "Conformational plasticity underpins AKIRIN2-mediated complex assembly" describes the structure of the AKIRIN2-IPO9 binary complex, stating "this structural plasticity illuminates a dynamic assembly mechanism for large nuclear import complexes". However, if and how this pertains to proteasome import is unclear. Is this structure in absence of proteasome likely to be relevant, considering the variety of states the proteasome can adopt itself? It is not clear to the reviewer what is to be learned from the AKIRIN2-IPO9 binary complex structure.

7. It is stated that AKIRIN2 itself is rapidly degraded by nuclear proteasomes in a ubiquitin-independent manner, based on experiments with inhibitors. Is this degradation of AKIRIN2 19S or 20S-dependent?

8. Like most good studies, the work also raises many new questions. The authors provide some statements in the discussion that deserve more scrutiny. For example, a potential role for Midnolin as a trigger in AKIRIN2 degradation is mentioned, however, this appears to be not supported by any data. If this idea is correct, it should be relatively easy to test. Is Midnolin-dependent degradation relevant in the cell types used in this study, i.e., is it a plausible mechanism? Further, the authors mention proteins with analogous functions to AKIRIN2 for other large complexes, and describe a common architecture for these adaptor proteins, by which they could perform multiple critical functions. While interesting conceptually, it may be good to include a sentence on what remains to be demonstrated experimentally for these particular examples.

Reviewer #3

(Remarks to the Author)

Nuclear import of a wide range of substrates through nuclear pore complexes (NPCs) is a critical process in eukaryotic cells. While nucleocytoplasmic transport in general is well studied, most previous work has generally focused on a few transport pathways and simplified or small cargos. This paper takes an important step forward by identifying an intriguing multiple importin mechanism to import proteasomes. The authors build on their previous work where they identified AKIRIN2 as a factor that binds to 20S proteasomes and mediates their import. To identify functionally relevant residues, they developed three functional assays and monitored the effects of individually changing every amino acid in AKIRIN2 to every other amino acid or deleting it. They assessed (i) proteasomal import; (ii) nuclear proteasome activity; and (iii) cell survival. All screens yielded highly consistent results, confirming previous results and identifying new sites of interest as potentially crucial interaction points or regulatory sites. This was well done and is an important functional dataset for AKIRIN2. They then focused on identifying import receptors that interacted with AKIRIN2. Using co-immunoprecipitation coupled with mass spectrometry, they identified AKIRIN2 as one of the most enriched interactors with IPO9. AKIRIN2, IPO9, and 20S proteasomes formed a complex separable on a sucrose density gradient. Cryo-EM was used to solve the structure of this complex, which revealed two AKIRIN2 dimers and one IPO9 molecule at the top end of the proteasome. Critical interaction sites were consistent with critical residues identified in their functional screens. They then focused on the largely disordered N-terminal region of AKIRIN2 (not visible in the cryo-EM structure), which has a bipartite NLS. As a super-complex of KPNA2 + KPNB1 (the canonical importin alpha/beta pair used for nuclear import), IPO9, AKIRIN2, and 20S could be isolated, this led to model in which the import complex consists of one molecule of IPO9 and 4 importin alpha/beta heterodimers bound to a proteasome mediated by two AKIRIN2 dimers. The KPNB1/KPNA2 receptors were dissociated by RanGTP in the expected manner. The IPO9 dissociation mechanism remains unclear but is likely coupled with the ubiquitin-independent proteasomal degradation of AKIRIN2. As there are other proteins with analogous functions to AKIRIN2, the import mechanism identified may be a new paradigm for large protein complexes. This is a well-written and convincing paper that provides a major advance.

There are numerous places within the figures and captions where it should be made clearer what is being shown so that it is easier to understand the experiment for a non-expert, particularly with regards to abbreviations.

The nuclear import assays in Fig. 4 should be improved/clarified. The imaging timepoint is not stated. The Methods say 30-60 min, but this should really be done with a kinetic curve to clarify the effects of the different transport factors. Is there any reason this cannot be done by collecting an image every minute or so? The cells are washed and it takes about 15 min before they image, during which material could leak out (if they haven't confirmed that it does not). The amount of import should be quantified for all conditions. The conditions of Fig. 4d (i.e., the transport factors used) are not stated. This shows only a 2-fold increase with a very high baseline import. A few changes could increase import efficiency. An ATP regenerating system works in the permeabilized cell transport assay, but GTP is better as Ran is a GTPase. Ran is typically purified in both GDP and GTP forms, which can be separated on an anion exchange column. It is not clear if they did this. Cytoplasmic RanGTP would dissociate import complexes. NTF2 should be added to shuttle Ran into the nucleus.

The model presented is quite interesting, and logically follows from their data. However, I wonder about stability in the complex environment of the NPC permeability barrier. The IPO9 forms an open structure when attached to proteasomes, but it does appear fairly firmly attached to AKIRIN2. However, the KPNB1/KPNA2 heterodimers are predicted to be up to 18 nm away on flexible tethers. It seems like there could be a lot of strain on this structure as it is migrating through the pore and all the intervening space is filled by FG-polypeptides. Is there any evidence that the KPNB1/KPNA2 heterodimers have

additional attachments to the proteasome?

Minor points:

- 1) line 36 – ‘cyto- and nucleoplasm’ is a strange construction. Just expand ‘cyto-’ to cytoplasm.
- 2) lines 43 and 329 – Tu et al (EMBO J. 2013 32:3220) showed the effects of import receptor number on nuclear transport of a large protein cargo.
- 3) line 79 – at first mention, more clarity on the mCherry-MYC fusion protein is needed, i.e., that MYC is a transcription factor degraded by nuclear proteasomes (and not simply a common epitope tag).
- 4) lines 98-99 – this statement should be clarified as it does not appear to be true based on Fig. 1c and ED Fig. 1c,d.
- 5) lines 288-290 – it’s possible that an IPO9 recycling factor is needed for RanGTP dependent release.
- 6) Fig. 1c,d – (c) what dataset is this? Clarify as was done for ED Fig. 1c,d. (d) The caption identifies colored boxes as ‘squares’ when these are in fact rectangles. Can (c) and (d) be combined or aligned better so they match up exactly?
- 7) Fig. 2 – Do the StayGold and mCherry images in (a) correspond to the same cells? What is AAVS1? What are dashed lines and FC in (b)? How are proteins identified in (c)? The cartoon for Akirin2 suggests a fragment rather than the whole protein. Purification of a fragment was not found in the Methods. Single load lanes should be shown somewhere. How were proteins identified in the top gel if this is stain free? I didn’t figure out until the end of the paper that the top and bottom gels were different (one is fluorescence) – this needs to be indicated more prominently. There is no Fig. 2d (line 158).
- 8) Fig. 4 – (c) what is the imaging timepoint? (d) is the excess a ‘molar excess’? So, the maximum extra import of proteasome import is two-fold? This doesn’t sound like much. Is this a timepoint or kinetic constant?
- 9) lines 323-326 – These two sentences are confusing and should be re-written.
- 10) ED Fig. 1b – show StayGold and mCherry images separately.
- 11) ED Fig 2a – I presume that these are the types of images evaluated for their three screens. It would be helpful to provide a bit more information indicating this.
- 12) ED Fig. 3 – (a) KPNB1 and KPNA2 were not indicated in the caption. (d) ‘cooked’ in the caption is not a commonly used expression, so some explanation is needed. This sentence also has two consecutive ‘which’ phrases, an awkward construction. (e) and (f) what are MST and GCI? The number format 40 [10,100] is unclear. ka and kd do not have units. It is unclear what is shown in the (f) panel (what is fit and what is data?) – for dissociation, the fitted curves don’t match well with experiment.
- 13) ED Fig. 7b – AF2 should be spelled out.

Reviewer #4

(Remarks to the Author)

Version 1:

Reviewer comments:

Reviewer #1

(Remarks to the Author)

The authors appropriately addressed all of my comments both in the text and in the figures. I am pleased with the revised manuscript and fully support the publication of this manuscript.

Reviewer #2

(Remarks to the Author)

The manuscript has been significantly improved. However, there are still some important remaining questions.

Reviewer 2:

1. Importantly, it is not clear whether the described phenotypes in the study depend exclusively on a direct effect of AKIRIN2 on proteasome shuttling, or whether independent functions of AKIRIN2 also contribute. Are there alternative explanations for the observed effects due to AKIRIN2 loss-of-function? Preferably, a rescue-type experiment should be performed, in which restored proteasome localization rescues the effect of AKIRIN2 loss-of-function. The authors should at least adequately

discuss this issue concerning inference of causality. Additionally, would it be feasible to fuse a nuclear export sequence to AKIRIN2 and look for its effect on proteasome localization and function?

We appreciate the reviewer's question regarding causality and alternative explanations for our observed phenotypes. Our comprehensive saturation mutagenesis screen employed three independent readouts including proteasome localization, MYC protein levels, and cell growth, all of which showed highly concordant results. AKIRIN2 mutants that disrupt proteasome nuclear localization consistently exhibit both elevated MYC levels and growth defects, while control variants that maintain proteasome import function do not exhibit these phenotypes. This extensive dataset suggests a direct causal relationship between AKIRIN2's proteasome import function and the observed cellular effects.

Our earlier work (PMID: 34711951) demonstrated that AKIRIN2 loss-of-function does not affect immediate transcriptional responses, which argues against indirect transcriptional effects as an alternative explanation. The tight correlation between proteasome mislocalization and substrate accumulation across multiple independent variants supports direct causality rather than pleiotropic effects from other AKIRIN2 functions.

Regarding rescue experiments, our saturation mutagenesis approach already provides the functional equivalent and goes far beyond what traditional rescue experiments could offer.

Testing numerous variants with different functional capacities provides more robust evidence for causality than relying on a single rescue construct.

The suggested AKIRIN2-NES fusion experiment would add complexity without providing meaningful additional insight. Of note already deletion of the NLS leads to cytoplasmic accumulation of AKIRIN2. Given that our current approach already suggests causality through multiple independent readouts across numerous variants, additional complex experimental manipulations are beyond the scope of our manuscript.

We believe our comprehensive approach provides definitive evidence that AKIRIN2's effects on cellular phenotypes are mediated through its role in proteasome nuclear import.

Comment: Their arguments are received. Note that, the observation that AKIRIN2 mutations have similar effects on proteasome localization and on MYC level is consistent with but does not directly support a "causal relationship between AKIRIN2's proteasome import function and the observed cellular effects", because their current results do not exclude the possibility that AKIRIN2 may serve as an import adaptor for other key factors in the nucleus (there are actually a few AKIRIN2-interacting proteins essential in the nucleus according to Biogrid) or other import-independent essential functions. Most mutations may perturb the fold or stability of AKIRIN2, leading to general LOF. Adding to this suspicion is the fact that AKIRIN2 is codependent with factors in various biological functions on Depmap including ribosome process, DNA integrity etc. Although a few proteasome genes are codependent with AKIRIN2, most are the 19S subunits, inconsistent with their proposed mechanism. A rescue experiment would be perhaps the simplest way to complete their argument.

In addition, there appears some discrepancy between MYC stability and proteasome localization (supp. fig 1. c/d). Are those due to technical reasons or have biological underpinnings? It would be useful to have e.g. a scatter plot to correlate these two effects and highlight the outliers. Could they also indicate the functionally important sites on the ternary structure?

2. The introduction and discussion provide some examples on nuclear import of large protein complexes. However, there appears to be minimal mention of a number of pertinent previous studies on proteasome localization, besides references to some previous work done in yeast (Sts1). It would be appropriate to include a more detailed overview of key studies on proteasome localization in different model systems, which should indicate how AKIRIN2 compares to previously identified factors (e.g., Rpn2 bipartite NLS [PMID: 15210724]) and how the new findings fit into the larger picture. Are there different ways by which the proteasome can enter the nucleus?

We added a more detailed overview in the introduction about previous studies about proteasome import including rpn2 and HSP90 (Xenopus), Sts1 (Yeast), AKIR-1 (C.elegans). The mentioned study of Rpn2 bipartite NLS was focusing on yeast. While proteasome subunits are quite conserved the import machinery in yeast and higher eukaryotes seems to be quite different which can be probably partially explained by the different requirements in closed vs open mitosis. Further, from these studies there seems to be evidence that the 19S (probably also base and lid) are separately imported from the 20S. We strongly believe that AKIRIN2 is the major import factor for 20S proteasomes in mammalian cells. While our previous study suggests that also the import of 19S proteasomes strongly depends on AKIRIN2, it needs further investigations to dissect if 19S proteasomes are separately imported or associated with the core particle as 26S proteasomes.

Comment: A more detailed overview of background studies has been included. The authors state that differences in import machinery between yeast and higher eukaryotes may be partially explained by open vs. closed mitosis. Is data available to back this idea for proteasome specifically? The basis of this difference seems mechanistically important and it would be worthwhile to discuss this in the paper.

3. The results and interpretation of the saturation mutagenesis screen should be discussed in more detail, i.e., how is a critical residue defined exactly and what are the key findings (specific mutation -> phenotypes)? Critically, the authors should perform validation experiments to show that the identified critical residues from their screen are indeed responsible for the observed phenotypes. Validation of these key AKIRIN2 mutations in independent experiments can be performed without reliance on reporters, e.g., by looking at endogenous MYC. To what extent do specific mutations phenocopy AKIRIN2 KO?

Critical residues are defined based on statistical thresholds across our three readouts, with specific cutoffs detailed in the methods section.

Regarding validation experiments, our study design already incorporates extensive validation through multiple independent approaches. Our saturation mutagenesis screen tested every possible single amino acid substitution across AKIRIN2, providing comprehensive validation of critical residues through three distinct readouts including proteasome localization, cell growth (dropout screen), and MYC reporter levels. Importantly, one of our readouts is entirely reporter-independent, eliminating concerns about reporter-specific artifacts.

We performed additional independent validation experiments for selected critical mutants as shown in Extended Data Figure 2, which demonstrates that individual mutations recapitulate the key phenotypes including defective proteasomal localization, elevated MYC levels, and altered protein expression. These validation experiments confirm that specific

mutations

phenocopy AKIRIN2 knockout effects to varying degrees depending on the severity of the functional disruption.

The coherence across all three readouts provides robust validation that goes beyond traditional approaches. Rather than testing a handful of mutations in isolation, our comprehensive screen validates the functional importance of each residue position across the entire protein, with independent confirmation through multiple phenotypic measures. This approach provides more thorough validation than selective testing of individual mutants, as it demonstrates the consistency of structure-function relationships across the entire protein.

We added discussion of more specific examples of individual mutations and their quantitative effects across the different phenotypic readouts to better illustrate these relationships.

Comment: OK

4. The authors focus on the 20S proteasome “to reduce the complexity”, however, the implications of this are inadequately discussed. Other forms of the proteasome may also be transported. What is known about the distributions, transport, and functions of 19S, 20S, and 26S proteasomes in nucleus vs. cytoplasm? AKIRIN2 binding likely prevents double capped proteasome from being translocated into the nucleus? The authors should comment on this in greater detail and discuss the limitations of their experiments and of their presented model, as well as how these issues can potentially be addressed in future work.

We agree that focusing on the 20S proteasome is a limitation of this study and appreciate the opportunity to discuss this more thoroughly. The nuclear transport and assembly of different proteasome forms remains incompletely understood, representing an important area for future investigation.

Our previous study demonstrated that 19S subunit nuclear import also depends on AKIRIN2, as AKIRIN2 knockout resulted in nuclear exclusion of tagged 19S subunits that were confirmed to be fully incorporated into functional 19S/26S complexes rather than orphaned. However, it remains unclear whether 19S subunits are imported independently or piggyback with 20S proteasomes. Studies in *Xenopus* suggest that 20S import occurs independently from most 19S components (Savulescu et al. 2011, PMID: 21289101), while yeast nuclear import adaptors like Sts1/Cut8 interact directly with 19S components, suggesting species-specific differences.

Our quantitative analysis of nuclear versus cytoplasmic distribution shows that 20S proteasomes are more than twice as abundant in the nucleus, while 19S subunits show only 30% nuclear enrichment. This differential distribution suggests that nuclear assembly and regulation of different proteasome forms may involve distinct mechanisms. Tomographic studies by the Baumeister lab suggest that nuclei also contain 30S proteasomes. However, how the equilibrium between different assemblies of the proteasome is regulated remains to be understood despite the decades of research.

The nucleus contains specific proteasome activators like PA200 and PA28 γ . Both contain their own NLS and are exclusively nuclear suggesting that they enter independently. Our current model based on 20S import provides a foundation for understanding AKIRIN2-mediated proteasome transport, but future work must address how different proteasome forms and activators are coordinately imported and assembled. The potential steric hindrance of double-capped proteasomes by AKIRIN2 binding represents another important question requiring investigation.

Comment: OK. Including all of these points briefly would greatly improve the quality of the discussion section, as they represent important questions for followup.

5. The cryoEM structures appear to be of good quality. Is there a potential role for other IPO9 adaptors? The authors highlight some specific features of the structures, e.g., relevance of residue K94 (e.g., its role in stabilizing the interaction with the proteasome), but it should be made clearer what is speculation based on the structure and what is supported by experiment.

Currently, no other IPO9 adaptor proteins have been identified in the literature, and AKIRIN2 appears to be the primary adaptor mediating proteasome nuclear import. Our nuclear import assays demonstrate that AKIRIN2 together with IPO9 and RanGTP are sufficient for proteasome nuclear import, indicating that additional adaptors are not required for this process. Our comprehensive screen also did not reveal evidence for other essential factors in this pathway.

Regarding the structural interpretations, we will clarify what represents experimental validation versus structural speculation. The importance of residue K94 and in fact all highlighted residues is supported by experimental evidence from our saturation mutagenesis screen, where mutations at this position consistently disrupted proteasome import function across all three readouts. The structural analysis provides the molecular basis for this functional requirement, showing that K94 likely stabilizes the AKIRIN2-proteasome interaction through specific contacts visible in our cryoEM structure. We have included a detailed zoom view highlighting these interactions to illustrate the structural basis for K94's functional importance.

Comment: OK

6. The section “Conformational plasticity underpins AKIRIN2-mediated complex assembly” describes the structure of the AKIRIN2-IPO9 binary complex, stating “this structural plasticity illuminates a dynamic assembly mechanism for large nuclear import complexes”. However, if and how this pertains to proteasome import is unclear. Is this structure in absence of proteasome likely to be relevant, considering the variety of states the proteasome can adopt itself? It is not clear to the reviewer what is to be learned from the AKIRIN2-IPO9 binary complex structure.

The question about the relevance of the AKIRIN2-IPO9 binary complex structure is an important point to address. This structure is crucial for understanding the complete proteasome import mechanism.

The binary complex reveals the conformational flexibility of IPO9 when bound to AKIRIN2 alone, which contrasts with the constrained arrangement in the ternary complex. This demonstrates that proteasome binding requires specific conformational selection in the IPO9- AKIRIN2 interface essential for cooperative assembly. The transition from open to closed conformations is a common regulatory mechanism in importins and necessary for understanding the complete import cycle.

The binary complex is thermodynamically more stable than the AKIRIN2-proteasome complex, and given that AKIRIN2 concentrations exceed those of proteasomes during import, this binary state may be the most prevalent form in cells. Most importantly, the binary structure elucidates how IPO9 accommodates different AKIRIN2 stoichiometries. While the binary complex shows IPO9 interacting with AKIRIN2 homodimers through one interface set, the ternary complex reveals how

conformational rearrangements allow two AKIRIN2 dimers to bind simultaneously through distinct configurations. This structural plasticity is fundamental to understanding how multiple importin complexes are recruited to proteasome import complexes. The binary structure provides essential mechanistic insight into the assembly pathway and reveals the conformational flexibility enabling cooperative binding in the functional import complex.

Comment: The authors comment on the importance of state transitions in regulating importins and the import cycle. The statement that "this binary state may be the most prevalent form in cells" seems tenuous at best, as there does not appear to be experimental data available to support this (?). While we agree that insight into structural plasticity by means of the assembly pathway will be important, the lack of proteasomes in this structure obscures proper interpretation.

7. It is stated that AKIRIN2 itself is rapidly degraded by nuclear proteasomes in a ubiquitin-independent manner, based on experiments with inhibitors. Is this degradation of AKIRIN2 19S or 20S-dependent?

Our data strongly suggests that AKIRIN2 degradation is predominantly 20S proteasome-dependent through a ubiquitin-independent mechanism. Evidence supports this conclusion: (1) AKIRIN2 itself is highly turned over by the proteasome which we have seen in our previous study by knocking out a proteasomal subunit, it scored amongst the most enriched proteins. (2) AKIRIN2 contains substantial intrinsically disordered regions, a recognized feature of proteins directly degraded by the 20S core particle without ubiquitination (PMID: 25250704).

(3) Our experiments with the E1 inhibitor TAK243 demonstrate that AKIRIN2 degradation occurs independently of ubiquitination.

(4) In preliminary in vitro degradation assays (not included in the manuscript), we observed that purified 20S proteasomes generated characteristic GFP-AKIRIN2 degradation products (AKIRIN2 degradation with stable GFP remaining), while 26S proteasomes did not produce these fragments. We included this data for the reviewer (additional Figure 2) as well as a method description (additional information point 2).

While we cannot completely exclude some contribution of the 19S regulatory particle through recently described ubiquitin-independent degradation pathways of the 26S proteasomes (PMID: 37616343), our combined data points to AKIRIN2 being primarily a direct 20S substrate. We believe the mechanism of AKIRIN2 degradation merits dedicated investigation beyond the scope of this study, which focuses on its functional role rather than its turnover mechanisms.

Comment: The in vitro degradation data (4) is compelling and we agree that it would be very interesting to explore the mechanism of AKIRIN2 degradation in more detail in future studies. It may be good to mention these preliminary results in the manuscript.

8. Like most good studies, the work also raises many new questions. The authors provide some statements in the discussion that deserve more scrutiny. For example, a potential role for Midnolin as a trigger in AKIRIN2 degradation is mentioned, however, this appears to be not supported by any data. If this idea is correct, it should be relatively easy to test. Is Midnolin-dependent degradation relevant in the cell types used in this study, i.e., is it a plausible mechanism? Further, the authors mention proteins with analogous functions to AKIRIN2 for other large complexes, and describe a common architecture for these adaptor proteins, by which they could perform multiple critical functions. While interesting conceptually, it may be good to include a sentence on what remains to be demonstrated experimentally for these particular examples.

We agree that the potential role of Midnolin is purely speculative and rather unlikely, as it lacks the ubiquitous expression pattern of AKIRIN2. We have therefore softened our statement regarding Midnolin-dependent degradation to: 'Whether additional components similar to the recently discovered protein Midnolin are required to target nuclear AKIRIN2 for nuclear degradation remains to be investigated.'

Comment: OK

In the discussion point about nuclear import adaptors, we added a sentence of the remaining questions: 'It remains to be determined whether this proposed common architecture is essential for a nuclear import adaptor function and whether the constituent building blocks are interchangeable among different adaptors.'

Comment: OK

Reviewer #3

(Remarks to the Author)

This is an excellent manuscript, and almost all of my concerns have been addressed. My remaining comments, with the exception of those on the transport assay, are minor.

75 μ M GFP-AKIRIN2 in the import assay is a very high concentration. Is this a typo? What is the AKIRIN concentration in the cell? The need for such a high concentration leads to questions of specificity, and should be discussed.

The NTF2 experiment (additional figure 4b) is very unclear due to faint images, and the data were not quantified. NTF2 is essential for the import of RanGDP (Ribbeck, 1998, EMBO J, 17:6587). If there is no NTF2 dependence, this raises questions about the RanGTP dependence of import. Perhaps they did not understand my previous comment, because it was not addressed. The Ran obtained from *E. coli* cells is isolated as a mixture of RanGTP and RanGDP forms, which can be readily separated on an anion exchange column (RanGDP elutes first). It is not clear if the authors separated these two forms of Ran. If not, the Ran added to the transport assays is a mixture of forms, and the RanGTP in the mix will dissociate import complexes leading to low transport efficiency. RanGTP also cannot be transported by NTF2, which may explain the lack of an observed effect. Finally, the authors should note that nuclei swell without addition of an osmolyte after digitonin treatment – whether damage occurs upon such swelling is unclear.

Minor points:

1) lines 41-44 – soften this statement a bit. The effect of the number of NLSs on nucleoplasmin was investigated by Dingwall (Cell 1982 30:449).

2) line 82 – 'any possible amino acid' is confusing; perhaps change to 'each of the other 19 amino acids'

- 3) Fig. 1d – the SYVS-motif is identified as YVS in the figure. Be consistent.
- 4) Fig. 2c and elsewhere – BioRad's stainfree gels require a trihalo compound for visualization after photoactivation. Therefore, 'stainfree' is misleading for this approach (recognizing that this is a brandname) as there is a visualizing agent. The approach used should be clarified in the methods.
- 5) line 325 – shouldn't this be Supplementary Fig. 3b,c?
- 6) lines 355-56 – why aren't AKIRIN2's NLSs exposed before binding to the proteasome and IPO9. Are they hidden somewhere?
- 7) Supplementary Fig. 3b,c – the markers don't appear accurate, as IPO9 moves from ~100 kD to ~150 kD
- 8) Supplementary Fig. 3e,f – Since multiple Kd value measurements were made (indicated by the range), what is n? No units are given for ka and kd (this was raised previously); no n values are provided either. 4 significant figures for the Kd is unreasonable, and 3 for the ka and kd are also likely unrealistic. It is unclear what is shown in the (f) panel (what is the fit and what is data? There seem to be three curves in each group) – for dissociation, the fitted curves don't match well with experiment. This last point was not addressed in the authors' rebuttal.
- 9) Supplementary Fig. 8 & additional figure 4 – what are the numbers on the top lines of the figure panels? Are these concentrations in micromolar? Time? In 8b and 8c, these are indicated as 20 but in the caption it states 1 μ M. Image exposures are very weak, and even expanded it is hard to see what is going on.
- 10) Supplementary Fig. 9 – what are the three columns? Three trials? What are the concentrations? Why do the c distributions look so different but the others do not? It would be helpful if the key was shown to the right of each row. The format of additional figure 5c seems better.
- 11) It is not clear why additional figures 3 and 4a are not in the Supplementary Data.

Reviewer #4

(Remarks to the Author)

Version 2:

Reviewer comments:

Reviewer #2

(Remarks to the Author)

The authors addressed most of my questions.

The authors have demonstrated clearly that AKIRIN2 regulates nuclear localization of the 20S and degradation of nuclear substrates, which are essential for cell growth. I did not question this connection but wonder if AKIRIN2 may regulate additional factors contributing to the phenotypes. Since the defects of depleting proteasome from the nucleus are very strong, this question should best be addressed by a rescue experiment. This type of experiment has been performed using nuclear anchors (e.g. Tsuchiya H. et al Biochemical and Biophysical Research Communications 436 (2013) 372–376). But I agree implementing this strategy in their system may take some time. So, I am satisfied with their current data in this part. AKIRIN2's dependency score not only has a low correlation with those of proteasome subunits among cancer cell lines, but is also much lower. The score falls out of the "essential gene" range (<-1) in ~50% of cancer cell lines. Since proteasome contains many important targets in the nucleus, perhaps the most likely explanation is that there are mechanisms rescuing proteasome localization in the absence of AKIRIN2. The author should discuss this.

Minor comment: please check if the literature references are up to date. For example, it appears the cited preprint by Monda et al. (Ref. 21) is currently published in a peer-reviewed journal.

Reviewer #3

(Remarks to the Author)

My concerns involving NTF2 and RanGTP have not been fully addressed. In their reply, the authors indicate that RanGDP and RanGTP differ in hydrodynamic radius, but they did not provide a reference that this is sufficient to separate on a size-exclusion column. They assumed that the second peak was RanGDP, but they did not do any experiments to verify this. My expectation is that RanGTP would have the 'closed' conformation, and hence would elute second on a size-exclusion column. If they add RanGTP to their import mix, any transport complexes should disassemble, reducing import. NTF2 should significantly increase the Ran gradient by importing RanGDP. NTF2 does not import RanGTP. Import can indeed be observed in the absence of RanGTP and NTF2 since correctly assembled import complexes can transport independent of

these molecules, and then get stuck in the nucleus by binding to nuclear macromolecules.

The images in the bottom row of Fig. 4c are so weak that I cannot interpret what is going on. These must be made brighter. My impression is that there is no improvement in import with KPNB1 and KPNA2; in fact, the IPO9 only data looks like more import. These data are not quantitatively compared. Due to the RanGTP and NTF2 issues discussed above, differences in the import efficiencies and import rates with the various importins could become more apparent if these are done properly. The import data as-is indicates inefficient transport and does not convincingly indicate what combination of the tested transport factors yields the best transport. I agree with the authors that this is not of central importance to their story as their gel data is convincing, but this is an opportunity missed, at the very least.

The images in Sup Fig. 8a,b are also very dim. These should also be made brighter and enlarged (fewer cells) so the reader does not have to strain to see what is going on. Use of the nucleoplasm/cytoplasm intensity ratio is problematic here. The 0,10 image in B clearly has dark nuclei, so the ratio cannot be 1 as indicated in C. Moreover, the proteosomes clearly bind very tightly to the cytoplasmic compartment, so the denominator signal is artificially high and dependent on bound transport factors (clearly observed in Fig. 4). This raises questions about the quantification in Sup Fig. 8c.

Minor points:

- 1) Note to authors: Sucrose does not adequately prevent nuclear swelling, as it is smaller than the size-exclusion limit of NPCs. This is information, it does not need to be addressed.
- 2) Sup Fig. 3f – the black fits to the dissociation data are very poor: straight lines fitting to obvious curves. The model is clearly wrong here.
- 3) Ref. 10 on line 55 does not appear to be correct.

Reviewer #4

(Remarks to the Author)

Revisions Point-by-point responses

Reviewer 1

1. Structural plasticity of Akirin2 (& IPO9) is interesting and may be crucial to its function in proteasome nuclear import. However, this aspect is not clearly explained. In their previous study (Almeida et al. Nature 2021), 76% of Akirin2 was unresolved. In the present study, what % of Akirin2 is resolved in binary & ternary structures (IPO9-Akirin2 & IPO9-Akirin2-proteasome complexes)? Also, are different regions of Akirin2 disordered vs. ordered between these 2 structures? Since certain arrangement/binding interface is stabilized in the ternary complex, detailed description of this aspect will be informative in understanding Akirin2 mechanisms.

In the new structure, we were able to resolve the full portion of the coiled-coil region of AKIRIN2 dimer I, corresponding to approximately 30% of the AKIRIN2 sequence, compared to 24% in the previous structure. Additionally, for one protomer, we were able to resolve the wing-helix domain, bringing the total resolved portion to approximately 44% of the AKIRIN2 sequence. The remaining parts of the protein are predicted to be intrinsically disordered and are therefore unlikely to be resolved by cryo-EM under the current experimental conditions. It is unclear if the wing-helix is stabilized in the complex with IPO9 or if the higher resolution compared to the previous structure 20S-AKIRIN2 (7NHT) made it possible to resolve this part. We agree that this was not described in enough detail in the manuscript, so we added the percentage of the maximum resolved protomer.

2. As for Akirin2 clustering on the proteasome, authors should validate the proposed stoichiometry: 4 Akirin2 dimers relative to one IPO9 or 20S proteasome. In Fig. 2C sedimentation data, this information is unclear.

We agree that determining the stoichiometric composition of the import complex would be ideal. However, AKIRIN2 is over 50% intrinsically disordered, and we found that the purified protein is prone to aggregation and surface adsorption. These properties have made several biochemical approaches, including those aimed at defining stoichiometry, technically challenging or infeasible. To approximate the increase in molecular mass associated with import complex formation, we performed size exclusion chromatography followed by mass photometry. These analyses revealed a stepwise increase in molecular weight in the following order: 20S < 20S:AKIRIN2 < 20S:AKIRIN2:IPO9 < 20S:AKIRIN2:KPNA2:KPNB1 < 20S:AKIRIN2:IPO9:KPNA2:KPNB1. In the presence of KPNA2 and KPNB1, we observed multiple distinct molecular weight species, indicating the formation of heterogeneous complexes, likely reflecting dynamic and variable stoichiometries. We added this data to the manuscript text and in Fig. 4f and Supplemental Fig. 8, and added a method description.

3. In addition to the schematics (Fig. 3J) depicting Akirin2 clustering on the proteasome, authors should include the actual cryo-EM data (several different views) to show the relative positioning of Akirin2 dimer I & II on the 20S. The following specific aspects need to be addressed.

We added the actual cryo-EM maps in Fig. 3a and g, and included another zoom view in Fig. 3d to show the interaction of the wing helix residue K94 with alpha3. For a better assessment and visualization we recommend the reader to look at the Supplementary Movie 1 and 2, as well as access the cryo-EM data with the deposited pdb/emdb codes.

- a. As key regions that mediate Akirin2 binding/clustering to the proteasome, authors should clearly describe where the *4 identical* SYVS motifs (of Akirin2 dimer I & II) individually dock onto *4 different* binding sites on the heteroheptameric alpha ring of the proteasome.

Also, where do 2 *identical* coiled-coil region & the wing helix binds to 2 *different* sites on the proteasome?

The binding mode of AKIRIN2 dimer I in our ternary complex (PDB: 9QOO) recapitulates the configuration observed in our previous structure (PDB: 7NHT). Within dimer I, protomer A (chain C in 9QOO) anchors its C-terminal SYVS motif into the binding pocket formed at the interface of alpha2/alpha4 subunits. Concurrently, protomer B of dimer I (chain B in 9QOO) engages its SYVS motif with the binding pocket between the alpha4/alpha7 subunits. The coiled-coil domain extends across the gate to the opposite side of the alpha ring. Furthermore, we have now resolved the wing helix domain of protomer A, which interacts with a negatively charged pocket on alpha3 via residue K94. In contrast, for AKIRIN2 dimer II we can not resolve an interaction of the SYVS motifs. Instead, our structural data reveal limited interaction interfaces: the wing helix of protomer C contacts the negatively charged surface of alpha6 that faces the exterior of the alpha ring, while a glutamic acid-rich patch in the coiled-coil region of protomer D interacts with the predicted nuclear localization signal (NLS) of alpha6. For a comprehensive visualization of these distinct binding modes, we have provided Supplementary Video 1, which systematically examines each interaction interface. Our structural and biochemical analyses suggest that while IPO9 stabilizes AKIRIN2 dimer II within the complex, the proteasome itself preferentially accommodates only one AKIRIN2 dimer possibly on both sides.

- b. Authors should reconcile how Akirin2 dimer I & II can be arranged in parallel in the ternary complex (Fig. 3J). Akirin2 C-termini would be at the periphery of 20S, based on the previous paper (Almeida et al. Nature 2021). However, the N-termini should point toward the central pore (gate) of the 20S alpha ring based on the previous study where coiled-coil interacts with alpha 1, 5, 6 gating termini. The N-termini of Akirin2 dimers would then be expected to converge toward the center, rather than assuming the depicted parallel arrangement.

The schematic in 3j should illustrate how the different binding interfaces on IPO9 are occupied by one AKIRIN2 dimer in the binary complex (right) in comparison to two AKIRIN2 dimers in the ternary complex (left). In this schematic look from the top on IPO9 which is projected to be flat. The two dimers indeed align in a parallel manner (to each other) in the ternary complex however AKIRIN2 dimer I crosses the alpha ring of the 20S in top view while dimer II is positioned on the outside of the alpha ring. To better illustrate the relative position we added a schematic for the 20S for the ternary complex in 3j (left). In our previous study we reported that the coiled-coil of AKIRIN2 stretches above the gate interacting with the N-termini of alpha 1, 5, 6 which form the gate. For clarification it's not the AKIRIN2 N-termini that interacts but the middle part of the coiled-coil. The arrangement of AKIRIN2 dimer I doesn't change in the new ternary complex.

- c. If alpha3 pocket is used to house the SYVS motif as described in the previous paper, authors should clarify how this same pocket is used for binding both SYVS motif as well as the critical lys94.

We admit that the nomenclature of the different proteasomal subunits leads to confusion. In the 2021 paper we referred to nomenclature that was introduced by Wolfgang Baumeister 1998 and is commonly used in the proteasome field. In the current manuscript we decided to use the Uniprot protein names to unify the description. We added here a translation table:

Uniprot gene name	Uniport protein name (current manuscript)	Baumeister et al. 1998 (Almeida et al. 2021)
PSMA1	$\alpha 1$	$\alpha 6$

PSMA2	$\alpha 2$	$\alpha 2$
PSMA3	$\alpha 3$	$\alpha 7$
PSMA4	$\alpha 4$	$\alpha 3$
PSMA5	$\alpha 5$	$\alpha 5$
PSMA6	$\alpha 6$	$\alpha 1$
PSMA7	$\alpha 7$	$\alpha 4$

In the proteasome alpha ring always two neighbouring alpha subunits form a binding pocket. The one between alpha2(PSMA2) and alpha4(PSMA4) as well as the one between alpha4 and alpha7(PSMA7) can each bind to one C-termini (SYVS) of the AKIRIN2 dimer. Lys94 of the wing helix of AKIRIN2 protomer A binds into a negatively binding pocket of alpha3(PSMA3). Importantly, this binding pocket forms within the subunit itself but is distinct to the binding pockets between individual alpha subunits. For better understanding we added the Uniprot gene names to the corresponding alpha subunits as they are unique in the manuscript.

- Line 189-190. The wing helix of the Akirin2 interacts with the negatively charged surface of alpha6 (fig. 3e). Which residues of alpha6 are involved in this interaction? Because the previous study showed that akirin2 coiled coil reaches out to alpha6 N-termini. Is it generally similar position of the alpha 6 N-termini?

The interaction of AKIRIN2 dimer II with the alpha6(PSMA6) negatively charged surface is different to the N-termini of the alpha6 that forms the gate of the proteasome. While the N-termini are facing the inside, the negatively charged surface is facing the outside (residues 207-244, in particular D209, E244), those two binding interfaces are about 46 Å apart from each other. We include additional Figure 1 for the reviewer that shows the distance length between the N-terminus and the second binding interface. Additionally, we edited Figure 3 and labeled the residues that form the negatively charged binding surface.

- Authors claim that IPO9 assumes the “open” state in the ternary complex (Fig. 3J). However, functional significance of the open vs. closed state is unclear. Authors should test this aspect by forcing the closed state (mutating the contact sites specific to the open state either on Akirin2 or IPO9). How does such disruption (keeping IPO9 in the closed state) influences Akirin2 clustering, proteasome nuclear import, or further recruitment of other importins?

Due to their HEAT repeat structure, importins are highly flexible proteins that are well known to adopt open and closed conformations in equilibrium (PMID: 23277578). Trapping these specific conformations has not been achieved outside of natural binders. Based on our structural analysis, we couldn't identify obvious strategies for achieving a forced close state through minimal interventions. Instead, we tried to mutate potential binding interfaces of IPO9 and assess if those mutations interfere with the proteasome import complex formation. We purified four single-point mutations that are located in one of the three binding interfaces (see Fig. 3j) with AKIRIN2. Complex formation was assessed by in-vitro reconstitution using size exclusion chromatography followed by mass photometry. We observed that the mutant E54K seems to be deficient in 20S:AKIRIN2 binding, while the other mutants behave similar to the wildtype and form higher molecular weight species (earlier fractions contain bigger complexes B2>B1>C1). In the B2 fraction complexes are up to 1334 kDa \pm 150 kDa big which would sum up to 1x20S (750 kDa), 8xGFP-AKIRIN2 (each 50 kDa), 2xIPO9 (each 120 kDa). Here we assume binding partially occurs on both sides of the proteasome. We included this data in

additional Figure 5 for the reviewers and included a method description in the additional information point 1. Unfortunately, we were unable to design a mutant that would only partially disrupt complex formation, which would prevent the second AKIRIN2 dimer from binding while maintaining overall complex stability, though it remains questionable whether such a partially disrupted complex would be stable.

6. Line 199-203. Authors should clarify whether “multiple importins” in this sentence refers to multiple IPO9s or other importins, which are described later in the paper. Based on the arrangement of IPO9-Akirin2-20S ternary complex, it is difficult to imagine how multiple IPO9s can be recruited by 2 Akirin2 dimers to the same side of the proteasome.

Here we changed the text to 'each of them exposing a bipartite NLS'. At this point we didn't introduce the other importins KPNA2/KPNB1 therefore we decided to not specify which importins might bind.

7. Is akirin2 clustering on the proteasome specific to the ternary complex or also observed on Akirin2-20S complex? In Extended Data Fig. 3e, f, authors describe that Akirin2-20S affinity is low as compared to Akirin2-IPO9 affinity. What happens to Akirin2-20S affinity when IPO9 is present?

More AKIRIN2 binds in the presence of IPO9 revealed by the cryo-EM structure. Mixture of IPO9 and AKIRIN2 is prone to precipitation which can for example be seen in gradients (Supplementary Fig. 3a,b). Here large amounts of AKIRIN2 (and IPO9) are found in the pellet fractions (21), suggesting oligomerization. We tried to measure KD values for proteasome-AKIRIN2 in presence of IPO9 unfortunately this wasn't successful due to AKIRIN2's characteristics.

Reviewer 2:

1. Importantly, it is not clear whether the described phenotypes in the study depend exclusively on a direct effect of AKIRIN2 on proteasome shuttling, or whether independent functions of AKIRIN2 also contribute. Are there alternative explanations for the observed effects due to AKIRIN2 loss-of-function? Preferably, a rescue-type experiment should be performed, in which restored proteasome localization rescues the effect of AKIRIN2 loss-of-function. The authors should at least adequately discuss this issue concerning inference of causality. Additionally, would it be feasible to fuse a nuclear export sequence to AKIRIN2 and look for its effect on proteasome localization and function?

We appreciate the reviewer's question regarding causality and alternative explanations for our observed phenotypes. Our comprehensive saturation mutagenesis screen employed three independent readouts including proteasome localization, MYC protein levels, and cell growth, all of which showed highly concordant results. AKIRIN2 mutants that disrupt proteasome nuclear localization consistently exhibit both elevated MYC levels and growth defects, while control variants that maintain proteasome import function do not exhibit these phenotypes. This extensive dataset suggests a direct causal relationship between AKIRIN2's proteasome import function and the observed cellular effects.

Our earlier work (PMID: 34711951) demonstrated that AKIRIN2 loss-of-function does not affect immediate transcriptional responses, which argues against indirect transcriptional effects as an alternative explanation. The tight correlation between proteasome mislocalization and substrate accumulation across multiple independent variants supports direct causality rather than pleiotropic effects from other AKIRIN2 functions.

Regarding rescue experiments, our saturation mutagenesis approach already provides the functional equivalent and goes far beyond what traditional rescue experiments could offer.

Testing numerous variants with different functional capacities provides more robust evidence for causality than relying on a single rescue construct.

The suggested AKIRIN2-NES fusion experiment would add complexity without providing meaningful additional insight. Of note already deletion of the NLS leads to cytoplasmic accumulation of AKIRIN2. Given that our current approach already suggests causality through multiple independent readouts across numerous variants, additional complex experimental manipulations are beyond the scope of our manuscript.

We believe our comprehensive approach provides definitive evidence that AKIRIN2's effects on cellular phenotypes are mediated through its role in proteasome nuclear import.

2. The introduction and discussion provide some examples on nuclear import of large protein complexes. However, there appears to be minimal mention of a number of pertinent previous studies on proteasome localization, besides references to some previous work done in yeast (Sts1). It would be appropriate to include a more detailed overview of key studies on proteasome localization in different model systems, which should indicate how AKIRIN2 compares to previously identified factors (e.g., Rpn2 bipartite NLS [PMID: 15210724]) and how the new findings fit into the larger picture. Are there different ways by which the proteasome can enter the nucleus?

We added a more detailed overview in the introduction about previous studies about proteasome import including rpn2 and HSP90 (Xenopus), Sts1 (Yeast), AKIR-1 (C.elegans). The mentioned study of Rpn2 bipartite NLS was focusing on yeast. While proteasome subunits are quite conserved the import machinery in yeast and higher eukaryotes seems to be quite different which can be probably partially explained by the different requirements in closed vs open mitosis. Further, from these studies there seems to be evidence that the 19S (probably also base and lid) are separately imported from the 20S. We strongly believe that AKIRIN2 is the major import factor for 20S proteasomes in mammalian cells. While our previous study suggests that also the import of 19S proteasomes strongly depends on AKIRIN2, it needs further investigations to dissect if 19S proteasomes are separately imported or associated with the core particle as 26S proteasomes.

3. The results and interpretation of the saturation mutagenesis screen should be discussed in more detail, i.e., how is a critical residue defined exactly and what are the key findings (specific mutation -> phenotypes)? Critically, the authors should perform validation experiments to show that the identified critical residues from their screen are indeed responsible for the observed phenotypes. Validation of these key AKIRIN2 mutations in independent experiments can be performed without reliance on reporters, e.g., by looking at endogenous MYC. To what extent do specific mutations phenocopy AKIRIN2 KO?

Critical residues are defined based on statistical thresholds across our three readouts, with specific cutoffs detailed in the methods section.

Regarding validation experiments, our study design already incorporates extensive validation through multiple independent approaches. Our saturation mutagenesis screen tested every possible single amino acid substitution across AKIRIN2, providing comprehensive validation of critical residues through three distinct readouts including proteasome localization, cell growth (dropout screen), and MYC reporter levels. Importantly, one of our readouts is entirely reporter-independent, eliminating concerns about reporter-specific artifacts.

We performed additional independent validation experiments for selected critical mutants as shown in Extended Data Figure 2, which demonstrates that individual mutations recapitulate the key phenotypes including defective proteasomal localization, elevated MYC levels, and altered protein expression. These validation experiments confirm that specific mutations

phenocopy AKIRIN2 knockout effects to varying degrees depending on the severity of the functional disruption.

The coherence across all three readouts provides robust validation that goes beyond traditional approaches. Rather than testing a handful of mutations in isolation, our comprehensive screen validates the functional importance of each residue position across the entire protein, with independent confirmation through multiple phenotypic measures. This approach provides more thorough validation than selective testing of individual mutants, as it demonstrates the consistency of structure-function relationships across the entire protein.

We added discussion of more specific examples of individual mutations and their quantitative effects across the different phenotypic readouts to better illustrate these relationships.

4. The authors focus on the 20S proteasome “to reduce the complexity”, however, the implications of this are inadequately discussed. Other forms of the proteasome may also be transported. What is known about the distributions, transport, and functions of 19S, 20S, and 26S proteasomes in nucleus vs. cytoplasm? AKIRIN2 binding likely prevents double capped proteasome from being translocated into the nucleus? The authors should comment on this in greater detail and discuss the limitations of their experiments and of their presented model, as well as how these issues can potentially be addressed in future work.

We agree that focusing on the 20S proteasome is a limitation of this study and appreciate the opportunity to discuss this more thoroughly. The nuclear transport and assembly of different proteasome forms remains incompletely understood, representing an important area for future investigation.

Our previous study demonstrated that 19S subunit nuclear import also depends on AKIRIN2, as AKIRIN2 knockout resulted in nuclear exclusion of tagged 19S subunits that were confirmed to be fully incorporated into functional 19S/26S complexes rather than orphaned. However, it remains unclear whether 19S subunits are imported independently or piggyback with 20S proteasomes. Studies in *Xenopus* suggest that 20S import occurs independently from most 19S components (Savulescu et al. 2011, PMID: 21289101), while yeast nuclear import adaptors like Sts1/Cut8 interact directly with 19S components, suggesting species-specific differences.

Our quantitative analysis of nuclear versus cytoplasmic distribution shows that 20S proteasomes are more than twice as abundant in the nucleus, while 19S subunits show only 30% nuclear enrichment. This differential distribution suggests that nuclear assembly and regulation of different proteasome forms may involve distinct mechanisms. Tomographic studies by the Baumeister lab suggest that nuclei also contain 30S proteasomes. However, how the equilibrium between different assemblies of the proteasome is regulated remains to be understood despite the decades of research.

The nucleus contains specific proteasome activators like PA200 and PA28 γ . Both contain their own NLS and are exclusively nuclear suggesting that they enter independently. Our current model based on 20S import provides a foundation for understanding AKIRIN2-mediated proteasome transport, but future work must address how different proteasome forms and activators are coordinately imported and assembled. The potential steric hindrance of double-capped proteasomes by AKIRIN2 binding represents another important question requiring investigation.

5. The cryoEM structures appear to be of good quality. Is there a potential role for other IPO9 adaptors? The authors highlight some specific features of the structures, e.g., relevance of residue K94 (e.g., its role in stabilizing the interaction with the proteasome), but it should be made clearer what is speculation based on the structure and what is supported by experiment.

Currently, no other IPO9 adaptor proteins have been identified in the literature, and AKIRIN2 appears to be the primary adaptor mediating proteasome nuclear import. Our nuclear import assays demonstrate that AKIRIN2 together with IPO9 and RanGTP are sufficient for

proteasome nuclear import, indicating that additional adaptors are not required for this process. Our comprehensive screen also did not reveal evidence for other essential factors in this pathway.

Regarding the structural interpretations, we will clarify what represents experimental validation versus structural speculation. The importance of residue K94 and in fact all highlighted residues is supported by experimental evidence from our saturation mutagenesis screen, where mutations at this position consistently disrupted proteasome import function across all three readouts. The structural analysis provides the molecular basis for this functional requirement, showing that K94 likely stabilizes the AKIRIN2-proteasome interaction through specific contacts visible in our cryoEM structure. We have included a detailed zoom view highlighting these interactions to illustrate the structural basis for K94's functional importance.

6. The section “Conformational plasticity underpins AKIRIN2-mediated complex assembly” describes the structure of the AKIRIN2-IPO9 binary complex, stating “this structural plasticity illuminates a dynamic assembly mechanism for large nuclear import complexes”. However, if and how this pertains to proteasome import is unclear. Is this structure in absence of proteasome likely to be relevant, considering the variety of states the proteasome can adopt itself? It is not clear to the reviewer what is to be learned from the AKIRIN2-IPO9 binary complex structure.

The question about the relevance of the AKIRIN2-IPO9 binary complex structure is an important point to address. This structure is crucial for understanding the complete proteasome import mechanism.

The binary complex reveals the conformational flexibility of IPO9 when bound to AKIRIN2 alone, which contrasts with the constrained arrangement in the ternary complex. This demonstrates that proteasome binding requires specific conformational selection in the IPO9-AKIRIN2 interface essential for cooperative assembly. The transition from open to closed conformations is a common regulatory mechanism in importins and necessary for understanding the complete import cycle.

The binary complex is thermodynamically more stable than the AKIRIN2-proteasome complex, and given that AKIRIN2 concentrations exceed those of proteasomes during import, this binary state may be the most prevalent form in cells. Most importantly, the binary structure elucidates how IPO9 accommodates different AKIRIN2 stoichiometries. While the binary complex shows IPO9 interacting with AKIRIN2 homodimers through one interface set, the ternary complex reveals how conformational rearrangements allow two AKIRIN2 dimers to bind simultaneously through distinct configurations.

This structural plasticity is fundamental to understanding how multiple importin complexes are recruited to proteasome import complexes. The binary structure provides essential mechanistic insight into the assembly pathway and reveals the conformational flexibility enabling cooperative binding in the functional import complex.

7. It is stated that AKIRIN2 itself is rapidly degraded by nuclear proteasomes in a ubiquitin-independent manner, based on experiments with inhibitors. Is this degradation of AKIRIN2 19S or 20S-dependent?

Our data strongly suggests that AKIRIN2 degradation is predominantly 20S proteasome-dependent through a ubiquitin-independent mechanism. Evidence supports this conclusion: (1) AKIRIN2 itself is highly turned over by the proteasome which we have seen in our previous study by knocking out a proteasomal subunit, it scored amongst the most enriched proteins. (2) AKIRIN2 contains substantial intrinsically disordered regions, a recognized feature of proteins directly degraded by the 20S core particle without ubiquitination (PMID: 25250704).

(3) Our experiments with the E1 inhibitor TAK243 demonstrate that AKIRIN2 degradation occurs independently of ubiquitination.

(4) In preliminary in vitro degradation assays (not included in the manuscript), we observed that purified 20S proteasomes generated characteristic GFP-AKIRIN2 degradation products (AKIRIN2 degradation with stable GFP remaining), while 26S proteasomes did not produce these fragments. We included this data for the reviewer (additional Figure 2) as well as a method description (additional information point 2).

While we cannot completely exclude some contribution of the 19S regulatory particle through recently described ubiquitin-independent degradation pathways of the 26S proteasomes (PMID: 37616343), our combined data points to AKIRIN2 being primarily a direct 20S substrate. We believe the mechanism of AKIRIN2 degradation merits dedicated investigation beyond the scope of this study, which focuses on its functional role rather than its turnover mechanisms.

8. Like most good studies, the work also raises many new questions. The authors provide some statements in the discussion that deserve more scrutiny. For example, a potential role for Midnolin as a trigger in AKIRIN2 degradation is mentioned, however, this appears to be not supported by any data. If this idea is correct, it should be relatively easy to test. Is Midnolin-dependent degradation relevant in the cell types used in this study, i.e., is it a plausible mechanism? Further, the authors mention proteins with analogous functions to AKIRIN2 for other large complexes, and describe a common architecture for these adaptor proteins, by which they could perform multiple critical functions. While interesting conceptually, it may be good to include a sentence on what remains to be demonstrated experimentally for these particular examples.

We agree that the potential role of Midnolin is purely speculative and rather unlikely, as it lacks the ubiquitous expression pattern of AKIRIN2. We have therefore softened our statement regarding Midnolin-dependent degradation to: 'Whether additional components similar to the recently discovered protein Midnolin are required to target nuclear AKIRIN2 for nuclear degradation remains to be investigated.'

In the discussion point about nuclear import adaptors, we added a sentence of the remaining questions: 'It remains to be determined whether this proposed common architecture is essential for a nuclear import adaptor function and whether the constituent building blocks are interchangeable among different adaptors.'

Reviewer 3:

- I. There are numerous places within the figures and captions where it should be made clearer what is being shown so that it is easier to understand the experiment for a non-expert, particularly with regards to abbreviations.

We tried to make the captions clearer, we included a description for the abbreviations (d)sgRNA, IBB, FC, MST, GCI, NGS. Please let us know if there are more that are unclear.

- II. "The nuclear import assays in Fig. 4 should be improved/clarified.
The imaging timepoint is not stated. The Methods say 30-60 min, but this should really be done with a kinetic curve to clarify the effects of the different transport factors. Is there any reason this cannot be done by collecting an image every minute or so? The cells are washed and it takes about 15 min before they image, during which material could leak out (if they haven't confirmed that it does not).
The amount of import should be quantified for all conditions.
The conditions of Fig. 4d (i.e., the transport factors used) are not stated. This shows only a 2-fold increase with a very high baseline import.

A few changes could increase import efficiency. An ATP regenerating system works in the permeabilized cell transport assay, but GTP is better as Ran is a GTPase. Ran is typically purified in both GDP and GTP forms, which can be separated on an anion exchange column. It is not clear if they did this. Cytoplasmic RanGTP would dissociate import complexes. NTF2 should be added to shuttle Ran into the nucleus."

We performed a time-course experiment (0, 5, 15, 30, 45, 60 min), which demonstrated that nuclear import plateaus after approximately 45 min, with no further increase observed at 60 min. Representative images are shown in Supplementary Fig. 8a, and the corresponding quantification is presented in Fig. 4d. For all other experiments, we used a 60 min incubation time unless stated otherwise. Timepoints are now clearly indicated in each experiment.

Due to the requirement of a wash step prior to imaging, necessary to reduce background fluorescence from unbound fluorophores, it is unfortunately not feasible to image the same sample at multiple timepoints within the same experiment.

We further explored the effect of increasing AKIRIN2 concentrations on nuclear import. Instead of using a defined molar excess over the proteasome, we performed a titration across a broader range, extending above the measured dissociation constant (KD) of 5 μ M. As shown in the new supplementary data (Supplementary Fig. 8b,c and additional Fig. 4a). AKIRIN2 displays a concentration-dependent shift from predominantly cytoplasmic to nuclear localization with concentration ≥ 5 μ M promoting robust nuclear import. Interestingly, at very high concentrations (50 μ M), nuclear import is impaired, which may reflect oversaturation of binding interfaces. We note this observation and plan to investigate the underlying mechanism in future work.

We have clarified the composition of the importin complex and other relevant experimental details in the legend for Fig. 4d. The ATP regeneration system is commonly used in nuclear import assays to support GTP recharging (via NDPK-mediated conversion of GDP to GTP). While we agree that including a Ran recycling system better reflects physiological conditions, we tested nuclear import efficiency upon addition of NTF2 and RanBP1 in our system but observed no appreciable improvement. Given that the assay already requires up to ten different purified components, we opted to omit these factors in subsequent experiments to maintain a simplified system. For completeness, we have now included microscopy images comparing nuclear import in the absence and presence of NTF2 for the reviewer (additional Fig. 4b).

- III. The model presented is quite interesting, and logically follows from their data. However, I wonder about stability in the complex environment of the NPC permeability barrier. The IPO9 forms an open structure when attached to proteasomes, but it does appear fairly firmly attached to AKIRIN2. However, the KPNB1/KPNA2 heterodimers are predicted to be up to 18 nm away on flexible tethers. It seems like there could be a lot of strain on this structure as it is migrating through the pore and all the intervening space is filled by FG-polypeptides. Is there any evidence that the KPNB1/KPNA2 heterodimers have additional attachments to the proteasome?

Yes, the structural stability of the complex during nuclear transport is of great interest. We believe there may be a misunderstanding regarding our structural model that we can clarify. The 18 nm measurement represents the maximum theoretical extension of AKIRIN2's flexible N-terminal regions, not the actual working distance during transport. This measurement was intended to illustrate that sufficient spatial flexibility exists for multiple AKIRIN2 protomers to simultaneously engage KPNA2/KPNB1 heterodimers without steric hindrance.

During actual nuclear transport, the flexible regions of AKIRIN2 would not need to be fully extended and would likely adopt more compact conformations that minimize surface area while maintaining functional contacts. The FG-rich environment of the nuclear pore would likely

support rather than destabilize this arrangement through entropic effects. Additionally we performed gradient centrifugation on the complexes with more than 100g of acceleration introducing significant higher forces than facilitated diffusion could. In these experiments the complexes stay together which suggests that force is not a concern.

Regarding additional KPNB1/KPNA2 binding sites on the proteasome itself, our current structural and biochemical data provide no evidence for direct importin-proteasome contacts beyond those mediated by AKIRIN2. However, we acknowledge that our analysis cannot definitively rule out weak or transient additional interactions that might contribute to complex stability during transport. Literature suggests that there are more NLS sequences in the proteasome but they are either already occupied by features of our import complex or not accessible.

We revised the manuscript to clarify that the 18 nm measurement represents theoretical maximum flexibility rather than the actual working geometry, and emphasize that the cooperative importin binding likely enhances rather than compromises transport complex stability.

1. q line 36 – ‘cyto- and nucleoplasm’ is a strange construction. Just expand ‘cyto-’ to cytoplasm.

We edited this in the manuscript.

2. lines 43 and 329 – Tu et al (EMBO J. 2013 32:3220) showed the effects of import receptor number on nuclear transport of a large protein cargo.

We edited this in the manuscript.

3. line 79 – at first mention, more clarity on the mCherry-MYC fusion protein is needed, i.e., that MYC is a transcription factor degraded by nuclear proteasomes (and not simply a common epitope tag).

We edited this in the manuscript.

4. lines 98-99 – this statement should be clarified as it does not appear to be true based on Fig. 1c and ED Fig. 1c,d.

If the reviewer is referring to the outlier at residue 70 where the STOP codon does not appear to be dysfunctional, this is due to a technical artifact. Specifically, during pooled cloning, some constructs may be lost. This particular construct has less than 5% of the median reads, and therefore it does not pass filters and gets no meaningful score. We now account for such technical limitations by stating 'generally' instead of 'all' in the text.

5. lines 288-290 – it's possible that an IPO9 recycling factor is needed for RanGTP dependent release.

We are also quite interested in this observation. We are certain that IPO9 must be fully released from the proteasome when arrived in the nucleus. Unfortunately, we couldn't determine the condition or additional factor required for the full release of IPO9. But similar behavior for IPO9 was observed before in the H2A/H2B import study and suggests that RanGTP binding changes the conformation of IPO9 preparing the cargo release.

6. Fig. 1c,d – (c) what dataset is this? Clarify as was done for ED Fig. 1c,d. (d) The caption identifies colored boxes as ‘squares’ when these are in fact rectangles. Can (c) and (d) be combined or aligned better so they match up exactly?

The dataset in 1c) is the cumulative analysis meaning merged data from all three datasets. In ED fig 1c,d individual plots are shown.

We changed the caption squares to rectangles.

We aligned 1d more precisely to 1c for better comparison.

7. "Fig. 2 – Do the StayGold and mCherry images in (a) correspond to the same cells?
Yes, in a) StayGold and mCherry images are the individual channels of the same cells.

What is AAVS1?

AAVS1 is a non-essential gene locus used as a CRISPR-Cas9 KO control (cuts the genome in a DOX inducible manner but shouldn't affect the reporter).

What are dashed lines and FC in (b)?

Dashed line indicates an arbitrarily chosen but commonly used significance and effect size cut-off based on p-value and fold change of IPO9 hits. FC means fold-change and was edited for clarification in the figure.

How are proteins identified in (c)?

Is answered together with the question: 'How were proteins identified in the top gel if this is stain free?'

The cartoon for Akirin2 suggests a fragment rather than the whole protein. Purification of a fragment was not found in the Methods.

We chose a simple cartoon for the AKIRIN2 dimer based on the coiled-coil we observe in our structure. While using this cartoon we refer to the whole protein not to a fragment, only in Fig.4a where we deleted the NLS of AKIRIN2, we modified this cartoon slightly to highlight the differences, otherwise we always used the full-length GFP-tagged version of AKIRIN2 if not specifically mentioned otherwise.

Single load lanes should be shown somewhere.

We included a figure showing a SDS-Page gel of all purified proteins individually. For AKIRIN2, we loaded a sample where only SDS sample buffer was added as well as a fully denatured sample (treated with heat before loading). One can observe that in the first sample, two bands are detected by MBS blue staining where one which is not fully denatured is still detectable by fluorescence (GFP) and is visible in the stainfree imaging technology of BIORAD. We include this data in additional figure 3 for the reviewer.

How were proteins identified in the top gel if this is stain free?

All proteins were individually purified and analysed by SDS PAGE and by mass spectrometry. Therefore, we could simply differentiate between the individual proteins by their molecular weight in the stainfree SDS-PAGE gels imaged in a stainfree imaging system by BIORAD after running sucrose density gradients. In the same gels we imaged the fluorescence of GFP-AKIRIN2 which is sometimes hardly visible in the stainfree gels, we will add corresponding labeling

I didn't figure out until the end of the paper that the top and bottom gels were different (one is fluorescence) – this needs to be indicated more prominently.

The lower images are the GFP signal of the AKIRIN2 construct. To make this more clear we adjusted the figure caption.

There is no Fig. 2d (line 158)."

Mislabeling Fig. 2d was edited to Fig. 2c.

8. Fig. 4 – (c) what is the imaging timepoint? (d) is the excess a ‘molar excess’? So, the maximum extra import of proteasome import is two-fold? This doesn’t sound like much. Is this a timepoint or kinetic constant?

We added the exact imaging time point. We further added an experiment titrating AKIRIN2 concentrations (0.1 -50 μ M). We can only image at a particular time point in the case of Fig. 4c after 1 h of incubation, followed by a wash step. We included an experiment where we tested different incubation times (0,5,15,30,45 and 60 min) and included the microscopy images for the reviewer.

9. lines 323-326 – These two sentences are confusing and should be re-written.

We edited the respective paragraph.

10. ED Fig. 1b – show StayGold and mCherry images separately.

We added the individual images of the StayGold and the mCherry channel.

11. ED Fig 2a – I presume that these are the types of images evaluated for their three screens. It would be helpful to provide a bit more information indicating this.

We edited our manuscript accordingly by adding: To validate these findings, we individually tested dysfunctional variants by introducing single point mutations into our reporter cell line. We assessed their functional impact by imaging proteasome subcellular localization and mCherry-myc reporter signals with confocal microscopy, and western blot analysis to quantify AKIRIN2 protein levels.

12. ED Fig. 3 – (a) KPNB1 and KPNA2 were not indicated in the caption. (d) ‘cooked’ in the caption is not a commonly used expression, so some explanation is needed. This sentence also has two consecutive ‘which’ phrases, an awkward construction. (e) and (f) what are MST and GCI? The number format 40 [10,100] is unclear. k_a and k_d do not have units. It is unclear what is shown in the (f) panel (what is fit and what is data?) – for dissociation, the fitted curves don’t match well with experiment.

We added KPNA2/KPNB1 in the caption.

We changed ‘cooked’ to ‘fully denatured’. Text edit: *AKIRIN2 shows a double band if the sample is not fully denatured before loading on SDS-PAGE which allows imaging of GFP in the SDS-PAGE gel.

We included the full description Microscale Thermophoresis assays for MST and Grating-coupled interferometry for GCI. For measurements with small replicates it is common and more accurate to display the range instead of the standard deviation which requires the assumption of a normal distribution.

13. ED Fig. 7b – AF2 should be spelled out.

We edited this in the manuscript.

Additional information for the reviewers

1. IPO9 mutants

To determine whether disruption of a single binding interface between IPO9 and AKIRIN2 interferes with proteasome import complex formation, we introduced the following point mutations in IPO9: V49E, E54K, K661E, and R787E. Mutant and wild-type IPO9 proteins were individually expressed, purified, and assembled into complexes with 20S proteasome and

AKIRIN2. Complex assembly reactions were carried out in buffer MP (50 mM HEPES pH 7.5, 50 mM NaCl, 0.5 mM TCEP) using the following input protein amounts: 100 pmol 20S proteasome, 1000 pmol AKIRIN2, and 500 pmol IPO9. Complexes were analyzed by size-exclusion chromatography on a Superose 6 Increase 3.2/300 column (Cytiva) pre-equilibrated with buffer MP. Elution was performed at 4 °C and 50 µL fractions were collected. Peak fractions were analyzed by SDS-PAGE. In addition, fractions B2, B1, and C1 were selected for mass photometry measurements as described in the Methods section “Stoichiometry estimation by chromatography and mass photometry.”

2. In-vitro degradation assay 20S proteasomes vs. 26S proteasomes

Degradation reactions were carried out as 100 µL reaction in activity buffer (25 mM Tris pH 7.5, 5 mM MgCl₂, 5% glycerol, 4 mM ATP, 1 mM DTT, 1 mM GTP). The following concentrations were used: 25 nM 20S or 26S proteasomes, 50 nM IPO9, 250 nM RanQ69L, 0.5 mM MG132, 100 nM GFP-AKIRIN2. To reduce unspecific binding, reaction tubes were blocked for 30 min with 100 µL blocking buffer (2 mg/mL BSA in activity buffer) prior sample preparation. All components were assembled on ice, with GFP-AKIRIN2 added last. Samples were mixed immediately, and an aliquot was taken as the 0 min time point. Reactions were incubated at 37 °C, and additional aliquots were collected at multiple time points (5-60 min), then quenched by the addition of SDS loading buffer. Samples were resolved on 4-20% gradient SDS-PAGE gels (Bio-Rad) run at 200 V for 40 min. Gels were imaged using a Bio-Rad ChemiDoc system with stain-free detection and GFP fluorescence channels.

Additional figure 1: reviewer 1 - comment 4
Distance of proteasomal subunit 6 N-terminal and C-terminal binding interfaces

Additional figure 2: reviewer 2 - comment 7
 Degradation assays of GFP-AKIRIN2 by 20S and 26S proteasomes

Additional figure 3: reviewer 3 - comment 7
Single load lines of the individually purified proteins used for reconstitutions

Additional figure 4: reviewer 3 - comment 0b
 Nuclear import assays - AKIRIN2 titration

a

b

Additional figure 5: reviewer 1 - comment 5
IPO9 mutants

a

b

c

Point-to-point reply

We would like to thank all reviewers for again revising our manuscript and for their constructive feedback.

REVIEWER COMMENTS

Reviewer #1 (Remarks to the Author):

The authors appropriately addressed all of my comments both in the text and in the figures. I am pleased with the revised manuscript and fully support the publication of this manuscript.

Reply: We thank the reviewer for thoroughly revising our manuscript and their comments.

Reviewer #2 (Remarks to the Author):

The manuscript has been significantly improved. However, there are still some important remaining questions.

1. **Previous comment:** Importantly, it is not clear whether the described phenotypes in the study depend exclusively on a direct effect of AKIRIN2 on proteasome shuttling, or whether independent functions of AKIRIN2 also contribute. Are there alternative explanations for the observed effects due to AKIRIN2 loss-of-function? Preferably, a rescue-type experiment should be performed, in which restored proteasome localization rescues the effect of AKIRIN2 loss-of-function. The authors should at least adequately discuss this issue concerning inference of causality. Additionally, would it be feasible to fuse a nuclear export sequence to AKIRIN2 and look for its effect on proteasome localization and function?

Reply: We appreciate the reviewer's question regarding causality and alternative explanations for our observed phenotypes. Our comprehensive saturation mutagenesis screen employed three independent readouts including proteasome localization, MYC protein levels, and cell growth, all of which showed highly concordant results. AKIRIN2 mutants that disrupt proteasome nuclear localization consistently exhibit both elevated MYC levels and growth defects, while control variants that maintain proteasome import function do not exhibit these phenotypes. This extensive dataset suggests a direct causal relationship between AKIRIN2's proteasome import function and the observed cellular effects. Our earlier work (PMID: 34711951) demonstrated that AKIRIN2 loss-of-function does not affect immediate transcriptional responses, which argues against indirect transcriptional effects as an alternative explanation. The tight correlation between proteasome mislocalization and substrate accumulation across multiple independent variants supports direct causality rather than pleiotropic effects from other AKIRIN2 functions. Regarding rescue experiments, our saturation mutagenesis approach already provides the functional equivalent and goes far beyond what traditional rescue experiments could offer. Testing numerous variants with different functional capacities provides more robust evidence for causality than relying on a single rescue construct. The suggested AKIRIN2-NES fusion experiment would add complexity without providing meaningful additional insight. Of note already deletion of the NLS leads to cytoplasmic accumulation of AKIRIN2. Given that our current approach already suggests causality through multiple independent readouts across numerous variants, additional complex experimental manipulations are beyond the scope of our manuscript. We believe our comprehensive approach provides definitive evidence that AKIRIN2's effects on cellular phenotypes are mediated through its role in proteasome nuclear import.

Comment: Their arguments are received. Note that, the observation that AKIRIN2 mutations have similar effects on proteasome localization and on MYC level is consistent with but does not directly support a “causal relationship between AKIRIN2's proteasome import function and the observed cellular effects”, because their current results do not exclude the possibility that AKIRIN2 may serve as an import adaptor for other key factors in the nucleus (there are actually a few AKIRIN2-interacting

proteins essential in the nucleus according to Biogrid) or other import-independent essential functions. Most mutations may perturb the fold or stability of AKIRIN2, leading to general LOF. Adding to this suspicion is the fact that AKIRIN2 is codependent with factors in various biological functions on Depmap including ribosome process, DNA integrity etc. Although a few proteasome genes are codependent with AKIRIN2, most are the 19S subunits, inconsistent with their proposed mechanism. A rescue experiment would be perhaps the simplest way to complete their argument.

Reply: We agree that one must be careful comparing proteasome nuclear localization and MYC levels to dissect the effects of AKIRIN2 mutants, however all of our data supports that the upregulation of MYC as well as other nuclear substrates can be attributed to lack of nuclear proteasomes which target them for degradation. In our previous manuscript, we showed that almost all proteins that enrich after the knockout of AKIRIN2 are also stabilized upon knockout of the proteasomal subunit PSMA3 (Fig. 3a,b). Gene ontology analysis revealed that all proteasome target proteins that are also enriched upon AKIRIN2 knockout were enriched for nuclear localization (Fig. 3d). Additionally, we could show that the knockout of AKIRIN2 does not have a strong transcriptional effect (Extended Data Fig. 4a-c). As well as with a cycloheximide chase experiment that MYC protein levels are stabilized upon AKIRIN2 knockout (Extended Data Fig. 4d). We included here Figure 3 and Extended Data Fig. 4 from Almeida et al., 2021).

Fig. 3: a, b, Whole proteome changes after induced AKIRIN2 (a) or PSMA3 (b) knockout. Quantitative mass spectrometry was performed 2 d (sgAKIRIN2 and sgAAVS1) or 3 d (sgPSMA3) after Cas9 induction. Genes with significant upregulation of protein (adjusted $P \leq 0.01$ and $FC \geq 1.5$, limma-moderated Benjamini-Hochberg-corrected two-sided t -test, $n = 2$ biological replicates) but not mRNA levels were classified as AKIRIN2 targets ($n = 124$, red box in a, highlighted red in b) or proteasome targets ($n = 289$, blue box and dots), respectively. c, Immunoblot of MYC and AKIRIN2 after acute AKIRIN2 or PSMA3 knockout. Data are representative of two independent experiments. Time points as in a and b. Uncropped scans are provided in Supplementary Fig. 1. d, Top differentially enriched GO terms (Δ FDR, Benjamini-Hochberg-corrected two-tailed Fisher's exact test) between AKIRIN2-responsive ($n = 89$) and AKIRIN2-independent proteasome targets ($n = 153$). Complete results are shown in Supplementary Table 4. BP, biological process; CC, cellular component; FC, fold change; MF, molecular function.

Extended Data Fig. 4: a–c, Transcriptional changes after acute knockout of AKIRIN2 (**a**) or PSMA3 (**b**). RNA-seq of iCas9-RKO cells was performed 2 (sgAKIRIN2, sgAAVS1) or 3 days (sgPSMA3) after Cas9 induction (n = 3 biological replicates). Genes significantly up- or downregulated ($P \leq 0.01$; Benjamini-Hochberg corrected two-sided Wald-test) at least two-fold are highlighted in orange, TP53 target genes according to 20 in red. **c**, Principal component (PC) analysis of the 1000 most highly expressed genes. **d, e**, AKIRIN2 and MYC protein half-life quantification. Immunoblot time-series of iCas9-RKO cells treated with cycloheximide (CHX) 2 days after Cas9 induction (**d**) and half-life quantification (**e**) of AKIRIN2 (half-life = 46 min, 95% CI = 37–58 min) and MYC (half-life = 25 and 178 min with 95% CI = 21–31 and 51–238 min in sgAAVS1 control and sgAKIRIN2 cells, respectively). Data is shown as mean \pm s.d. (n \geq 3 independent experiments). Dashed lines indicate half-lives. **f, g**, Quantitative proteomics following induced AKIRIN2 or PSMA3 knockout. Samples were obtained as described for a-c (n = 2 biological replicates). **f**, Principal component analysis of the 1000 most highly expressed proteins. **g**, Scatter plot of transcriptome- versus proteome-changes upon acute AKIRIN2 knockout compared to sgAAVS1 control. AKIRIN2 targets as in Fig. 3a, b (orange; n = 124) are upregulated only on protein-, but not on mRNA-level. TP53 target genes as in **a, b** are shown in red. **h**, Western blot of selected AKIRIN2 targets after AKIRIN2 or proteasome knockout. iCas9-RKO cells expressing the indicated sgRNAs were harvested before, and 2 and 3 days after Cas9 induction. **i**, Euler diagram of proteasome targets and AKIRIN2 targets as defined in Fig. 3a, b. CI, confidence interval; FC, fold change.

We carefully examined reported BioGRID partners and DepMap co-dependencies. While we think that low genetic co-dependency between AKIRIN2 and the proteasome does not contradict our model.

The proteasome is one of the most prominent protein complexes in the cell with various functions across several cellular programs, it would be even surprising if there is a genetic co-dependency on a localisation regulator. Of note, checking across various known proteasomal interactors such as PSME3, PSMF1, or ECM29 no significant co-dependencies can be found in Depmap.

However, we think that the BioGRID data with more than 50% of the reported interactors belonging to the proteasome and its import machinery rather supports our data. Of note, in our previous study (Almeida et al., 2021) we performed a genome-wide knockout screen to identify MYC regulators, and none of the BioGRID AKIRIN2 interactors besides the proteasome and the E3 ligase UBR5 showed significant effect on MYC levels.

Further, regarding other import cargos, we found in our prior co-IP/MS (Almeida et al., 2021) proteasome subunits, most prominently 20S core subunits, were the top AKIRIN2 interactors. Deleting the short C-terminal proteasome-binding motif (YVS; last three residues) abolished proteasome co-purification while leaving other detected interactors unchanged in our hands. We note that the Δ YVS mutant that disrupt proteasome binding do so without detectable effects on AKIRIN2 protein stability or abundance. This data again strongly supports a primary role in proteasome import underlying the observed MYC changes.

We don't think that most mutations perturb the fold which can be argued first by AKIRIN2's high degree of disorder, and most mutations in the structured region are functional, besides the exchange to proline which has indeed the characteristic to disrupt helical structures. Further, most mutations that have an effect show clear stripe patterns which means that they are dysfunctional, if mutated to most or all other amino acids, this clearly speaks for a lost interaction rather than a folding or stability problem, which can also be seen in some of our validation mutants (Suppl. fig. 2).

Unfortunately, we are not sure what kind of rescue experiment the reviewer asks for. If we understand the reviewer correctly, we should find a way to restore nuclear proteasomes in the absence of AKIRIN2? We are afraid there is not a trivial way to do this. Even if we could add an artificial NLS to each proteasomal subunit it's the question if there is enough space to bind to several importins.

Nevertheless, we believe that our findings nicely and coherently show that the cell developed a complex mechanism to efficiently import proteasomes.

Comment: In addition, there appears some discrepancy between MYC stability and proteasome localization (supp. fig 1. c/d). Are those due to technical reasons or have biological underpinnings? It would be useful to have e.g. a scatter plot to correlate these two effects and highlight the outliers. Could they also indicate the functionally important sites on the ternary structure?

Reply: We strongly believe that the minimal discrepancies between the individual readouts are due to technical reasons. Importantly to mention here is that especially those residues that show a stripe pattern (meaning not a single amino acid swap is dysfunctional but the change to several amino acid that have similar characteristics e.g. polar and/or charged) can be interpreted as the most meaningful.

We added here and, in the manuscript (Suppl. Fig. 1e), a scatter plot which shows the correlation between the two screens, which shows that there are very few outliers.

Additional fig.: Scatter blot comparing effect of the myc and the PSMB4 localisation screen, correlation coefficient 0.73.

We tried to map the functional important residues on the ternary structure however, we can only visualize the structured parts of AKIRIN2 which is 30-44% of its sequence and since several promoters are present in the structure the visualization in an image is not well representative that's why we included movie 1 which zooms in the individual binding interfaces and highlights the functionally important residues in the structured parts.

2. **Previous comment:** The introduction and discussion provide some examples on nuclear import of large protein complexes. However, there appears to be minimal mention of a number of pertinent previous studies on proteasome localization, besides references to some previous work done in yeast (Sts1). It would be appropriate to include a more detailed overview of key studies on proteasome localization in different model systems, which should indicate how AKIRIN2 compares to previously identified factors (e.g. Rpn2 bipartite NLS [PMID: 15210724]) and how the new findings fit into the larger picture. Are there different ways by which the proteasome can enter the nucleus?

Reply: We added a more detailed overview in the introduction about previous studies about proteasome import including *rpn2* and *HSP90* (*Xenopus*), *Sts1* (Yeast), *AKIR-1* (*C.elegans*). The mentioned study of *Rpn2* bipartite NLS was focusing on yeast. While proteasome subunits are quite conserved the import machinery in yeast and higher eukaryotes seems to be quite different which can be probably partially explained by the different requirements in closed vs open mitosis. Further, from these studies there seems to be evidence that the 19S (probably also base and lid) are separately imported from the 20S. We strongly believe that AKIRIN2 is the major import factor for 20S proteasomes in mammalian cells. While our previous study suggests that also the import of 19S proteasomes strongly depends on AKIRIN2, it needs further investigations to dissect if 19S proteasomes are separately imported or associated with the core particle as 26S proteasomes.

Comment: A more detailed overview of background studies has been included. The authors state that differences in import machinery between yeast and higher eukaryotes may be partially explained by open vs. closed mitosis. Is data available to back this idea for proteasome specifically? The basis of this difference seems mechanistically important and it would be worthwhile to discuss this in the paper.

Reply: We agree that it would be highly interesting to understand why distinct pathways of proteasome nuclear import have evolved in yeast versus higher eukaryotes. However, in most metazoans, including all vertebrates, no *Sts1* homolog exists, and conversely, no AKIRIN2 homolog is found in yeast with only a single Fungi species (*Jimgerdemannia flammicorona*) having an AKIRIN2 homolog but its mitosis status is unknown. Interestingly, *Drosophila* has both STS1 and a AKIRIN2 homolog and is known to have semi open-mitosis forming thus a consistent intermediate. While nuclear import of the proteasome has been well studied in yeast, little was known in higher eukaryotes, which was the focus of our study. Directly comparing yeast and human pathways is therefore beyond the scope of our work.

3. **Previous comment:** The results and interpretation of the saturation mutagenesis screen should be discussed in more detail, i.e., how is a critical residue defined exactly and what are the key findings (specific mutation -> phenotypes)? Critically, the authors should perform validation experiments to show that the identified critical residues from their screen are indeed responsible for the observed phenotypes. Validation of these key AKIRIN2 mutations in independent experiments can be performed without reliance on reporters, e.g., by looking at endogenous MYC. To what extent do specific mutations phenocopy AKIRIN2 KO?

Reply: Critical residues are defined based on statistical thresholds across our three readouts, with specific cutoffs detailed in the methods section.

Regarding validation experiments, our study design already incorporates extensive validation through multiple independent approaches. Our saturation mutagenesis screen tested every possible single amino acid substitution across AKIRIN2, providing comprehensive validation of critical residues through three distinct readouts including proteasome localization, cell growth (dropout screen), and MYC reporter levels. Importantly, one of our readouts is entirely reporter-independent, eliminating concerns about reporter-specific artifacts.

We performed additional independent validation experiments for selected critical mutants as shown in Extended Data Figure 2, which demonstrates that individual mutations recapitulate the key phenotypes

including defective proteasomal localization, elevated MYC levels, and altered protein expression. These validation experiments confirm that specific mutations

phenocopy AKIRIN2 knockout effects to varying degrees depending on the severity of the functional disruption.

The coherence across all three readouts provides robust validation that goes beyond traditional approaches. Rather than testing a handful of mutations in isolation, our comprehensive screen validates the functional importance of each residue position across the entire protein, with independent confirmation through multiple phenotypic measures. This approach provides more thorough validation than selective testing of individual mutants, as it demonstrates the consistency of structure-function relationships across the entire protein.

We added discussion of more specific examples of individual mutations and their quantitative effects across the different phenotypic readouts to better illustrate these relationships.

Comment: OK

4. Previous comment: *The authors focus on the 20S proteasome “to reduce the complexity”, however, the implications of this are inadequately discussed. Other forms of the proteasome may also be transported. What is known about the distributions, transport, and functions of 19S, 20S, and 26S proteasomes in nucleus vs. cytoplasm? AKIRIN2 binding likely prevents double capped proteasome from being translocated into the nucleus? The authors should comment on this in greater detail and discuss the limitations of their experiments and of their presented model, as well as how these issues can potentially be addressed in future work.*

Reply: *We agree that focusing on the 20S proteasome is a limitation of this study and appreciate the opportunity to discuss this more thoroughly. The nuclear transport and assembly of different proteasome forms remains incompletely understood, representing an important area for future investigation. Our previous study demonstrated that 19S subunit nuclear import also depends on AKIRIN2, as AKIRIN2 knockout resulted in nuclear exclusion of tagged 19S subunits that were confirmed to be fully incorporated into functional 19S/26S complexes rather than orphaned. However, it remains unclear whether 19S subunits are imported independently or piggyback with 20S proteasomes. Studies in *Xenopus* suggest that 20S import occurs independently from most 19S components (Savulescu et al. 2011, PMID: 21289101), while yeast nuclear import adaptors like Sts1/Cut8 interact directly with 19S components, suggesting species-specific differences. Our quantitative analysis of nuclear versus cytoplasmic distribution shows that 20S proteasomes are more than twice as abundant in the nucleus, while 19S subunits show only 30% nuclear enrichment. This differential distribution suggests that nuclear assembly and regulation of different proteasome forms may involve distinct mechanisms. Tomographic studies by the Baumeister lab suggest that nuclei also contain 30S proteasomes. However, how the equilibrium between different assemblies of the proteasome is regulated remains to be understood despite the decades of research. The nucleus contains specific proteasome activators like PA200 and PA28 γ . Both contain their own NLS and are exclusively nuclear suggesting that they enter independently. Our current model based on 20S import provides a foundation for understanding AKIRIN2-mediated proteasome transport, but future work must address how different proteasome forms and activators are coordinately imported and assembled. The potential steric hindrance of double-capped proteasomes by AKIRIN2 binding represents another important question requiring investigation.*

Comment: OK. Including all of these points briefly would greatly improve the quality of the discussion section, as they represent important questions for follow up.

Reply: We included a paragraph in the discussion section that focuses on the limitations of our study and discusses what’s known about the import of proteasomal activators.

5. Previous comment: The cryoEM structures appear to be of good quality. Is there a potential role for other IPO9 adaptors? The authors highlight some specific features of the structures, e.g., relevance of residue K94 (e.g., its role in stabilizing the interaction with the proteasome), but it should be made clearer what is speculation based on the structure and what is supported by experiment.

Reply: Currently, no other IPO9 adaptor proteins have been identified in the literature, and AKIRIN2 appears to be the primary adaptor mediating proteasome nuclear import. Our nuclear import assays demonstrate that AKIRIN2 together with IPO9 and RanGTP are sufficient for proteasome nuclear import, indicating that additional adaptors are not required for this process. Our comprehensive screen also did not reveal evidence for other essential factors in this pathway. Regarding the structural interpretations, we will clarify what represents experimental validation versus structural speculation. The importance of residue K94 and in fact all highlighted residues is supported by experimental evidence from our saturation mutagenesis screen, where mutations at this position consistently disrupted proteasome import function across all three readouts. The structural analysis provides the molecular basis for this functional requirement, showing that K94 likely stabilizes the AKIRIN2-proteasome interaction through specific contacts visible in our cryoEM structure. We have included a detailed zoom view highlighting these interactions to illustrate the structural basis for K94's functional importance.

Comment: OK

6. Previous comment: The section “Conformational plasticity underpins AKIRIN2-mediated complex assembly” describes the structure of the AKIRIN2-IPO9 binary complex, stating “this structural plasticity illuminates a dynamic assembly mechanism for large nuclear import complexes”. However, if and how this pertains to proteasome import is unclear. Is this structure in absence of proteasome likely to be relevant, considering the variety of states the proteasome can adopt itself? It is not clear to the reviewer what is to be learned from the AKIRIN2-IPO9 binary complex structure.

Reply: The question about the relevance of the AKIRIN2-IPO9 binary complex structure is an important point to address. This structure is crucial for understanding the complete proteasome import mechanism. The binary complex reveals the conformational flexibility of IPO9 when bound to AKIRIN2 alone, which contrasts with the constrained arrangement in the ternary complex. This demonstrates that proteasome binding requires specific conformational selection in the IPO9- AKIRIN2 interface essential for cooperative assembly. The transition from open to closed conformations is a common regulatory mechanism in importins and necessary for understanding the complete import cycle. The binary complex is thermodynamically more stable than the AKIRIN2-proteasome complex, and given that AKIRIN2 concentrations exceed those of proteasomes during import, this binary state may be the most prevalent form in cells. Most importantly, the binary structure elucidates how IPO9 accommodates different AKIRIN2 stoichiometries. While the binary complex shows IPO9 interacting with AKIRIN2 homodimers through one interface set, the ternary complex reveals how conformational rearrangements allow two AKIRIN2 dimers to bind simultaneously through distinct configurations. This structural plasticity is fundamental to understanding how multiple importin complexes are recruited to proteasome import complexes. The binary structure provides essential mechanistic insight into the assembly pathway and reveals the conformational flexibility enabling cooperative binding in the functional import complex.

Comment: The authors comment on the importance of state transitions in regulating importins and the import cycle. The statement that “this binary state may be the most prevalent form in cells” seems tenuous at best, as there does not appear to be experimental data available to support this (?). While we agree that insight into structural plasticity by means of the assembly pathway will be important, the lack of proteasomes in this structure obscures proper interpretation.

Reply: Our IPO9 Co-IP/MS data (Fig. 2b) indicates that AKIRIN2 is one of the most prominent binding partners of IPO9 in comparison to any proteasome subunits. Further, our binding affinity measurements estimated much higher binding affinity between AKIRIN2 and IPO9 than with the 20S. Additionally,

we estimate the cellular concentrations of AKIRIN2 in the same order of magnitude as the one of the proteasome, while IPO9 is a bit lower. Given the massive affinity differences, it is consistent that the binary complex is the most prevalent abundant complex in cells. Finally conformational plasticity is a hallmark of the known transport receptors, where opening and closing of the ring structure is essential for the transport mechanism.

7. Previous comment: It is stated that AKIRIN2 itself is rapidly degraded by nuclear proteasomes in a ubiquitin- independent manner, based on experiments with inhibitors. Is this degradation of AKIRIN2 19S or 20S-dependent?

Reply: Our data strongly suggests that AKIRIN2 degradation is predominantly 20S proteasome-dependent through a ubiquitin-independent mechanism. Evidence supports this conclusion: (1) AKIRIN2 itself is highly turned over by the proteasome which we have seen in our previous study by knocking out a proteasomal subunit, it scored amongst the most enriched proteins. (2) AKIRIN2 contains substantial intrinsically disordered regions, a recognized feature of proteins directly degraded by the 20S core particle without ubiquitination (PMID: 25250704). (3) Our experiments with the E1 inhibitor TAK243 demonstrate that AKIRIN2 degradation occurs independently of ubiquitination. (4) In preliminary in vitro degradation assays (not included in the manuscript), we observed that purified 20S proteasomes generated characteristic GFP-AKIRIN2 degradation products (AKIRIN2 degradation with stable GFP remaining), while 26S proteasomes did not produce these fragments. We included this data for the reviewer (additional Figure 2) as well as a method description (additional information point 2). While we cannot completely exclude some contribution of the 19S regulatory particle through recently described ubiquitin-independent degradation pathways of the 26S proteasomes (PMID: 37616343), our combined data points to AKIRIN2 being primarily a direct 20S substrate. We believe the mechanism of AKIRIN2 degradation merits dedicated investigation beyond the scope of this study, which focuses on its functional role rather than its turnover mechanisms.

Comment: The in vitro degradation data (4) is compelling and we agree that it would be very interesting to explore the mechanism of AKIRIN2 degradation in more detail in future studies. It may be good to mention these preliminary results in the manuscript.

Reply: We have added more data into the supplement of our manuscript, while we want to point out that more experiments are necessary to elucidate this fully.

8. Previous comment: Like most good studies, the work also raises many new questions. The authors provide some statements in the discussion that deserve more scrutiny. For example, a potential role for Midnolin as a trigger in AKIRIN2 degradation is mentioned, however, this appears to be not supported by any data. If this idea is correct, it should be relatively easy to test. Is Midnolin- dependent degradation relevant in the cell types used in this study, i.e., is it a plausible mechanism? Further, the authors mention proteins with analogous functions to AKIRIN2 for other large complexes, and describe a common architecture for these adaptor proteins, by which they could perform multiple critical functions. While interesting conceptually, it may be good to include a sentence on what remains to be demonstrated experimentally for these particular examples.

Reply: We agree that the potential role of Midnolin is purely speculative and rather unlikely, as it lacks the ubiquitous expression pattern of AKIRIN2. We have therefore softened our statement regarding Midnolin-dependent degradation to: 'Whether additional components similar to the recently discovered protein Midnolin are required to target nuclear AKIRIN2 for nuclear degradation remains to be investigated.'

Comment: OK

In the discussion point about nuclear import adaptors, we added a sentence of the remaining questions: 'It remains to be determined whether this proposed common architecture is essential for a nuclear import

adaptor function and whether the constituent building blocks are interchangeable among different adaptors."

Comment: OK

Reviewer #3 (Remarks to the Author):

This is an excellent manuscript, and almost all of my concerns have been addressed. My remaining comments, with the exception of those on the transport assay, are minor. 75 μM GFP-AKIRIN2 in the import assay is a very high concentration. Is this a typo? What is the AKIRIN concentration in the cell? The need for such a high concentration leads to questions of specificity, and should be discussed.

Reply: We are very sorry this was indeed a typo, and the correct concentration is 7.5 μM which we have now corrected in the manuscript. We titrated AKIRIN2 across concentration below and above the estimated K_d (4 μM) for the AKIRIN2-20S proteasomes interaction. At lower concentrations, such as 1 μM , nuclear import of AKIRIN2 was insufficient (Suppl. fig. 8b). We therefore decided to use 7.5 μM in the subsequent experiments to ensure robust import.

While the openCell database estimates the cellular AKIRIN2 levels at ~ 50 nM compared to ~ 2 μM for a 20S proteasome subunit meaning ~ 1 μM of 20S proteasomes, we would like to emphasise that AKIRIN2 is upregulated shortly before mitosis, leading to transient but pronounced increases in its concentration. Although we did not directly quantify AKIRIN2 concentrations across different cell cycle phases, we analysed the lysates from different cell states (mitotic, S, G2, G1 early/late phase) of RKO cells. Immunoblotting, normalized to total protein content, revealed clear peak in AKIRIN2 protein level during mitosis. These findings strongly suggest that acute AKIRIN2 concentrations around mitosis are substantially higher than steady-state estimates, consistent with our previous data showing that proteasome import occurs predominantly shortly after mitosis. We added this data, plus method description to the manuscript to provide a more comprehensive picture.

New Fig. 5c: Immunoblot of RKO cells synchronized to different cell cycle states. Total protein concentration was normalized.

Comment: The NTF2 experiment (additional figure 4b) is very unclear due to faint images, and the data were not quantified. NTF2 is essential for the import of RanGDP (Ribbeck, 1998, EMBO J, 17:6587). If there is no NTF2 dependence, this raises questions about the RanGTP dependence of import. Perhaps they did not understand my previous comment, because it was not addressed. The Ran obtained from *E. coli* cells is isolated as a mixture of RanGTP and RanGDP forms, which can be readily separated on an anion exchange column (RanGDP elutes first). It is not clear if the authors separated these two forms of Ran. If not, the Ran added to the transport assays is a mixture of forms, and the RanGTP in the mix will dissociate import complexes leading to low transport efficiency. RanGTP also

cannot be transported by NTF2, which may explain the lack of an observed effect. Finally, the authors should note that nuclei swell without addition of an osmolyte after digitonin treatment – whether damage occurs upon such swelling is unclear.

Reply: While NTF2 is commonly used to enhance import dynamics, other studies (including work from Prof. Ralph Kehlenbach's laboratory, whose protocols we adopted) have successfully performed RanGTP-dependent import assays without adding NTF2 (or cell lysate), yet still observed robust functional import. We believe this is feasible for several reasons: (1) residual RanGTP is present in the nucleus, (2) residual NTF2 in the nucleus, and (3) in the absence of RanGTP, import complexes can still enter the nucleus but may not be efficiently disassembled. As a result, complexes could undergo multiple rounds of shuttling, leading to a dynamic equilibrium between entry and exit.

In our own assays, the primary purpose was to demonstrate that the identified components are sufficient for proteasome import. This was confirmed by clear loss of import when either AKIRIN2 or the importins were omitted. From our reconstitution experiments, we could also clearly observe that import complexes only form in the absence of RanGTP, consistent with RanGTP dependence. We therefore conclude that the assays fulfil their intended purpose, even without additional NTF2.

We acknowledge the concern regarding the Ran source. In our hands, anion exchange chromatography did not provide a clear separation of RanGDP and RanGTP, likely due to the large quantities we applied. We therefore continued with size-exclusion chromatography, which distinctly resolved two Ran species. As previously reported, RanGDP and RanGTP differ in hydrodynamic radius, enabling such separation. For the nuclear import assays, we exclusively used the second peak, which we assume to be enriched in RanGDP.

We agree that digitonin-permeabilized nuclei can swell or change morphology in the absence of osmolytes. To counteract this, our transport buffer contained 250 mM sucrose. Unfortunately, this information was missing from the methods section due to reliance on an earlier version of the protocol. We have now corrected the method description accordingly.

Minor points:

1) lines 41-44 – soften this statement a bit. The effect of the number of NLSs on nucleoplasmin was investigated by Dingwall (Cell 1982 30:449).

Reply: We changed the text to: 'has not been comprehensively explored'

2) line 82 – 'any possible amino acid' is confusing; perhaps change to 'each of the other 19 amino acids'

Reply: We changed the text accordingly.

3) Fig. 1d – the SYVS-motif is identified as YVS in the figure. Be consistent.

Reply: We unified all to YVS motif.

4) Fig. 2c and elsewhere – BioRad's stainfree gels require a trihalo compound for visualization after photoactivation. Therefore, 'stainfree' is misleading for this approach (recognizing that this is a brandname) as there is a visualizing agent. The approach used should be clarified in the methods.

Reply: We added 'Biorad TGX' in the captions and method description for clarification.

5) line 325 – shouldn't this be Supplementary Fig. 3b,c?

Reply: Thanks, yes indeed we wanted to refer to Supplementary Fig. 3b instead of Supplementary Fig. 3c. We corrected the figure reference.

6) lines 355-56 – why aren't AKIRIN2's NLSs exposed before binding to the proteasome and IPO9. Are they hidden somewhere?

Reply: We agree that this phrasing is confusing. AKIRIN2's NLS are in a disordered region and are presumably always accessible. We changed the text to: In this arrangement, each N-terminal NLS of every AKIRIN2 protomer is accessible for recognition by the canonical import machinery.

7) Supplementary Fig. 3b,c – the markers don't appear accurate, as IPO9 moves from ~100 kD to ~150 kD

Reply: Yes, we agree the gels didn't run perfectly straight and the marker wasn't aligned well enough. We adjusted this in the figure.

8) Supplementary Fig. 3e,f – Since multiple K_d value measurements were made (indicated by the range), what is n? No units are given for k_a and k_d (this was raised previously); no n values are provided either. 4 significant figures for the K_d is unreasonable, and 3 for the k_a and k_d are also likely unrealistic. It is unclear what is shown in the (f) panel (what is the fit and what is data? There seem to be three curves in each group) – for dissociation, the fitted curves don't match well with experiment. This last point was not addressed in the authors' rebuttal.

Reply: Both measurements were done in three replicates which is mentioned in the method description. We additionally now added n=3 to the figure caption. We changed the significant digits to two.

Range of K_d values from MST doesn't come from the multiple measurements performed, but instead from the built-in confidence interval analysis within PALMIST (data analysis software for MST traces) based on the Chi-square of the current best fit. In our case, the confidence interval applied was 0.95.

For GCI measurements, AKIRIN2 was injected in a 1:1 dilution series in the range of 93.75 nM-12 μM. A total of three replicates were measured (shown in Suppl. Fig. 3f as magenta curves). Subsequently, the data was fitted using a one-to-one binding model (Suppl. Fig. 3f black curves). For more precise measurements, higher concentrations of AKIRIN2 would have to be used in this experiment. However, due to properties of AKIRIN2 (tends to aggregate, or phase separate), we cannot reach such concentrations. We can only estimate the affinity of AKIRIN2 for the 20S proteasome, which we show is in the micromolar range.

9) Supplementary Fig. 8 & additional figure 4 – what are the numbers on the top lines of the figure panels? Are these concentrations in micromolar? Time? In 8b and 8c, these are indicated as 20 but in the caption it states 1 μM. Image exposures are very weak, and even expanded it is hard to see what is going on.

Reply: Thank you for catching this. This was a labelling error. In Supplementary Fig. 8 and additional fig. 4, the values should be reported as μM concentrations. For the importins, we inadvertently showed molar excess relative to the proteasome rather than absolute concentration. We have corrected the labels in the revised manuscript and now state the units explicitly, the underlying data and conclusions are unchanged.

10) Supplementary Fig. 9 – what are the three columns? Three trials? What are the concentrations? Why do the c distributions look so different but the others do not? It would be helpful if the key was shown to the right of each row. The format of additional figure 5c seems better.

Reply: Yes, these are three individual measurements of the same sample to visualize variances in the measurements. As we mentioned in the first round of revisions, the measurement of the import complex stoichiometry is quite challenging due to the properties of AKIRIN2 and the complexity of the import complex (20S proteasome which is itself a complex of 28 subunits, AKIRIN2 which forms dimers and binds in multiple copies, and three importins) size differences are rather small and one can see that

peaks sometimes appear rather broad instead of separating into two individual peaks (for two different species).

11) It is not clear why additional figures 3 and 4a are not in the Supplementary Data.

Reply: We included additional figure 3 as Suppl. fig. 7c, additional Fig. 4a is already included in Suppl. fig. 8b excluding the channels for AKIRIN2.

Reviewer #2 (Remarks to the Author):

The authors addressed most of my questions. The authors have demonstrated clearly that AKIRIN2 regulates nuclear localization of the 20S and degradation of nuclear substrates, which are essential for cell growth. I did not question this connection but wonder if AKIRIN2 may regulate additional factors contributing to the phenotypes. Since the defects of depleting proteasome from the nucleus are very strong, this question should best be addressed by a rescue experiment. This type of experiment has been performed using nuclear anchors (e.g. Tsuchiya H. et al Biochemical and Biophysical Research Communications 436 (2013) 372–376). But I agree implementing this strategy in their system may take some time. So, I am satisfied with their current data in this part.

Reply: We agree that this would be an interesting follow-up study.

AKIRIN2's dependency score not only has a low correlation with those of proteasome subunits among cancer cell lines, but is also much lower. The score falls out of the "essential gene" range (<-1) in ~50% of cancer cell lines. Since proteasome contains many important targets in the nucleus, perhaps the most likely explanation is that there are mechanisms rescuing proteasome localization in the absence of AKIRIN2. The author should discuss this.

Reply: In our current and previous study there is no evidence that there are rescue mechanisms in RKO, K562, or MIA-PaCa-2 cells which is also supported by a study in C.elegans (doi: 10.1016/j.isci.2023.107886). If there are other pathways in different cell types remains to be studied. We have also screened 40 more cell types through other studies and find AKIRIN2 to be essential in all of these. Importantly, we have also found that the success of a CRISPR knockout of AKIRIN2 is especially depended on the choice of sgRNAs where there is large diversity in the data accumulated by Depmap. Without having large folded domains several perturbations may not lead to large effect sizes. However, when we use a robust strategy using a dual guide approach we find significant strong effect on any cell line tested. These effects are also reflected in the fact that Depmap scores the gene as common essential in 1146 out of 1186 cell lines. Proteasome naturally are much better targets for any kind of guide because all subunits are tightly folded domains and this give generally higher scores.

Minor comment: please check if the literature references are up to date. For example, it appears the cited preprint by Monda et al. (Ref. 21) is currently published in a peer-reviewed journal.

Reply: Reference was updated.

Reviewer #3 (Remarks to the Author):

My concerns involving NTF2 and RanGTP have not been fully addressed. In their reply, the authors indicate that RanGDP and RanGTP differ in hydrodynamic radius, but they did not provide a reference that this is sufficient to separate on a size-exclusion column. They assumed that the second peak was RanGDP, but they did not do any experiments to verify this. My expectation is that RanGTP would have the 'closed' conformation, and hence would elute second on a size-exclusion column. If they add RanGTP to their import mix, any transport complexes should disassemble, reducing import. NTF2 should significantly increase the Ran gradient by importing RanGDP. NTF2 does not import RanGTP. Import can indeed be observed in the absence of RanGTP and NTF2 since

correctly assembled import complexes can transport independent of these molecules, and then get stuck in the nucleus by binding to nuclear macromolecules.

Reply: We performed an experiment to test our 'RanGDP' purification by running gradients with IPO9 in comparison to RanGTP. Here we could clearly see that the RanGTP binds to IPO9 by a shift and a clear peak of RanGTP. Our purified 'RanGDP' didn't peak in these fractions nor did it lead to shift of IPO9 to the higher molecular weight fractions. We believe this should prove that the 'RanGDP' which we used for the nuclear import assays is predominantly GDP bound. We agree that the nuclear import assays could still be improved with a functional NTF2 Ran cycle which we didn't achieve but will work on in future experiments.

The images in the bottom row of Fig. 4c are so weak that I cannot interpret what is going on. These must be made brighter. My impression is that there is no improvement in import with KPNA1 and KPNA2; in fact, the IPO9 only data looks like more import. These data are not quantitatively compared. Due to the RanGTP and NTF2 issues discussed above, differences in the import efficiencies and import rates with the various importins could become more apparent if these are done properly. The import data as-is indicates inefficient transport and does not convincingly indicate what combination of the tested transport factors yields the best transport. I agree with the authors that this is not of central importance to their story as their gel data is convincing, but this is an opportunity missed, at the very least.

Reply: The image brightness was increased for images 2,4,5 and 6, and we mention in the caption that there's a different threshold. Yes, we agree that we didn't see an improvement by adding KPNA2/KPNA1, but we also observe import if we only add KPNA2/KPNA1 in the absence of IPO9. We believe that these import assays show the transport capability of these importins to the proteasome cargo, however in the cell the efficiency and dynamics depend on the cellular concentrations and the interplay between the different importins.

The images in Sup Fig. 8a,b are also very dim. These should also be made brighter and enlarged (fewer cells) so the reader does not have to strain to see what is going on. Use of the nucleoplasm/cytoplasm intensity ratio is problematic here. The 0,10 image in B clearly has dark nuclei, so the ratio cannot be 1 as indicated in C. Moreover, the proteasomes clearly bind very tightly to the cytoplasmic compartment, so the denominator signal is artificially high and dependent on bound transport factors (clearly observed in Fig. 4). This raises questions about the quantification in Sup Fig. 8c.

Reply: The image brightness was increased was increased for Suppl. Fig. 8a and b.

Minor points:

1) Note to authors: Sucrose does not adequately prevent nuclear swelling, as it is smaller than the size-exclusion limit of NPCs. This is information, it does not need to be addressed.

Reply: We acknowledge the reviewer's comment.

2) Sup Fig. 3f – the black fits to the dissociation data are very poor: straight lines fitting to obvious curves. The model is clearly wrong here.

Reply: As we mentioned before we couldn't reach higher concentrations of AKIRIN2 which behaved poorly. We discussed the model fitting with several experts and agreed that the chosen model is the best we can do here.

3) Ref. 10 on line 55 does not appear to be correct.

Reply: That's right. Reference table was updated

Reviewer #4 (Remarks to the Author): I co-reviewed this manuscript with one of the reviewers who provided the listed reports. This is part of the Nature Communications initiative to facilitate training in peer review and to provide appropriate recognition for Early Career Researchers who co-review manuscripts.

Reply: We thank the reviewer for their revising our manuscript.